# Giant near-field nonlinear electrophotonic effects in an angstrom-scale plasmonic junction

Shota Takahashi [1], Atsunori Sakurai [1,2] ✉, Tatsuto Mochizuki [1,2] &
Toshiki Sugimoto [1,2] ✉

Plasmons facilitate a strong confinement and enhancement of near-field light, offering exciting opportunities to enhance nonlinear optical responses at the nanoscale. However, despite significant advancements, the electrically tunable range of the nonlinear optical responses at nanometer-scale plasmonic structures remains limited to a few percents per volt. Here, we transcend the limitation of the nanometer regime by expanding the concept of electro-photonics into angstrom-scale platform, enabling high-performance modulation of near-field nonlinear optical responses inaccessible in prior architectures. We demonstrate ~2000% enhancement in second-harmonic generation (SHG) within 1 V of voltage application by utilizing an angstrom-scale plasmonic gap between a metallic tip and a flat metal substrate in a scanning tunneling microscope. Extending this near-field SHG scheme to sum-frequency generation that is accompanied by large frequency upconversion, we also found that such giant electrical modulation of plasmon-enhanced nonlinear optical phenomena is effective over mid-infrared to visible broad wavelength range. Our results and concepts lay the foundation for developing near-field-based angstrom-scale nonlinear electrophotonics with significant modulation depth at low driving voltage.

An ultimate goal in photonics is to design photonic phenomena with desired functions by intentionally controlling light–matter interactions. While the photonic functionalities are fundamentally determined and limited by the intrinsic material properties such as permittivity and conductivity, much effort has been devoted to tune and exploit those functionalities. In particular, tailoring nonlinear optical effects is of paramount importance with the growing demand for diverse photonic functionalities, such as light frequency conversion[1], all-optical switching[2], optical imaging[3], and spectroscopic analysis of materials[4,5]. Since the nonlinear optical processes depend superlinearly on the incident electric field, integrating with plasmons that intensely confine and enhance light within nanometric volumes should dramatically augment near-field nonlinear optical phenomena at the nanoscale beyond the diffraction limit. This concept underpins a

fascinating research field of nonlinear plasmonics[6–10], encompassing various applications such as enhanced up/down conversion of optical frequency[11–13], plasmonic sensing of local environments[14–17], and nonlinear optical imaging of nanomaterials[18–25]. These plasmonic functionalities are fundamentally governed by the intrinsic optical properties of constituent materials and the static geometry of the plasmonic structures, such as size, shape, material composition, and surrounding environment[26–31]. Although these static structural properties are typically difficult to adjust at their post-fabrication stage, designing active plasmonic systems that respond to external stimuli, such as light, heat, mechanical force, or electric field, provides a route toward active modulation of both linear[32–39] and nonlinear[40–46] plasmonic responses. Since such tunable plasmonic platforms are essential for advanced nanophotonic applications[47,48], approaches that allow

[1]Institute for Molecular Science, National Institutes of Natural Sciences, Okazaki, Aichi, Japan. [2]Graduate Institute for Advanced Studies, SOKENDAI, Okazaki, Aichi, Japan. ✉e-mail: asakurai@ims.ac.jp; toshiki-sugimoto@ims.ac.jp

more flexible and efficient active control of plasmonic responses are highly desirable.

One promising strategy for active control of plasmonic responses is to directly exploit the metallic nature of plasmonic systems. Metal plasmonic structures can serve not only as light-enhancing media in a near-field optical scheme but also as electrodes that can facilitate various electrical perturbations[49–53]. These dual optical and electrical functionalities provide an interesting platform for electrical modulation of plasmonic nonlinear optical processes, which are critical for on-chip integrated optoelectronics, including tunable nanolasers[54–56] and optical modulators[57,58]. By applying the voltage across plasmonic gap structures, an electrostatic field is generated within these gaps, which can affect the plasmonic nonlinear optical properties inside the gaps. Although several studies have successfully reported the possibility of the plasmonic gap-based modulation[40–44], most have focused on relatively wide (sub-100 nm) gap structures, as fabricating and maintaining angstrom-scale narrow gaps remains a significant technical challenge. Consequently, while valuable progress has been made, the reported modulation depths have typically been limited to less than a 10% signal increase per volt[40–44]. Therefore, a practical level of electric modulation reaching 1000% signal increase has typically required as high as ~100 V of voltage application, preventing the application from being used in a realistic device.

In this study, we expand the platform of electrophotonics from the sub-100 nm into the angstrom regime, unveiling the potential of an angstrom-scale plasmonic junction as a groundbreaking platform for electrophotonic control of near-field nonlinear optical effects. As a first experimental realization of such angstrom-scale electrophotonics, we leverage a voltage-applicable and precisely position-controllable plasmonic gold tip integrated with a scanning tunneling microscope (STM). This system achieves giant electrical modulation of plasmonic nonlinear optical responses, demonstrating a ~2000% signal enhancement with a minimal voltage sweep of 1 V. Our work represents the first experimental realization of this near-field nonlinear electrophotonic effect in second-harmonic generation (SHG) excited by femtosecond near-infrared (IR) laser pulses. Moreover, by temporally and spatially superimposing the near- and mid-IR laser pulses, we extend this effect to the sum-frequency generation (SFG) process that is accompanied by large frequency upconversion from mid-IR to the visible region. Showing broadband operability of the gigantic near-field nonlinear electrophotonic effects, our work shows great potential for advancing functionalities in angstrom-scale electrophotonic systems.

## Results
### Angstrom-scale platform of nonlinear plasmonics
SHG is a second-order nonlinear optical process in which two photons with identical frequencies interact with a material and are converted into a single photon with doubled frequency of the original photons

(Fig. 1a). The possibility of electric modulation of the near-field SHG signals from angstrom-scale plasmonic gap was explored in the experimental setup illustrated in Fig. 1c. For our first demonstration, we adopted a recently constructed experimental platform based on an angstrom-scale gap that consisted of an electrochemically etched Au tip[59] (Fig. 1b) and an atomically flat Au(111) substrate equipped in an ultra-high vacuum STM unit ($<1 \times 10^{-7}$ Pa)[60–63]. In this system, we can control the distance of the plasmonic gap between the tip apex and the substrate at the angstrom scale by precisely regulating the tunneling current (Supplementary Notes 3 and 4). Prior to the SHG experiments, the tip underwent Ar⁺ sputtering for 3 h to clean the tip apex[64]. This sputtering enabled reproducible nonlinear optical experiments presented below, as well as clear STM imaging.

In the gap, ~6 Å-thick self-assembled monolayer (SAM) of 4-methylbenzenthiol (MBT) was formed on the Au substrate as a model ultrathin dielectric layer with well-defined homogenous structure[65] (see Supplementary Fig. 1 for the STM image of an MBT SAM). To induce the tip-enhanced SHG (TE-SHG) process, we exposed the gap to near-IR excitation pulses (1500 nm, 280 fs, FWHM: 19 nm) with p-polarization at a high repetition rate (50 MHz). In general, when the optical structure is considerably smaller than the optical wavelength, the phase-matching condition breaks down, resulting in a dipole-like radiation pattern of the output signal[66]. Thus, we collected TE-SHG emissions separately in forward- and backward-scattering directions (see the "Methods" section for details). As reported in our recent study[60], the TE-SHG output exhibits long-time stability and quadratic dependence on the excitation intensity, provided the excitation power is maintained sufficiently low ($\leq 0.7$ mW). Therefore, to ensure minimal signal fluctuations, the excitation intensity in all TE-SHG measurements presented in the subsequent sections was set to 0.5 mW (10 pJ/pulse). Indeed, employing this intensity allowed us to obtain clear STM images even under light irradiation (Supplementary Fig. 11), ensuring that the optical damaging effect induced by the laser pulse is sufficiently suppressed[60].

### Near-field SHG and its giant electric modulation
When the substrate was retracted from the tip by ~30 nm, thereby deactivating the plasmonic excitation, no backward-scattered SHG signal was detected (gray curve in Fig. 2a). Owing to the phase-matching condition of far-field nonlinear optical processes, normal far-field SHG signal from the flat Au(111) substrate without plasmonic excitation was detected only in the forward-scattering direction (Supplementary Fig. 7). Although tip plasmons of the tip apex alone could be excited even in the 30 nm-retracted condition, their contribution can be safely disregarded because the backward-scattered signal emission is absent despite identical power and integration time for the measurements of forward- and backward-scattered signals (Supplementary Note 5). The minimal contribution from the tip plasmon was further corroborated by our numerical simulation, revealing

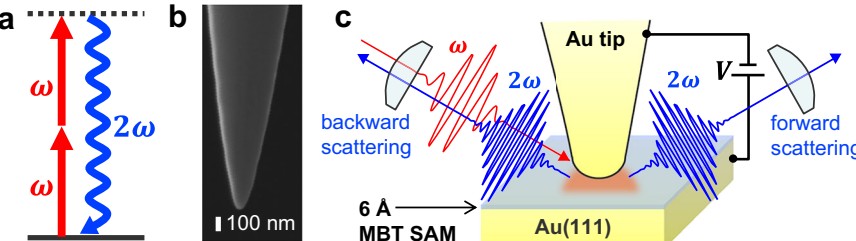

**Fig. 1 | Near-field SHG experiments in the angstrom-scale plasmonic junction of STM. a** Energy level diagram of the SHG process. **b** Scanning electron micrograph of the Au tip used in the experiments. **c** Schematic depiction of near-field SHG experiment conducted under room temperature and ultra-high vacuum ($<1 \times 10^{-7}$ Pa) conditions. A Au tip and a Au substrate were mounted on an STM unit

and formed an angstrom-scale gap with an applied bias voltage $V$. An ultrathin (~6 Å) dielectric layer composed of MBT SAM was formed on the Au substrate. The gap region was irradiated by a femtosecond near-IR laser with a frequency of $\omega$, and near-field SHG with a frequency of $2\omega$ was detected in both forward- and backward-scattering geometries.

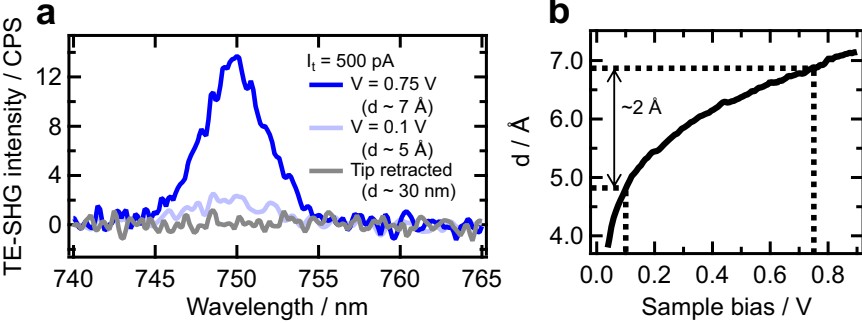

**Fig. 2 | Tip-enhanced SHG and its relationship with the applied bias and tip–substrate distance. a** The spectra of TE-SHG excited by 1500 nm laser pulses and obtained at sample biases of 0.1 V (light blue, $d$ ~ 5 Å) and 0.75 V (dark blue, $d$ ~ 7 Å) with a constant tunneling current of 500 pA. The tip–substrate distance $d$ is defined as the metal-to-metal separation between the tip apex and the substrate surface. Gray curve indicates the signal obtained when the substrate was retracted enough from the tip ($d$ ~ 30 nm) to deactivate the plasmonic enhancement effects. **b** Dependence of the tip–substrate distance $d$ on the sample bias. The measurement was performed at a constant tunneling current of 500 pA. The voltage increase from 0.1 to 0.75 V under this constant current condition elongated the tip–substrate distance from ~5 to ~7 Å. For further details of the estimation of the absolute tip–substrate gap distance, see Supplementary Note 3.

that the plasmonic enhancement caused by a tip alone is more than one order of magnitude smaller than that in an angstrom-scale gap (Supplementary Fig. 18).

Then, the tip-substrate distance ($d$), defined as the metal-to-metal separation between the tip apex and the substrate surface, was reduced from ~30 nm to ~5 Å under the sample bias of 0.1 V. In this condition, not only did the intensity of the forward-scattered SHG signal increase (light blue curve in Supplementary Fig. 7), but also the backward-scattered SHG signal newly appeared (light blue curve in Fig. 2a). In this case, since the movement of the piezoelectric stage is quite small (~30 nm) compared to the optical wavelength of light, the signal collection efficiency should remain unchanged even after the formation of the angstrom-scale junction. Moreover, while the tip apex is expected to slightly indent into the SAM layer at a tip–substrate distance of ~5 Å, the potential influence of such contact, including chemical enhancement effects[67], should be negligibly small because the terminal methyl group of MBT molecules is chemically inert (Supplementary Note 3). Therefore, the signal increase in the forward-scattering geometry (light blue curve in Supplementary Fig. 7) and the appearance of the backward-scattered signal (light blue curve in Fig. 2a) are obviously due to the near-field enhancement activated within the angstrom-scale tip-substrate gap, and thus the observed signals correspond to TE-SHG emission. Note that the technical basis for detecting such tip-enhanced nonlinear optical signals separately from far-field signals has already been established and reported in our recent publications[60–63].

While we also detected TE-SHG from the bare Au(111) surface without molecular adsorption (Supplementary Fig. 13), the observed intensity was much lower than that observed for the Au substrate coated with MBT SAM. We consider that the TE-SHG signal observed in the absence of the SAM is attributed to the nonlinear optical response of the surface electrons of the Au tip and substrate. Since the electric field enhancement at the central region of the gap is expected to be much stronger than that near the top surfaces of the Au substrate and Au tip[68], the SAM film placed within the gap region can produce significantly stronger TE-SHG emission than the surface electrons of the Au tip and substrate. Therefore, the main source of TE-SHG signals at 0.1 V shown in Fig. 2a and Supplementary Fig. 7 is the vibrationally non-resonant $\chi^{(2)}$ response of the MBT SAM, which is amplified through the field enhancement effect caused by the tip–substrate gap (Supplementary Note 10). In this case, the observed spectral shape of TE-SHG reflects that of the near-IR excitation pulses (Supplementary Fig. 2).

As also shown in Fig. 2a and Supplementary Fig. 7, we found that the TE-SHG intensity markedly increased when the sample bias ($V$) was increased from 0.1 to 0.75 V, with the tunneling current setpoint maintained constant (500 pA). This voltage increase at a constant tunneling current (500 pA) extended the tip–sample distance from ~5 to ~7 Å (Fig. 2b). Based on conventional classical electrodynamic simulations, the near-field enhancement strength is expected to decrease by several tens of percent during this ~2 Å tip–substrate distance elongation[69–74]. However, at the gap distances of <1 nm, the influences of quantum mechanical phenomena, such as electron spill-out from the metal surface and the overlap of electronic wavefunctions across the gap, begin to quench plasmon excitations and suppress the increase in field enhancement[68,71,72,75–78]. Particularly, substantially short tip–substrate distance of <4 Å leads to a regime where such quantum plasmonic quenching becomes dominant, resulting in a steep decrease in the electric field enhancement factors[68,71,72,75–78]. In contrast, the 4–7 Å regime represents a crossover region just before this steep decline. Thus, in this distance range, the quantum suppression effects and the classically expected enhancement nearly cancel each other, making the overall field enhancement effectively independent of the tip–substrate distance[68,71,72,75–78]. Therefore, the near-field enhancement strength should remain essentially unaffected during the distance elongation from 5 to 7 Å. Despite such constant field enhancement, the TE-SHG intensity significantly increased with the applied voltage (Fig. 2a). Importantly, this distance expansion occurred when the bias voltage was raised from 0.1 to 0.75 V at a constant tunneling current (500 pA). With this voltage increase, the electrostatic field below the tip apex ($V/d$) increased from $2 \times 10^8$ to $1 \times 10^9$ V m$^{-1}$. Therefore, both the TE-SHG intensity and the electrostatic field increased concurrently, indicating the presence of substantial electrostatic field-induced effects that boost the overall TE-SHG intensity.

To gain more quantitative insights into the field-effect modulation of the TE-SHG intensity, we fixed the tip–substrate distance at $d$ ~ 7 Å and monitored the voltage dependent variation in the TE-SHG intensity (Fig. 3). At this tip–substrate distance, the classically predicted near-field enhancement and the quantum plasmonic quenching effects arrive at a nearly optimal balance, resulting in almost maximum near-field enhancement values[68,71,72,75–78]. Moreover, as long as the tip–substrate distance is kept constant, variations in the STM bias voltage on the order of 1 V essentially do not affect the near-field-enhancement factor within the gap[79]. Therefore, fixing the tip–substrate distance at ~7 Å can be regarded as the optimal condition under which the field enhancement can be maintained at its nearly maximum value regardless of the applied bias voltage. This enables us to exclusively examine the influence of electrostatic fields on the TE-

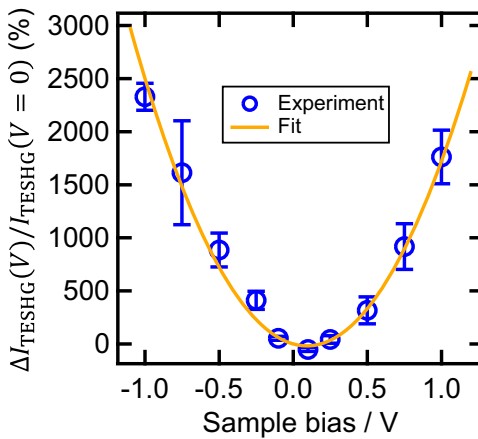

**Fig. 3 | Giant electric modulation of TE-SHG.** The voltage-dependent change in the TE-SHG intensity ($\Delta I_{\text{TESHG}}(V)$) normalized by the signal intensity at $V = 0$ V ($I_{\text{TESHG}}(V = 0)$) is depicted by blue open circles. The reference intensity $I_{\text{TESHG}}(V = 0)$ was approximated as the average TE-SHG intensities obtained at small positive and negative biases ($\pm 0.1$ V): $I_{\text{TESHG}}(V = 0) \approx [I_{\text{TESHG}}(V = +0.1 \text{ V}) + I_{\text{TESHG}}(V = -0.1 \text{ V})]/2$. The modulation depth of ~2000% is achieved at a bias voltage of $\pm 1$ V. The orange curve is the result of curve fitting with a quadratic function. The measurements were performed in the backward-scattering geometry under constant excitation intensity (0.5 mW) and constant tip–substrate distance (~7 Å). The error bars represent the standard error of the mean determined from three independent repetitions of the same voltage-dependent measurement.

SHG process, while eliminating the contributions of variations in the near-field enhancement. Note that the constant tip–substrate distance of 7 Å was kept by adjusting the tunneling current setpoint in the range of 40–1750 pA synchronously with the voltage sweep (Supplementary Note 4). During the voltage sweep, the excitation intensity of the incident laser was maintained low and constant at 0.5 mW. Despite such fixed tip–substrate distance and excitation power, the TE-SHG intensity significantly grew up when increasing the sample bias voltage from 0.1 to 1 V (Fig. 3). Importantly, such a significant TE-SHG signal modulation was also observed when we swept the bias voltage inversely from −0.1 to −1 V, exhibiting a quadratic dependence on the applied voltage (Fig. 3). More remarkably, the experimentally obtained relative modulation in the TE-SHG signal intensity ($\Delta I_{\text{TESHG}}(V)/I_{\text{TESHG}}(V = 0)$) reached ~2000% at $V = \pm 1$ V (the blue open circles in Fig. 3), indicating the achievement of ~2000% $V^{-1}$ giant modulation depth. The stability and reproducibility of these results were confirmed through repeated measurements as described in detail in Supplementary Notes 6 and 12.

It should be noted that while the quantum plasmonic effects in an angstrom-scale junction have been discussed primarily by referring to previous studies[68,71,72,75–78] as qualitative support, the presence of such quantum plasmonic effects can also be ensured from our experimental observations. As shown in Supplementary Fig. 14, voltage-controlled quadratic modulation of TE-SHG, similar to that shown in Fig. 3, was also observed under the constant tunneling current conditions, where the tip–substrate distance varies in the range of 4–6 Å during the voltage-dependent measurement. This clearly demonstrates that the overall field-enhancement strength remains essentially insensitive to the tip–substrate distance within the experimentally varied 4–6 Å distance range. Therefore, the presence of quantum plasmonic effects that cause distance-invariant field-enhancement behavior is supported not only by previous studies[68,71,72,75–78] but also by our experimental results (Fig. 3 and Supplementary Fig. 14). Moreover, the agreement of the bias-dependent profiles shown in Fig. 3 and Supplementary Fig. 14 also indicates negligible contributions of tunneling current to the observed variations in the TE-SHG intensity (Fig. 3). Consequently, we can reasonably conclude that the observed electrical modulation

behavior is predominantly governed by voltage-induced electro-photonic effects, distinct from variations in the near-field enhancement or contributions from the tunneling current.

Notably, the observed near-field nonlinear electrophotonic effects, characterized by a quadratic response and a large modulation depth of ~2000% $V^{-1}$, are in stark contrast with previous studies[40–44]. Over the past decade, electric-field modulations in plasmon-enhanced near-field nonlinear optics have been pioneered with sub-100 nm scale gap structures, within which organic molecules or inorganic insulators were embedded as nonlinear optical media. However, due to the inherent challenges in fabricating angstrom-scale gaps, the sub-100 nm scale structures used in these previous studies exhibited modulation depths of the order of 10% $V^{-1}$, with a linear response to the applied voltage[40–44]. In contrast, our angstrom-scale plasmonic gap occupied by a molecular SAM exhibits a quadratic voltage response with a significantly large electrical modulation depth of 2000% $V^{-1}$. Moreover, we confirmed that the surface electrons of gold also generate TE-SHG signals even without the SAM and exhibits a similar quadratic voltage dependence with an electrical modulation depth of ~1000% $V^{-1}$ (Supplementary Fig. 13). We consider that the giant TE-SHG modulation observed for the bare Au(111) surface (Supplementary Fig. 13) is likely attributed to the electric field-induced change in the free electron density at the topmost Au surface[80]. These results underscore that the angstrom-scale plasmonic gap serves as a medium-independent platform for achieving substantial nonlinear optical modulation depths of at least ~1000% $V^{-1}$ with a modest bias of ≤1 V.

Given that no structural changes were observed in the MBT SAM sample (Supplementary Fig. 12), the contribution of field-induced phase transitions or polarization-switching phenomena[81–84] can be considered insignificant. Consequently, the observed gigantic modulation of TE-SHG signals is attributed primarily to electro-optic interactions triggered by the electro-static field within the plasmonic gap ($E_{\text{DC}}$). Such electro-optic interaction in the second-order SHG output is likely to be induced by the third-order nonlinear effect of molecules, representing the mixed interaction between the incident light $E_{\text{gap}}(\omega)$ and the electrostatic field $E_{\text{DC}}$, $P^{(3)}(2\omega) = \varepsilon_0 \chi^{(3)}(2\omega; 0, \omega, \omega) E_{\text{DC}} E_{\text{gap}}(\omega) E_{\text{gap}}(\omega)$. In this case, the total TE-SHG signal intensity ($I_{\text{TESHG}}$) includes contributions not only from the second-order nonlinear polarization ($P^{(2)}(2\omega) = \varepsilon_0 \chi^{(2)}(2\omega; \omega, \omega) E_{\text{gap}}(\omega) E_{\text{gap}}(\omega)$) but also from the electrically tunable third-order nonlinear polarization ($P^{(3)}(2\omega)$), as follows:

$$I_{\text{TESHG}} \propto \left| \chi^{(2)} + \chi^{(3)} E_{\text{DC}} \right|^2 I_{\text{gap}}^2, \tag{1}$$

where $I_{\text{gap}}$ denotes the intensity of the tip-enhanced electric field of the incident excitation pulses ($I_{\text{gap}} \propto \left| E_{\text{gap}}(\omega) \right|^2$); and $\chi^{(2)}$ and $\chi^{(3)}$ are second- and third-order nonlinear optical susceptibility, respectively, of the medium sensing the enhanced-electric field in the angstrom-scale gap. Note that our recent work demonstrated that, when the tip apex size is on the order of several tens of nanometers as in the present study (Fig. 1b), the generated nonlinear polarizations are predominantly dipolar in nature, with negligible contributions from quadrupolar or higher-order multipolar components (Supplementary Note 14)[63]. Moreover, since such relatively large tip apex can support only dipolar plasmonic mode (Supplementary Fig. 21)[74], TE-SHG signal radiation observed in Figs. 2 and 3 should be mediated by dipolar plasmons (Supplementary Notes 13 and 14).

The incorporation of the field-induced $\chi^{(3)} E_{\text{DC}}$ term in Eq. (1) is referred to as the electric-field-induced second-harmonic (EFISH) effect. Notably, during the bias-dependent TE-SHG measurements (Fig. 3), the excitation intensity was fixed at 0.5 mW, and the field enhancement strength within the gap was kept constant by

maintaining a fixed tip–substrate distance (~7 Å). Under these controlled conditions, $I_{gap}$ in Eq. (1) can be regarded as constant throughout the bias-dependent TE-SHG experiments. Therefore, the observed substantial bias dependence (Fig. 3) represents the first experimental achievement in observing a giant near-field EFISH effect arising from the $\chi^{(3)}E_{DC}$ term in an angstrom-scale plasmonic gap structure, with sufficient stability and reproducibility (Supplementary Notes 6 and 12). Note that since the intragap fields in Eq. (1) ($E_{gap}$ and $E_{DC}$) are dominated by surface-normal ($Z$-directed) components (Supplementary Figs. 20 and 22), among the multiple tensor elements of $\chi^{(2)}$ and $\chi^{(3)}$, only $\chi^{(2)}_{ZZZ}$ and $\chi^{(3)}_{ZZZZ}$ components dominantly contribute to the generation of TE-SHG signals (Supplementary Note 14).

By expanding the right-hand side of Eq. (1), we obtain a voltage-independent term ($I_{TESHG}(V=0) = |\chi^{(2)}|^2 I_{gap}{}^2$) and voltage-dependent terms ($\Delta I_{TESHG}(V) = [|\chi^{(3)}|^2 E_{DC}^2 + 2Re(\chi^{(3)}\chi^{(2)*})E_{DC}]I_{gap}{}^2$). When the contribution from $\chi^{(3)}E_{DC}$ is significantly smaller than that from $\chi^{(2)}$, the voltage-dependent component is dominated by the linear $E_{DC}$ term[40–44,85,86], resulting in the modest electrical modulation depth on the order of at most 10% V$^{-1}$. In contrast, when the $\chi^{(3)}E_{DC}$ term far exceeds the $\chi^{(2)}$ term, the voltage-dependent component becomes dominated by the quadratic $E_{DC}$ term, leading to significantly enhanced modulation depth exceeding 1000% V$^{-1}$. Previous studies employing sub-100-nm plasmonic gap structures generated relatively modest electric fields (~10$^7$ V m$^{-1}$), resulting in linear voltage dependence with modest modulation depths (~10% V$^{-1}$)[40–44,85,86]. In this case, a high bias on the order of 100 V is required to achieve a modulation depth of ~1000%. In contrast, our approach leveraged an angstrom-scale plasmonic gap ($d < 10$ Å), facilitating a much more intense electrostatic field (~10$^9$ V m$^{-1}$) even under a moderate voltage application (~1 V), within a range that does not damage metal electrodes or molecules[87–89]. This resulted in a dominant contribution from the $\chi^{(3)}E_{DC}$ term, achieving a significantly larger modulation depth on the order of 1000% V$^{-1}$ with a quadratic voltage dependence. These findings highlight a pronounced near-field nonlinear electrophotonic effect unique to angstrom-scale gap structures.

Notably, the optical near field $E_{gap}$ and the electrostatic field $E_{DC}$ inherently exhibit non-uniform spatial distributions due to the curved geometry of the tip apex (Supplementary Figs. 20 and 22), thereby giving rise to position-dependent nonlinear polarizations within the nanoscale region beneath the tip apex. In such conditions, the TE-SHG electric fields emitted from these individual nonlinear polarizations depend on their absolute spatial position, and the experimentally observed TE-SHG intensity and its electrophotonic modulation arise from the coherent sum of these position-dependent contributions. Specifically, as described in Supplementary Note 15, the overall TE-SHG intensity ($I_{TESHG}$), including the contributions from both the bias-independent second-order signal and the $E_{DC}$-induced third-order signal, is given by the total intensity of SHG fields emitted from $N$ molecules within the gap:

$$I_{TESHG} \propto N^2 \left| \chi^{(2)} \left\langle L_{gap}(\mathbf{r}_n) K_{gap,Z}(\mathbf{r}_n)^2 \right\rangle + \chi^{(3)} \left\langle \frac{L_{gap}(\mathbf{r}_n) K_{gap,Z}(\mathbf{r}_n)^2}{d(\mathbf{r}_n)} \right\rangle V \right|^2 I_0^2, \tag{2}$$

where $\langle \cdots \rangle$ denotes spatial averaging over molecules present within the gap region; $\mathbf{r}_n$ is the position of $n$th molecule; $d(\mathbf{r}_n)$ represents the position-dependent tip–substrate distance determined by the curvature radius of the tip apex; $K_{gap,Z}(\mathbf{r}_n)$ is an enhancement factor of $Z$-directed field; $L_{gap}(\mathbf{r}_n)$ is efficiency of radiation from a nonlinear polarization located at the position $\mathbf{r}_n$; and $I_0$ is incident field intensity. To simplify this formulation, we introduce an effective tip–substrate

distance $d_{eff}$, which satisfies the relation

$$\left\langle \frac{L_{gap}(\mathbf{r}_n) K_{gap,Z}(\mathbf{r}_n)^2}{d(\mathbf{r}_n)} \right\rangle = \frac{\langle L_{gap}(\mathbf{r}_n) K_{gap,Z}(\mathbf{r}_n)^2 \rangle}{d_{eff}}. \tag{3}$$

Substituting this expression into Eq. (2) yields

$$I_{TESHG} \propto N^2 \left| \chi^{(2)} + \chi^{(3)} \left( \frac{V}{d_{eff}} \right) \right|^2 \left| \left\langle L_{gap}(\mathbf{r}_n) K_{gap,Z}(\mathbf{r}_n)^2 \right\rangle \right|^2 I_0^2. \tag{4}$$

By defining $V/d_{eff}$ term as $E_{DC,eff}$ and omitting $N^2$ and $|\langle L_{gap}(\mathbf{r}_n) K_{gap,Z}(\mathbf{r}_n)^2 \rangle|^2 I_0^2$ terms for simplicity, we arrive at the simplified expression of EFISH similar to that given in Eq. (1). Notably, $d_{eff}$ defined in Eq. (3) represents an effective spatial average of the tip–substrate distance within the gap, weighted by the near-field intensity distribution and the curvature of the tip apex. Thus, the expression of $E_{DC,eff} = V/d_{eff}$ effectively incorporates the influences of spatial field inhomogeneity. Moreover, even in the presence of such spatial inhomogeneity, the effective electrostatic field $E_{DC,eff}$ remains linearly proportional to the bias voltage $V$. Consequently, the overall TE-SHG intensity still exhibits a quadratic dependence on $V$, and thus the experimentally observed parabolic voltage dependence of TE-SHG intensity (Fig. 3) is a physically valid behavior.

Based on Eq. (4), the bias-dependent modulation of TE-SHG intensity is expressed as

$$\frac{\Delta I_{TESHG}(V)}{I_{TESHG}(V=0)} = \frac{|\chi^{(3)}|^2}{|\chi^{(2)}|^2} \left( \frac{V}{d_{eff}} \right)^2 + \frac{2Re(\chi^{(3)}\chi^{(2)*})}{|\chi^{(2)}|^2} \left( \frac{V}{d_{eff}} \right) \tag{5}$$

The fitting analysis of the bias-dependent intensity change shown in Fig. 3 using this equation yielded the coefficients of $V^2$ and $V$ in Eq. (5) as $|\chi^{(3)}|^2/d_{eff}^2|\chi^{(2)}|^2 = (21.1 \pm 0.7)$ V$^{-2}$ and $2Re(\chi^{(3)}\chi^{(2)*})/d_{eff}|\chi^{(2)}|^2 = (-3.9 \pm 0.6)$ V$^{-1}$, respectively. Moreover, for a tip apex curvature radius of ~50 nm (Fig. 1b) and an actual tip–substrate distance $d$ of ~7 Å, $d_{eff}$ is determined to be 9.7 Å (Supplementary Notes 14 and 15). Based on these estimations, the relative values of $\chi^{(2)}$ and $\chi^{(3)}$ were derived as $|\chi^{(3)}|^2/|\chi^{(2)}|^2 = (19.9 \pm 0.7)$ nm$^2$/V$^2$ and $2Re(\chi^{(3)}\chi^{(2)*})/|\chi^{(2)}|^2 = (-3.8 \pm 0.6)$ nm/V. From these relative values, we can also estimate the threshold field ($E_{DC,0}$), at which $\chi^{(2)}$ and $\chi^{(3)}E_{DC}$ effects equally contribute to the nonlinear optical generation ($|\chi^{(3)}E_{DC,0}| = |\chi^{(2)}|$), to be $2.2 \times 10^8$ V m$^{-1}$. Using the effective gap distance value of $d_{eff} = 9.7$ Å, the corresponding applied voltage is estimated to be ±0.21 V. Thus, when applied voltage exceeds this value, the contributions from $\chi^{(3)}E_{DC}$ become dominant over that from $\chi^{(2)}$.

It should be noted that although sub-angstrom (~0.5 Å) fluctuations in the actual distance $d$, which should induce variations in $E_{DC}$, are inevitably present under our ambient experimental conditions (Supplementary Fig. 11d), their influence was not incorporated into the fitting analysis. This is because our numerical analysis to examine the distance-sensitivity of EFISH effects revealed that sub-angstrom variations in the gap distance have only a minor influence on the measured electrophotonic modulation (Supplementary Fig. 24). Furthermore, the typical timescale of the fluctuations is on the order of microseconds and thus much shorter than the minute-scale signal integration time in our measurements. Consequently, the influence of the fluctuations in $E_{DC}$ would be effectively averaged out and would not manifest in the measured modulation curves. For these reasons, the fluctuations in $E_{DC}$ were not explicitly taken into account in our fitting analysis.

## Extension to tip-enhanced sum-frequency generation

Nonlinear optical processes generally involve multiphoton interaction and are often accompanied by drastic frequency conversion between the incoming and outgoing light. We recently demonstrated that such

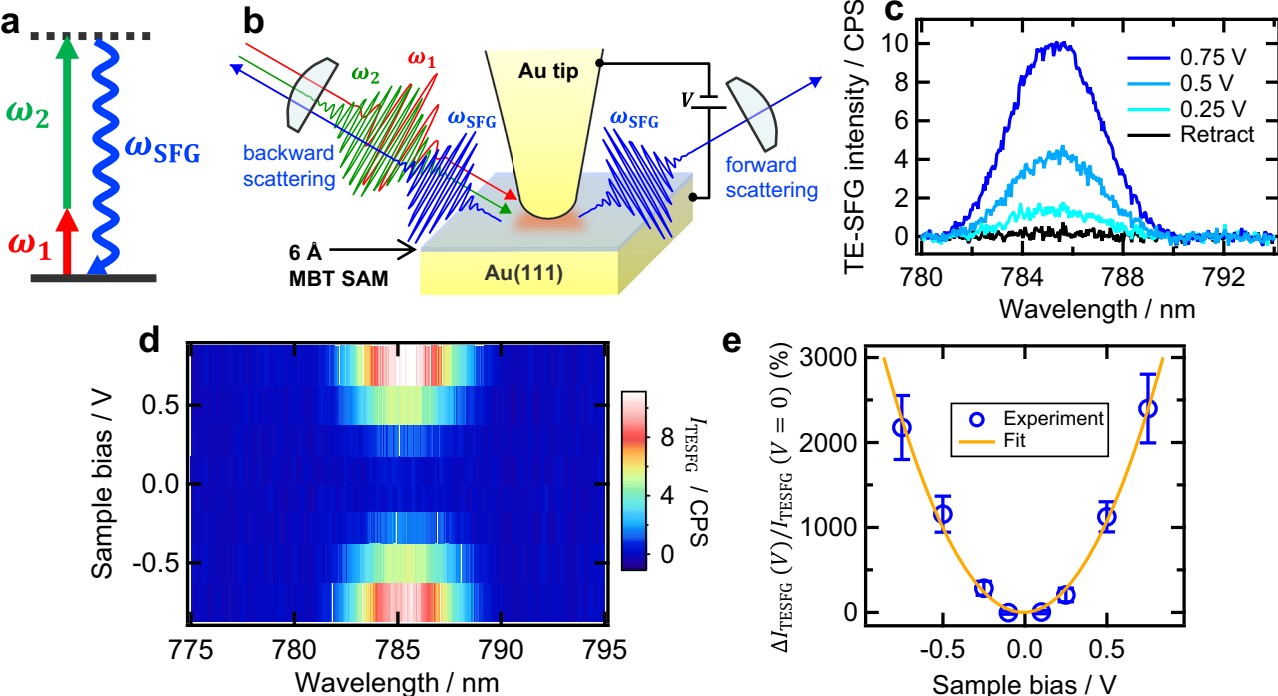

**Fig. 4 | Giant electric modulation of TE-SFG. a** Energy level diagram of the SFG process. Two-photon excitation with mid-IR ($\omega_1$) and near-IR ($\omega_2$) light induces radiation at the sum frequency of those light ($\omega_{SFG} = \omega_1 + \omega_2$). **b** Schematic depiction of the TE-SFG experiment using an Au tip and an Au substrate with an MBT SAM mounted on an STM unit. The experiments were conducted under ambient pressure in an air atmosphere at room temperature. **c** TE-SFG spectra obtained at sample biases of 0.25 V (cyan, $d \sim 6.7$ Å), 0.5 V (sky blue, $d \sim 7.6$ Å), and 0.75 V (dark blue, $d \sim 8.2$ Å) with a constant tunneling current setpoint of 250 pA. The black curve indicates the signal obtained when the tip and the substrate were retracted enough (30 nm) to deactivate the plasmonic enhancement effects. The excitation

intensity of mid-IR ($\omega_1$) and near-IR ($\omega_2$) was 20 and 0.5 mW, respectively. **d** Two-dimensional plot of TE-SFG spectra obtained with the various applied sample bias under the almost constant tip–substrate distance ($d \sim 7$ Å). **e** The voltage-dependent change in the TE-SFG output ($\Delta I_{TESFG}(V)$) normalized by the TE-SFG signal intensity at $V = 0$ V ($I_{TESFG}(V = 0)$) is depicted by blue open circles. Similar to the case of bias-dependent TE-SHG measurements (Fig. 3), the reference intensity $I_{TESFG}(V = 0)$ was approximated as the average TE-SFG intensities at $V = -0.1$ V and $V = 0.1$ V. The orange curve is the result of curve fitting with a quadratic function. The error bars represent the standard error of the mean determined from 10 independent repetitions of the same voltage-dependent measurement.

multicolored input and output light with separated frequencies can be simultaneously enhanced in a tip–substrate gap structure, thereby enabling the efficient plasmonic enhancement of nonlinear optical processes involving significant wavelength conversion[60] (Supplementary Note 13). The TE-SHG process discussed thus far exemplifies this phenomenon because it arises from the simultaneous enhancement of both near-IR excitation and visible emission[60]. While the giant nonlinear electrophotonic modulation has been demonstrated for this near-IR-to-visible TE-SHG process (Fig. 3), extending this modulation to other nonlinear optical processes involving even more pronounced wavelength conversion would further substantiate the broadband applicability of the observed nonlinear electrophotonic effects. In this context, we next focus on the voltage dependence of a tip-enhanced SFG (TE-SFG) process[61–63], another representative second-order nonlinear optical process characterized by a large frequency upconversion from the mid-IR input into near-IR or even visible output (Fig. 4a).

For the demonstration of voltage-dependent TE-SFG measurements in the angstrom-scale plasmonic junctions (Fig. 4b), we used spatially and temporally overlapped mid-IR ($\omega_1$: 3280 nm, 300 fs, FWHM: 60 nm, 20 mW) and near-IR ($\omega_2$: 1033 nm, 1 ps, FWHM: 1 nm, 0.5 mW) pulses as the excitation light with a repetition frequency of 50 MHz, both detuned from molecular resonance. Moreover, while the STM chamber was maintained at ultra-high vacuum conditions in the TE-SHG measurements, we exposed the chamber to the air atmosphere prior to the TE-SFG experiments in order to verify the operability of nonlinear electrophotonic modulation in open air under ambient conditions. Then, as shown in Fig. 4c, the vibrationally nonresonant TE-SFG signals were clearly observed in the visible region

around 785 nm. The observed spectral shape and band width predominantly reflect those of the incident mid-IR excitation pulse (Supplementary Fig. 3). Similar to the results of TE-SHG experiments in Fig. 2a, the TE-SFG intensity also increased with the sample bias voltage at constant current of 250 pA (Fig. 4c). While the tip–substrate distance varied from 6.7 to 8.2 Å in this experiment, the distance dependence of the electric field enhancement strength should be negligibly small due to the quantum plasmonic quenching effects[68,71,72,75–78]. Therefore, by analogy with the case of the bias-dependent TE-SHG (Fig. 2a), the observed TE-SFG intensity modulation shown in Fig. 4c can also be attributed to bias-induced effects.

Then, we investigated the impact of bias voltage while maintaining an almost constant tip–substrate distance ($d \sim 7$ Å). As shown in Fig. 4d, the intensity of the TE-SFG signals drastically increased with increasing the applied voltage in both positive (0.1 → 0.75 V) and negative (−0.1 → −0.75 V) directions. Consistent with the results of electrically controlled TE-SHG experiments (Fig. 3), the experimentally obtained $\Delta I_{TESFG}(V)$ (the blue open circles in Fig. 4e) exhibited a quadratic dependence on the applied voltage, with the value at $V \approx \pm 0.75$ V exceeding 2000% of the TE-SFG intensity at $V = 0$ V ($I_{TESFG}(V = 0)$). Similar bias dependences were also obtained using other Au tips (Supplementary Fig. 15), ensuring the reproducibility of our experiments. These findings demonstrate that the giant electric-field-induced modulation mechanism of plasmonic nonlinear optical effects is also highly effective for TE-SFG ($I_{TESFG} \propto |\chi^{(3)} E_{DC} + \chi^{(2)}|^2 I(\omega_1) I(\omega_2) \cong |\chi^{(3)} E_{DC}|^2 I(\omega_1) I(\omega_2)$) and operates over a broadband wavelength range, encompassing the mid-IR, near-IR, and visible regions. This broad-wavelength applicability is in

stark contrast to the previously reported electronic resonance-assisted EFISH study[90], where the range of excitation wavelength that can be used to detect the large EFISH effect was significantly restricted. Note that the slight difference in the minima of the TE-SHG (Fig. 3) and TE-SFG (Fig. 4e) curves may arise from minor variations in the atomic-scale shape of the Au tips and the resultant difference in the electrostatic field distribution within the gap.

Remarkably, the achievement of more than 2000% V$^{-1}$ modulation of TE-SFG signals in an angstrom-scale plasmonic gap (Fig. 4e) was realized under room temperature and ambient pressure conditions, rather than low temperature and ultra-high vacuum environments. Even under these ambient conditions, the stability of the signals under light irradiation (Supplementary Note 6) and the reproducibility of the results with different STM tips (Supplementary Fig. 15) were thoroughly verified. In addition, the possibility of the dielectric breakdown caused by an intense electrostatic field within the gap ($\sim$10$^9$ V m$^{-1}$) can also be reasonably ruled out in our ambient experimental conditions, because of the rare electron–gas collision within our angstrom-scale gap (Supplementary Note 9). This allows us to safely disregard any intense-field-induced modification of the tip or substrate, even under ambient experimental conditions. Furthermore, the near-field tip-enhancement scheme constructed under room temperature achieves markedly higher SHG and SFG conversion efficiencies than conventional far-field excitation, despite the much smaller number of molecules involved in the signal generation process in the near-field scheme (Supplementary Note 7). These findings highlight the promising potential of the observed nonlinear electoplasmonic phenomena and pave the way for the development of photonic devices that combine substantial modulation depth, operation under ambient conditions, and higher conversion efficiency.

Finally, it should be noted that our near-field-based giant electrical modulation depth on the order of 2000% V$^{-1}$ compares favorably with, and even exceeds, the recently reported record-high values (>1000% V$^{-1}$) for practical applications in the field of far-field-based electrophotonics[81–84,90]. In far-field schemes, the electric field across macroscopic electrodes is typically much weaker than that in near-field schemes. Such weaker electrostatic fields alone are generally insufficient to induce giant electro-optic interactions. Consequently, the record-high values in far-field approaches have been achieved primarily through some medium-design strategies. One of such approaches is to minimize the contribution of $\chi^{(2)}$ with respect to $\chi^{(3)}$ using centrosymmetric amorphous material films with $\sim$100 nm thickness[90]. Another important approach is to utilize field-induced phase transition or polarization switching phenomena in specific electroactive nonlinear optical media, such as transition metal dichalcogenides[81], superlattice structures[82,83], and ferroelectric sheets[84]. In contrast to these approaches that rely on specialized material designs, our near-field-based giant electrical modulation was achieved using a common SAM of electro-nonactive organic molecules that do not exhibit field-induced polarization switching and drastic phase transition. Furthermore, a similar extent of electric modulation ($\sim$1000% V$^{-1}$) was also observed even in the absence of the SAM (Supplementary Fig. 13). These results demonstrate that, in our near-field-based nonlinear optical modulation scheme, the choice of material is largely noncritical for achieving a modulation depth of at least 1000% V$^{-1}$. Therefore, the angstrom-scale plasmonic gaps have great potential to serve as an optimal platform for giant nonlinear optical modulation. A promising avenue for further exploration is the use of more electroactive nonlinear optical media, similar to those used in far-field-based approaches[81–84], to further improve modulation performance. Such advancements could achieve a practical level of modulation depth ($\sim$1000%) even under more modest bias voltages, paving the way for an atomic-scale electrophotonic information-processing technology.

## Discussion

By combining the STM and wavelength-tunable pulse laser system, we have demonstrated giant electric enhancement of plasmonic near-field SHG and SFG responses within an angstrom-scale plasmonic gap under room temperature, achieving a remarkable modulation depth of $\sim$2000% V$^{-1}$ not only at ultra-high vacuum conditions but also in open air under ambient conditions. The observed quadratic bias dependence and significant signal modulation are in stark contrast to the quasi-linear bias dependence and modest modulation depths ($\sim$10% V$^{-1}$) previously reported in nanoscale plasmonic gap systems[40–44]. Therefore, our findings lay the foundation for a paradigm shift from sub-100 nm to angstrom-scale electrophotonics, enabling ultrahigh-performance nonlinear optical modulation. We also identified several key advantages of angstrom-scale plasmonic structures: moderate driving voltage (<1 V), broad material compatibility without requiring specific electroactive media, operability under ambient conditions, and a broad operation wavelength range spanning the visible, near-IR, and mid-IR regions. The realization of electrophotonic systems that integrate all these properties into an ultra-compact platform far below the diffraction limit is a pivotal step toward the new era of atomic-scale electrophotonics. We believe that our results and concepts not only demonstrate the potential of near-field-based nonlinear electrophotonic modulation but also establish an experimental basis for integrating nanophotonics with atomic-scale electronics[91–93], driving further optimization of device architectures and advancements in angstrom-level photonic information-processing techniques.

## Methods

### Preparation of STM tip and sample

Gold tips for STM were prepared by electrochemical etching of gold wires with a diameter of 0.25 mm (Nilaco) in a 2.79 M KCl (FUJIFILM Wako Pure Chemical Corporation) aqueous solution[59]. After the etching process was completed, the prepared tips were washed with concentrated H$_2$SO$_4$ and ultrapure water and introduced into our STM chamber (USM1400, UNISOKU). Prior to the nonlinear optical measurements, we processed the Au tips with Ar$^+$ sputtering ($2 \times 10^{-3}$ Pa, 3 kV) for 3 h.

For the substrate to form the angstrom-scale gap structure, a 200 nm-thick gold thin film vapor-deposited on a mica substrate (UNISOKU) was employed. The substrate was first annealed in a butane flame to achieve an atomically flat Au(111) surface (Supplementary Fig. 1a) and cooled down to room temperature. Then, the Au substrate was immersed in 1 mM ethanolic solution of 4-Methylbenzenethiol (MBT) at room temperature for 48 h to form a $\sim$6 Å-thick self-assembled monolayer (SAM) of MBT on its surface (Supplementary Fig. 1b)[65]. Thereafter, the Au substrate was picked up from the solution, rinsed with pure ethanol, and dried with N$_2$ gas just before being placed into our STM chamber.

### Tip-enhanced nonlinear optical measurements

All the nonlinear optical measurements were performed at room temperature. Our optical system is based on an amplified Yb-fiber laser (1033 nm, 280 fs, 40 W, 50 MHz; Monaco-1035-40-40, Coherent). The fundamental output from the laser was divided into two portions by a beam splitter. The first portion (120 nJ) was used to drive a commercially available optical parametric oscillator (OPO, Levante IR, APE), generating two kinds of wavelength-tunable IR pulses: signal (1.3–2 μm) and idler (2.1–5 μm). The second portion was passed through an air-spaced Fabry–Pérot etalon to narrow down the spectral width. Three kinds of pulses (signal, idler, and narrow-band fundamental wave) were used as excitation sources of TE-SHG and TE-SFG experiments, as described below. See Supplementary Figs. 2 and 3 for the spectra of those pulses.

For TE-SHG experiments, the STM chamber was maintained at ultra-high vacuum conditions with a base pressure of $<1 \times 10^{-7}$ Pa. The

OPO's signal output centered at 1500 nm was used as an excitation source. The excitation light intensity was attenuated to 0.5 mW (pulse energy: 10 pJ, peak power density: $5 \times 10^6$ W/cm²) by a variable neutral density filter, and then focused onto the apex of the STM tip at an incident angle of 55° using a custom-made CaF₂ aspherical lens (NA = 0.34, focal length of 19.3 mm at 1033 nm) installed on the STM head. The incident light was linearly polarized along the axis of the tip (p-polarized). The backward-scattered TE-SHG signal at 750 nm was collimated with the same lens and separated from the incident light path using a custom-made dichroic mirror. The forward-scattered TE-SHG signal was collimated with another lens mounted in the reflection direction, and then the incident light was cut off by using a short-pass filter. Those two kinds of signals were separately input into a bundle fiber branched into two halves without polarization selection and directed into a spectrometer (Kymera 328i, Andor). Within the spectrometer, the signals from both directions were focused on the distinct vertical positions on the same electronically cooled CCD detector (iDus 416, Andor), allowing us to separately measure those two signals at the same time. The spectrometer wavelength was calibrated using an Hg/Ar lamp (SL2, StellarNet). When measuring voltage-dependent TE-SHG signals shown in Fig. 3, the measurements were repeated three times, with the order of the voltage sweeps varied for each measurement (Supplementary Note 6), and the signal accumulation time for each voltage point was 10 s.

For TE-SFG experiments, we extended this optical setup to a two-color excitation scheme, and the STM chamber was maintained at ambient pressure in an air atmosphere. To drive the TE-SFG process, the near-IR fundamental output from the Yb-fiber laser (0.5 mW, pulse energy: 10 pJ, peak power density: $5 \times 10^6$ W/cm²) and the mid-IR idler output (20 mW, pulse energy: 400 pJ, peak power density: $2 \times 10^6$ W/cm²) were spatially and temporally overlapped. The two laser pulses were combined collinearly at a dichroic mirror (reflective for 1033 nm and transparent for 1300–7000 nm) and then focused onto the apex of the STM tip by the same aspherical lens used in TE-SHG measurements. Both excitation pulses were linearly polarized along the axis of the tip (p-polarized). The detection of the signals was performed using the same method as that used for TE-SHG measurements. When measuring voltage-dependent TE-SFG signals shown in Fig. 4d and e, the applied voltages were varied in a random order, and the signal accumulation time for each voltage point was 300 s (Supplementary Note 6).

The relative modulations of TE-SHG/TE-SFG intensities plotted in Figs. 3 and 4e are defined as $\Delta I(V)/I(V=0)$, where $I(V=0)$ denotes the TE-SHG/TE-SFG intensity at $V = 0$ V, and $\Delta I(V) = I(V) - I(V=0)$ represents the bias-induced intensity change in the nonlinear optical signals. Under our STM-based experimental conditions, however, direct measurement of $I(V=0)$ is technically difficult. This is because a net tunneling current becomes zero at $V = 0$ V, preventing the operation of the feedback loop of STM to maintain an angstrom-scale tip–substrate gap distance. To circumvent this limitation, we approximated $I(V=0)$ as the average TE-SHG/TE-SFG intensities obtained at small positive and negative biases (±0.1 V): $I(V=0) \approx [I(V=+0.1\text{V})+I(V=-0.1\text{V})]/2$. This approximation is justified because the measured bias dependence of $I(V)$ is well described by a parabolic function, and the intensities at ±0.1 V are much smaller than those at ±1 V. Consequently, the variation of $I(V)$ within the ±0.1 V range is sufficiently small, and the averaged value not only provides a reasonable estimate of the zero-bias limit but also has almost no influence on the quantitative evaluation of the relative modulation depth $\Delta I(V)/I(V=0)$. Therefore, this averaged value was adopted as the reference intensity for quantitatively evaluating the relative modulation.

## Data availability
All the data generated in this study are provided in the Supplementary Information/Source Data file. Source data are provided with this paper.

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

## Acknowledgements

For technical assistance, we thank M. Aoyama, T. Kondo, N. Mizutani, T. Kikuchi, and T. Toyoda at the Equipment Development Center, Institute for Molecular Science (IMS), and E. Nakamura at the UVSOR synchrotron facility of IMS. SEM observation of tips was conducted at the Institute for Molecular Science, supported by "Advanced Research Infrastructure for Materials and Nanotechnology in Japan (ARIM)" of the Ministry of Education, Culture, Sports, Science and Technology (MEXT), Proposal Number JPMXP1225MS5011. T.S. acknowledges financial support from JSPS KAKENHI Grant-in-Aid for Scientific Research (A) (19H00865 and 22H00296) and for Transformative Research Areas (A) (24H02205); JST-PRESTO (JPMJPR1907); JST-CREST (JPMJCR22L2); the grant of OML Project by the National Institutes of Natural Sciences (NINS program No. OML032501); and Special Project by the Institute for Molecular Science (IMS program 25IMS1101). A.S. acknowledges financial support from JSPS KAKENHI Grant-in-Aid for Scientific Research (B) (23H01855) and for Challenging Research (Exploratory) (24K21759). S.T. acknowledges financial support from Grant-in-Aid for JSPS Fellows (22KJ3099).

## Author contributions

T.S. supervised the project. S.T., A.S., and T.S. designed the research; S.T., A.S., T.M., and T.S. built up the experimental setup; T.M. established the tip fabrication technique; S.T. performed the experiments with help from A.S. and T.M. and analyzed the data; S.T. and T.S. wrote the manuscript; and all authors discussed the results and commented on the manuscript.

## Competing interests

The authors declare no competing interests.
