## [Transparent Peer Review file · Nature Communications]

Giant near-field nonlinear electrophotonic effects in an angstrom-scale plasmonic junction

Corresponding Author: Professor Toshiki Sugimoto

Version 0:

Reviewer comments:

Reviewer #3

(Remarks to the Author)

I reviewed this manuscript before it was transferred to Nature Communications. While this work is along the lines of the previous work on EFISH by Mark Brongersma's, the demonstrated strong modulation would warrant publications in this journal. I am however still puzzled how the device operates at ambient conditions without electric discharges under such strong electric field in the junction. Unfortunately, the authors did not comment on this before they transferred the manuscript to Nature Communications.

Reviewer #4

(Remarks to the Author)

Dear Editor,

The manuscript by Takahashi et al., "Giant near-field nonlinear electrophotonic effects in an ångström-scale plasmonic junction," reports voltage-controlled enhancements of tip-enhanced SHG and SFG exceeding ~2000% within a 1 V bias across a ~6 Å Au–Au junction in an STM. By reducing the electrode spacing to the subnanometer scale, the authors generate quasi-static fields approaching 10^9 V/m and demonstrate that the EFISH contribution ($\chi(3) \cdot E_{DC}$) dominates the intrinsic $\chi(2)$ response—yielding quadratic bias dependence and unprecedented modulation depths. The experiments span UHV and ambient conditions, and cover mid-IR to visible upconversion, highlighting the platform's broadband potential.

The topic is timely and of broad interest to the nonlinear optics, nanophotonics, and scanning-probe communities. While the experiments are carefully executed, I have raised several technical questions and critiques that warrant clarification. In my view, the work has the potential to merit publication in Nature Communications once these points are adequately addressed. Below, I outline the key issues and recommended revisions for the authors' consideration.

1) Equation (1) forms the core of the authors' interpretation of the TE-SHG modulation, combining intrinsic second-order nonlinearities with the field-induced EFISH term. While the conceptual framework is sound, the fitting model appears to make several implicit assumptions that would benefit from clarification or further validation. Example:

> Gap Distance Sensitivity: Since $EDC = V/d$ and $EDC = V/d$, sub-ångström variations in the tip–sample distance (± 0.5 – 1 Å) could significantly impact the EFISH term. Could the authors please clarify whether this was accounted for in the fitting, or consider a brief sensitivity analysis?

> Field Enhancement Uniformity: The use of a scalar E_{gap} does not reflect the likely spatial nonuniformity in such confined junctions. Could the authors please clarify whether peak, average, or mode-volume field values were used would help? In summary, the current fitting model is a reasonable starting point, but I encourage the authors to address the above points either through clarifications, sensitivity analysis, or supplemental modeling to ensure the robustness and physical fidelity of their interpretation of the EFISH effect in such highly confined plasmonic junctions. As a suggestion a simple FEM/FDTD simulations showing how E_{gap} and EDC vary with d could strengthen the physical basis of the model and support the

claimed >1000%/V modulation.

2) This comment/suggestion is precisely the kind of control experiment needed to disentangle field-induced optical effects from tunneling-related carrier effects: Based on my understanding from their experimental procedure in STM, “constant current” mode means the system dynamically adjusts the tip height to maintain a fixed tunneling current (e.g., 500 pA) at a given bias. Importantly, this current is carried by tunneling electrons across the gap as stated by the authors. The authors use a constant-current STM configuration where the tip height is dynamically adjusted to maintain a fixed tunneling current (e.g., 500 pA) at a given bias. However, a high tunneling current inherently implies a significant injection of electrons across the plasmonic gap per second. This raises concerns that parasitic processes such as hot-carrier generation, inelastic scattering, impact ionization, photoluminescence, or electroluminescence may contribute to or amplify the observed second-order nonlinear signals (e.g., SHG or SFG).

These effects, especially in metal–molecule–metal junctions, could mimic or mask the optical nonlinearities the authors attribute to $\chi^{(2)}$ or $\chi^{(3)}$ processes. Therefore, I strongly encourage the authors to perform control measurements at lower setpoint currents (e.g., 10–50 pA) to test the robustness of their interpretation.

If the same bias-dependent modulation of TE-SHG intensity is observed at these reduced tunneling currents (I_1 , I_2 , I_3 ...), it would provide strong evidence that the effect is not driven by electron-induced mechanisms, but rather originates from a purely optical field-induced nonlinearity, such as EFISH ($\chi^{(3)} \cdot E_{DC}$), as claimed.

3) I am having also slight concern with the terminology: Definition of “Ångström-Scale” vs. SAM Thickness:

The tip–substrate distance is often referred to as “5–7 Å,” but includes a ~6 Å SAM layer (MBT). Therefore, the actual metal–metal distance may be larger. As a suggestion, I kindly recommend the clarification for the definition of “angstrom-scale gap” defined from metal-to-metal, tip apex to SAM, or through tunneling current equivalent?

4) Nonlocal and Spill-Out Effects: Nonlocal response and electron spill-out play important roles in plasmonic nanogaps, especially at sub-nanometer dimensions. In Section 9 of the Supplementary Information, the authors acknowledge the possibility of quantum quenching—such as tunneling and nonlocal screening—but argue that these effects are minimal due to a maintained tip–sample distance of ≥ 5 Å. While this is a plausible assumption, the claim remains qualitative. To strengthen their argument, it would be beneficial for the authors to include either:

- A reference to prior work that identifies the onset of quantum plasmonic suppression (e.g., electron tunneling or nonlocal damping) at angstrom-scale separations,
- Or, if possible, a simple simulation or estimate that demonstrates the transition from field-driven $\chi^{(3)}$ modulation to quantum-dominated behavior in their specific geometry.

Several key studies may be relevant here:

>Esteban R. et al. Bridging quantum and classical plasmonics with a quantum-corrected model. *Nat. Commun.* 3:825 doi: 10.1038/ncomms1806 (2012).

>Skjølstrup, E.J., Søndergaard, T. and Pedersen, T.G., 2018. Quantum spill-out in few-nanometer metal gaps: Effect on gap plasmons and reflectance from ultrasharp groove arrays. *Physical Review B*, 97(11), p.115429.

>Zhu, W., Esteban, R., Borisov, A. et al. Quantum mechanical effects in plasmonic structures with subnanometre gaps. *Nat Commun* 7, 11495 (2016). <https://doi.org/10.1038/ncomms11495>

5) I am also curious how the choice of crystalline Au influences the nonlinear modulation effects reported here. The authors utilize flame-annealed Au(111), which provides a flat, atomically ordered surface. However, what would they expect if an amorphous gold film or tip were used instead?

In particular, it would be helpful to know whether the electron scattering, plasmon damping, or local field enhancement in amorphous gold would significantly reduce the efficiency or stability of the observed TE-SHG and TE-SFG effects. Clarifying this would help evaluate how generalizable the approach is to other material systems and fabrication methods.

6) While the study offers compelling experimental evidence for giant near-field nonlinear modulation in an ångström-scale plasmonic junction, the theoretical underpinnings remain underdeveloped. I note that in Section 13 of the Supplementary Information, the authors introduce simulations to capture field enhancement and its spectral evolution. However, this section is quite limited in detail.

To enhance the clarity and reproducibility of this part of the work, the authors should explicitly state the underlying mechanism behind their numerical method used (FDTD), the dielectric functions employed for gold and MBT, and whether nonlocal or quantum-corrected models were considered. Given the sub-nanometer nature of the gap, such modeling choices are highly consequential.

More broadly, the manuscript would benefit significantly from deeper computational or analytical exploration of:

- The near-field distribution and its evolution under applied bias,
- The interaction between light and tunneling carriers in such confined geometries,
- The voltage-dependent transition from $\chi^{(2)}$ - to $\chi^{(3)}$ -dominated nonlinearities,
- And the gap-size dependence of TE-SHG and TE-SFG efficiencies.

In addition, the physical origin of the observed nonlinear peaks remains insufficiently explored. A decomposition of the nonlinear scattering response into electromagnetic modes—such as via quasi-normal mode analysis or pole fitting of the scattering matrix—could provide insights into whether dipolar, quadrupolar, or higher-order gap plasmons are involved. Incorporating these theoretical and mechanistic insights would not only support the authors’ experimental claims but also deepen the conceptual impact of this important work.

7) Additionally, I noticed that Fig. S15 in the current Supplementary Information appears closely related to Fig. 4 from the authors’ prior publication (Takahashi et al., *J. Phys. Chem. Lett.* 2023). However, it is unclear what modeling assumptions have changed between the two—particularly whether the current work accounts for the spacer layer (e.g., MBT) when defining the tip–substrate gap in the simulations. Could the authors clarify whether the SAM thickness was explicitly included in the field enhancement calculations presented in Fig. S15? It would also be helpful to briefly explain how the present modeling differs from the earlier work—both in physical assumptions and numerical parameters.

8) Page2, Lines35-39: Slightly, less critical but still important point is that regarding the very first part of introduction, the Authors stated that “Plasmonic functionalities are determined by the intrinsic optical properties of materials and the static

geometry of the plasmonic structures (e.g., size, shape, and surrounding environment), making it difficult to adjust the plasmonic properties after the geometry of the plasmonic nanostructure is fixed." While I understand the statement but it is misleading the potential readers from the point of when the geometry fixed it is hard to tune/control the optical properties. There are several studies counter actively showing the tunability of the intrinsic optical properties by changing the size and hybridizing the material content while keeping/maintaining the shape of the determined geometry.

Single material:

[1] Campos, A., Troc, N., Cottancin, E. et al. Plasmonic quantum size effects in silver nanoparticles are dominated by interfaces and local environments. *Nat. Phys.* 15, 275–280 (2019). <https://doi.org/10.1038/s41567-018-0345-z>

[2] Ringe, Emilie, et al. "Plasmon length: a universal parameter to describe size effects in gold nanoparticles." *The journal of physical chemistry letters* 3.11 (2012): 1479-1483.

[3] Henry, A.I., Bingham, J.M., Ringe, E., Marks, L.D., Schatz, G.C. and Van Duyne, R.P., 2011. Correlated structure and optical property studies of plasmonic nanoparticles. *The Journal of Physical Chemistry C*, 115(19), pp.9291-9305.

For the hybrid/colloidal formation:

[4] Rossner, Christian, Tobias AF König, and Andreas Fery. "Plasmonic properties of colloidal assemblies." *Advanced Optical Materials* 9.8 (2021): 2001869.

[5] Kilic, Ufuk, et al. "Broadband enhanced chirality with tunable response in hybrid plasmonic helical metamaterials." *Advanced Functional Materials* 31.20 (2021): 2010329.

[6] Kılıç, U., Mock, A., Feder, R., Sekora, D., Hilfiker, M., Korlacki, R., Schubert, E., Argyropoulos, C. and Schubert, M., 2019. Tunable plasmonic resonances in Si-Au slanted columnar heterostructure thin films. *Scientific reports*, 9(1), p.71.

I thereby kindly recommend that the author shines a light on the path of potential readers by showing the efforts being made and are indeed exist some studies available in the literature that some geometries can provide the opportunity still spectrally and amplitude-wise tailor the optical properties.

9) Lastly, the manuscript's novelty is enhanced by ambient-pressure operation. However, reviewer 3's concern about dielectric breakdown remains partially open. I think that authors kindly include the gas breakdown discussion in the main text or supplementary (it is currently only in their response letter), perhaps citing the limited gas volume ($<10^{-20}$ cm³) and impossibility of avalanche ionization as justification.

Reviewer #5

(Remarks to the Author)

The manuscript by S.Takahashi et al. presents a study on the tip-enhanced nonlinear optical processes on plasmonic thin films. They experimentally demonstrate the TE-SHG, TE-SFG and so on by utilizing an angstrom-scale plasmonic gap between a metallic tip and a flat metal substrate in a scanning tunneling microscope. They have found that such plasmonic gaps can experience strong near-field enhancement, which leads to a modulation factor up to 2000%/V in the EFISH process. Their results are detailed and well-organized, and may open new avenue for designing angstrom-level photonic devices. However, I still have several questions and comments about this manuscript before recommend it for publication.

Below are my specific comments:

1. The authors have measured the relative value of the second order and third order nonlinear susceptibilities. As a general question in the area of nonlinear optics, what is the SHG conversion efficiency? Is there a solution to improve this efficiency?
2. The authors have calculated the enhancement spectra of Au tip, but it seems that this resonance is not close to the fundamental wavelength. Is it possible to manipulate the plasmonic resonance of the Au tip to get a larger enhancement factor?
3. In the TE-SHG measurement, the SHG intensity first increases and then decreases while the gap becomes larger. I suggest the authors discuss more about this phenomenon. Where is the maximum intensity position? Is it related to the coupling effect or the interference of the plasmonic resonances?
4. P-polarized FWs are used in the measurement. Is it possible to measure the polarization states of the SHG and SFG signals?
5. Minor comments: a. the illustration of the backward/forward scattering schemes should be added; b. the font size in figures should be optimized.

Version 1:

Reviewer comments:

Reviewer #3

(Remarks to the Author)

The authors addressed reasonably my concerns.

Reviewer #4

(Remarks to the Author)

Dear Editor,

As I noted in my previous review preamble, the work submitted by by Takahashi et al. is timely and of broad interest to the nonlinear optics, nanophotonics, and scanning-probe communities. In my view, the work is merit to get published in *Nature Communications*. However I hve few more response t o the authors review and I believe that once the following points are also addressed, I can recommend for its publication in *Nature Communications*. Below, I further outline some points and suggestions below;

1) Plasmonic Tunability and Field Modulation: While the authors made substantial improvements in addressing physical modeling and parasitic effects, one point appears to remain unaddressed: the introductory statement implying that fixed-geometry plasmonic systems are untunable. Given the extensive literature on active modulation via hybridization and field-driven effects (including EFISH itself), I recommend revising the statement to better reflect the current state of tunability in plasmonic nanostructures.

2) Electrostatic Field Distribution: Clarification and Visual Support: The authors mention that the STM tip–substrate junction exhibits a non-uniform electrostatic field due to its curved apex geometry, in contrast to a parallel-plate capacitor. To enhance clarity and pedagogical value especially for readers outside the STM community, I recommend adding a simple schematic and FEM simulation illustrating this spatial field distribution as shown for their tip-substrate case. Such a visual comparison would strengthen the EFISH argument and help contextualize how Å-scale tip displacements can significantly modulate the local nonlinear response.

3) Clarification of Enhancement Factors and Reference Baselines: I have to be critical on the enhancement calculations or measurements. Throughout the manuscript, the authors cite various enhancement factors and field strengths (e.g., 2000%/V TE-SFG modulation, 10^8 – 10^9 V/m fields, and $>40\times$ enhancement in air). However, the methodology used to define and calculate these enhancement factors is unclear. Could the authors explicitly clarify:

What is the reference intensity or configuration used for computing each enhancement value (e.g., zero bias, retracted tip, or no-gap control)?

Is the reported 2000%/V slope extracted from normalized intensity fits (e.g., Fig. 4e)? If so, what was the normalization basis?

Have the authors performed any field-distribution simulations (e.g., FEM or FDTD) to validate that the field enhancement scales consistently with the tip–substrate separation and geometry?

Could they also provide a sensitivity analysis showing how ± 0.5 Å variations in gap size or voltage influence the enhancement trends?

Providing these clarifications would significantly strengthen the quantitative credibility of the reported modulation factors and ensure reproducibility by the broader community.

4) Follow-up to Comment (e) – Origin of Nonlinear Peaks: Thank you for outlining the general three-step process of near-field nonlinear emission. However, the claim that all processes are predominantly dipolar in nature would benefit from more direct or quantitative justification.

Given the Å-scale confinement and strong field gradients in the gap, higher-order modes (e.g., quadrupolar gap plasmons) could feasibly contribute to both the nonlinear polarization and radiated signal. I recommend the authors consider:

Providing simulation results (FEM or FDTD) showing the spatial and spectral distribution of field enhancements, or

A multipole decomposition or theoretical argument that supports dipolar dominance under their conditions, or

Empirical tests (e.g., polarization, angle, or tip-shape dependence) that would suppress or enhance higher-order modes.

This would significantly improve the mechanistic understanding of the TE-SHG/SFG signals and bolster the claim of dipolar origin.

Minor Comments:

>To further improve the TE-SHG and TE-SFG efficiencies, it is essential to tailor the enhancement spectrum such that the value of $|K_{\text{gap}} K_{\text{gap}} L_{\text{gap}}|^2$ terms in equations (S3) and (S4) are maximized.

It would be better state as follows $|K_{\text{gap}}(w_1) K_{\text{gap}}(w_2) L_{\text{gap}}|^2$.

>I also appreciate the authors' thoughtful response to my concerns regarding quantum plasmonic suppression and nonlocality. Their revised interpretation and supporting references now reflect the state-of-the-art understanding in this regime. However, I encourage the authors to make this clarification more visible in the main text, not just the Supplementary. Additionally, it would be valuable to indicate whether their modeling framework explicitly incorporates nonlocal corrections or if the cited works are used purely as qualitative support.

Reviewer #5

(Remarks to the Author)

My concerns have been addressed, and the manuscript has been obviously improved. I would like to recommend it for publication now.

Responses to reviewers' comments

The authors acknowledge the reviewers' fruitful comments and suggestions on our manuscript (manuscript number: NCOMMS-25-44074-T). Please find below a detailed response to each comment. The reviewers' comments and corresponding changes in the revised manuscript are highlighted in blue letters and yellow backgrounds, respectively.

Reviewer #3

Reviewer #3's comments

I reviewed this manuscript before it was transferred to Nature Communications. While this work is along the lines of the previous work on EFISH by Mark Brongersma's, the demonstrated strong modulation would warrant publications in this journal. I am however still puzzled how the device operates at ambient conditions without electric discharges under such strong electric field in the junction. Unfortunately, the authors did not comment on this before they transferred the manuscript to Nature Communications.

Reply to reviewer #3's comments

We greatly thank the reviewer for recognizing the significance of our development of angstrom-scale electrophotonics beyond the nanoscale regime and our demonstration of the giant electrophotonic effects within an angstrom-scale plasmonic gap. We also appreciate the reviewer's positive evaluation that our work would warrant publication in *Nature Communications*.

The reviewer still raises concern regarding the possibility of electric discharges under the strong electric field in our angstrom-scale junction. As replied in our previous response to the reviewer's comment 2, such discharges can be reasonably ruled out under our experimental conditions in terms of (i) an interelectrode electron transfer mechanism and (ii) the number of gas molecules within the gap:

(i) The angstrom-scale tip–substrate distance employed in our experiments ensures the overlap of electronic wavefunctions at the tip and substrate surfaces. Under these conditions, the interelectrode electron transfer occurs predominantly via the electron tunneling process, rather than ballistic field emission of electrons into the gas phase. This prevents electron–gas collisions that would otherwise trigger avalanche ionization of gas molecules, resulting in the absence of the electric discharges.

(ii) Given the number density of gas molecules under the ambient conditions and the volume of an angstrom-scale tip–substrate gap, at most only a few molecules can be present within the gap. Such a sparse molecular population cannot sustain the avalanche ionization processes of gas molecules, rendering electric discharge physically implausible even under ambient experimental conditions.

These considerations clearly rule out the possibility of electric discharges within our angstrom-scale
gap structure, and any discharge-induced modification of the tip or substrate can be safely disregarded.

In order to avoid a similar concern and promote the correct understanding for future readers,
we revised the main text and Supplementary Information to provide the discussion regarding the
absence of dielectric breakdown within the nanogap even under ambient conditions. Thank you for
your comment.

**Revisions**

Page 13, lines 353–357 in the main text: In addition, the possibility of the dielectric breakdown caused
by an intense electrostatic field within the gap ($\sim 10^9$ V/m) can also be reasonably ruled out in our
ambient experimental conditions, because of the extremely rare electron–gas collision within our
angstrom-scale gap (further details in Supplementary Section 9). This allows us to safely disregard any
intense-field-induced modification of the tip or substrate even under ambient experimental conditions.

Page 13, lines 310–329 in Supplementary Section 9: Notably, the absence of the structural
modification of the tip and substrate indicates that dielectric breakdown within the gap and associated
breakdown-induced damage to either the sample or the tip can be ruled out, even under the application
of a strong electrostatic field ($\sim 10^9$ V/m) across the gap. The impossibility of the dielectric breakdown
can be correctly interpreted by considering the probability of the electron–gas collision within our
angstrom-scale gap. Generally, the dielectric breakdown is induced by field emission of electrons from
an electrode into gas phase, avalanche ionization of gas molecules, and ballistic acceleration of charge
carriers within the interelectrode gap, leading to physical damage of the electrodes¹⁴. Thus, the
collision process between electrons and gas molecules plays a key role in the emergence of dielectric
breakdown. However, since the angstrom-scale tip–substrate distance employed in our experiments
ensures the overlap of electronic wavefunctions at the tip and substrate surfaces, the interelectrode
electron transfer occurs predominantly via the tunneling process, rather than ballistic field emission of
electrons into the gas phase. This prevents electron–gas collisions that would otherwise trigger
avalanche ionization of gas molecules, resulting in the absence of the dielectric breakdown. Moreover,
considering the number density of gas molecules under ambient conditions (2.69×10^{19} cm⁻³), the
extremely small volume of the angstrom-scale gap ($\sim 10^{-20}$ cm³) allows for the presence of at most a
few gas molecules within the gap. This is insufficient to sustain the cascade ionization processes,
making gas-phase electric breakdown physically impossible even under ambient experimental
conditions. Therefore, the possibility of electric discharges within our angstrom-scale gap structure
can be clearly ruled out, and any discharge-induced modification of the tip or substrate can be safely
disregarded.

Reviewer #4

**Reviewer #4's general comments**

The manuscript by Takahashi et al., "Giant near-field nonlinear electrophotonic effects in an ångström-
scale plasmonic junction," reports voltage-controlled enhancements of tip-enhanced SHG and SFG
exceeding ~2000% within a 1 V bias across a ~6 Å Au–Au junction in an STM. By reducing the
electrode spacing to the subnanometer scale, the authors generate quasi-static fields approaching 10^9
V/m and demonstrate that the EFISH contribution ($\chi^{(3)} \cdot E_{DC}$) dominates the intrinsic $\chi^{(2)}$
response—yielding quadratic bias dependence and unprecedented modulation depths. The experiments
span UHV and ambient conditions, and cover mid-IR to visible upconversion, highlighting the
platform's broadband potential.

The topic is timely and of broad interest to the nonlinear optics, nanophotonics, and scanning-
probe communities. While the experiments are carefully executed, I have raised several technical
questions and critiques that warrant clarification. In my view, the work has the potential to merit
publication in Nature Communications once these points are adequately addressed. Below, I outline
the key issues and recommended revisions for the authors' consideration.

**Reply to reviewer #4's general comments**

We sincerely thank the reviewer for your thorough and constructive evaluation of our cutting-edge
exploration of angstrom-scale nonlinear electrophotonics, extending beyond the conventional
nanoscale regime. We also greatly appreciate the reviewer's recognition of the timeliness and broad
relevance of our work to the nonlinear optics, nanophotonics, and scanning-probe communities, as
well as the reviewer's acknowledgment of our careful execution of the experiments. Note that
following the reviewer's comment, we re-examined the quantitative accuracy of the parameters
presented in this paper and found that the tip–substrate distance (d) employed in the bias-dependent
TE-SHG measurements (Fig. 3 in the main text) was ~7 Å rather than ~6 Å. While this correction leads
to minor changes in the estimated relative values of $\chi^{(2)}$ and $\chi^{(3)}$, it does not affect the main
conclusions or the overall concept of the paper.

In response to the reviewer's technical questions and critiques, we provide point-by-point
replies and associated revisions to our manuscript in the following. We believe that these revisions
will help us more clearly convey the reliability and impact of our achievements to broad readership of
*Nature Communications*.

**Reviewer #4's comment 1**

1) Equation (1) forms the core of the authors' interpretation of the TE-SHG modulation, combining
intrinsic second-order nonlinearities with the field-induced EFISH term. While the conceptual
framework is sound, the fitting model appears to make several implicit assumptions that would benefit
from clarification or further validation. Example:

> Gap Distance Sensitivity: Since $E_{DC} = V/d$, sub-ångström variations in the tip-sample distance
($\pm 0.5\text{--}1 \text{ \AA}$) could significantly impact the EFISH term. Could the authors please clarify whether this
was accounted for in the fitting, or consider a brief sensitivity analysis?
>Field Enhancement Uniformity: The use of a scalar E_{gap} does not reflect the likely spatial
nonuniformity in such confined junctions. Could the authors please clarify whether peak, average, or
mode-volume field values were used would help?
In summary, the current fitting model is a reasonable starting point, but I encourage the authors to
address the above points either through clarifications, sensitivity analysis, or supplemental modeling
to ensure the robustness and physical fidelity of their interpretation of the EFISH effect in such highly
confined plasmonic junctions. As a suggestion a simple FEM/FDTD simulations showing how E_{gap}
and EDC vary with d could strengthen the physical basis of the model and support the claimed
$>1000\%/V$ modulation.

**Reply to reviewer #4's comment 1**

We thank the reviewer for your valuable suggestions to clarify and validate the implicit assumptions
underlying our fitting model based on equation (1) in the main text. In order to ensure the robustness
and physical fidelity of our interpretation of the experimental result, the reviewer recommended that
we address (a) **Gap Distance Sensitivity** and (b) **Field Enhancement Uniformity**. In the following,
we provide point-by-point responses to these two issues.

**(a) Gap Distance Sensitivity**

As shown in Supplementary Fig. S11, the clear STM images were obtained under illumination of
excitation pulses for TE-SHG, indicating the absence of significant tip-substrate distance variations
that could disturb the STM topography during the TE-SHG measurements. While sub-angstrom
fluctuations of $\sim 0.5\text{--}1 \text{ \AA}$ cannot be entirely excluded, their typical timescale is on the order of
microsecond scale and much shorter than minute-scale signal integration time in our measurements.
Consequently, the influence of the fluctuations in the static electric field strength (E_{DC}) would be
effectively averaged out and would not manifest in the measured modulation curves. For this reason,
the fluctuations in E_{DC} were not explicitly taken into account in our fitting analysis.

In accordance with your comment, we added the above discussion to the main text to more
clearly convey that sub-angstrom variations in the tip-substrate distance can be disregarded in our
experimental conditions. Thank you for your comment.

**(b) Field Enhancement Uniformity**

As the reviewer pointed out, the enhanced electric field in the angstrom-scale gap (E_{gap}) is spatially
nonuniform and extends over the nanoscale region with a radius of $\sim 20 \text{ nm}$ (Fig. I). Notably, we also
performed theoretical analysis of the electrostatic field distribution within the gap based on the image

charge method [*Rev. Bras. Ensino Fís.* **31**, 3503 (2009)] and revealed that a comparable degree of
 spatial nonuniformity is also present in the electrostatic field across the gap (E_{DC} , Fig. II). In contrast
 to the case of a parallel-plate capacitor, where the electrostatic field strength between electrodes is
 spatially uniform, the tip–substrate junction exhibits a position-dependent field distribution due to the
 curved geometry of the tip apex. Consequently, both E_{gap} and E_{DC} exhibit spatial variations, giving
 rise to position-dependent nonlinear polarizations within the nanoscale region beneath the tip apex.
 Importantly, our recent experiments revealed that near-field second-order nonlinear optical signals
 originate from a nanoscale region of a few tens of nanometers directly beneath the tip apex
 [arXiv:2509.09179 [physics.optics]], which is quantitatively consistent with theoretically predicted
 ~ 20 -nm-scale lateral field enhancement areas (Figs. I and II). Therefore, the values of E_{gap} and E_{DC}
 in our analysis represent the effective field strengths averaged over this laterally extended ~ 20 nm
 region confined within the tip–substrate gap.

In accordance with the reviewer’s comment, we have added Figs. I and II to Supporting
 Information, together with a detailed characterization of these spatial distributions. Furthermore, we
 revised the main text to clarify that E_{gap} and E_{DC} in equation (1) represent the spatially averaged
 field values over ~ 20 -nm field enhancement region within the angstrom-scale tip–substrate gap. We
 thank the reviewer for your helpful comment.

 **Fig. I | Spatial distributions of the field enhancement factor K_{gap} .** The theoretical predictions of
 (a, b) surface-normal and (c, d) surface-parallel electric field components calculated through the

FDTD method are displayed. The model employs a rounded gold cone with a 30° opening angle, 50-
 170 nm apex radius, and 15000-nm length. The tip–substrate distance was set to be 2 nm. Panels (a) and
 171 (c) show the field in the XZ plane ($Y = 0$ nm), while panels (b) and (d) show the field in the XY plane
 ($Z = 0.5$ nm). The wavelength of incident light was 3280 nm. The coordinates $(X, Y) = (0$ nm, 0 nm)
 represent the position of minimum tip–substrate distance. (e) One dimensional spatial profiles of K_{gap}
 along the X-axis ($Y = 0$ nm and $Z = 0.6$ nm). Green and red curves represent the surface-normal and
 surface-parallel field components, respectively.

**Fig. II | Simulated spatial distributions of the electrostatic field E_{DC} under 1 V of voltage**
 **application across the tip-substrate nanogap.** The calculation was performed through the image
 charge method [*Rev. Bras. Ensino Fís.* **31**, 3503 (2009)]. The surface-normal (a) and surface-parallel
 (b) electrostatic field components are plotted in the XZ plane ($Y = 0$ nm). In the calculation, the tip was
 approximated by a nanosphere with a 50-nm apex radius, and the tip–substrate distance was set to be
 1 nm. The coordinates $(X, Y) = (0$ nm, 0 nm) represent the position of minimum tip–substrate distance.

**Revisions for (a)**

Page 10, lines 278–284 in the main text: It should be noted that although sub-angstrom (~ 0.5 – 1 Å)
 fluctuations in d , which should induce variations in E_{DC} , are inevitably present under our ambient
 experimental conditions (Supplementary Fig. S11d), their influence was not incorporated into the
 fitting analysis. This is because the typical timescale of the fluctuations is on the order of microsecond
 and thus much shorter than minute-scale signal integration time in our measurements. Consequently,
 the influence of the fluctuations in E_{DC} would be effectively averaged out and would not manifest in
 the measured modulation curves. For this reason, the fluctuations in E_{DC} were not explicitly taken
 into account in our fitting analysis.

**Revisions for (b)**

Page 9, lines 241–243 in the main text: Moreover, E_{gap} and E_{DC} in eq. (1) represent the spatially
 averaged field values over the ~ 20 -nm field enhancement region beneath the tip apex (see
 Supplementary Section 14 for details).

Page 26, lines 660–674 in Supplementary Section 14: In this section, we quantitatively estimate the
 spatial distribution of the electric field within the nanogap. Figure S19 displays the surface-normal and
 surface-parallel components of the electric field within the tip–substrate gap (E_{gap}), which were
 calculated through the FDTD simulation for a tip with a curvature radius of 50 nm. This tip radius is

comparable to the size of the tip apex used in our experiments (Fig. 1b in the main text). Note that the
tip–substrate gap was assumed to be vacuum, and the presence of the SAM layer was not incorporated.
As shown in Fig. S19, the enhanced electric field is spatially non-uniform and distributed over the
nanoscale region with a radius of ~20 nm. Furthermore, our theoretical analysis of the electrostatic
field distribution within the gap using the image charge method⁵⁴ revealed that the electrostatic field
across the gap (E_{DC}) also exhibits a similar spatial distribution (Fig. S20). Importantly, our recent
experiments revealed that near-field second-order nonlinear optical signals originate from a nanoscale
region with an area of a few tens of nanometers directly beneath the tip apex¹⁰. Since this result is
consistent with theoretically predicted ~20-nm-scale lateral field enhancement areas (Figs. S19 and
S20), the effective field strengths averaged over this ~20 nm region mainly contributes to the
generation of near-field second-order nonlinear optical signals.

Reviewer #4's comments 2 and 4

2) This comment/suggestion is precisely the kind of control experiment needed to disentangle field-
induced optical effects from tunneling-related carrier effects: Based on my understanding from their
experimental procedure in STM, “constant current” mode means the system dynamically adjusts the
tip height to maintain a fixed tunneling current (e.g., 500 pA) at a given bias. Importantly, this current
is carried by tunneling electrons across the gap as stated by the authors. The authors use a constant-
current STM configuration where the tip height is dynamically adjusted to maintain a fixed tunneling
current (e.g., 500 pA) at a given bias. However, a high tunneling current inherently implies a significant
injection of electrons across the plasmonic gap per second. This raises concerns that parasitic processes
such as hot-carrier generation, inelastic scattering, impact ionization, photoluminescence, or
electroluminescence may contribute to or amplify the observed second-order nonlinear signals (e.g.,
SHG or SFG).

These effects, especially in metal–molecule–metal junctions, could mimic or mask the optical
nonlinearities the authors attribute to $\chi^{(2)}$ or $\chi^{(3)}$ processes. Therefore, I strongly encourage the
authors to perform control measurements at lower setpoint currents (e.g., 10–50 pA) to test the
robustness of their interpretation.

If the same bias-dependent modulation of TE-SHG intensity is observed at these reduced tunneling
currents (I_1, I_2, I_3, \dots), it would provide strong evidence that the effect is not driven by electron-induced
mechanisms, but rather originates from a purely optical field-induced nonlinearity, such as EFISH
($\chi^{(3)} \cdot E_{DC}$), as claimed.

4) Nonlocal and Spill-Out Effects: Nonlocal response and electron spill-out play important roles in
plasmonic nanogaps, especially at sub-nanometer dimensions. In Section 9 of the Supplementary
Information, the authors acknowledge the possibility of quantum quenching—such as tunneling and
nonlocal screening—but argue that these effects are minimal due to a maintained tip–sample distance

of ≥ 5 Å. While this is a plausible assumption, the claim remains qualitative. To strengthen their
argument, it would be beneficial for the authors to include either:

- · A reference to prior work that identifies the onset of quantum plasmonic suppression (e.g., electron
tunneling or nonlocal damping) at angstrom-scale separations,
- · Or, if possible, a simple simulation or estimate that demonstrates the transition from field-driven
$\chi^{(3)}$ modulation to quantum-dominated behavior in their specific geometry.

Several key studies may be relevant here:

>Esteban R. et al. Bridging quantum and classical plasmonics with a quantum-corrected model. *Nat.*
*Commun.* 3:825 doi: 10.1038/ncomms1806 (2012).

>Skjølstrup, E.J., Søndergaard, T. and Pedersen, T.G., 2018. Quantum spill-out in few-nanometer
metal gaps: Effect on gap plasmons and reflectance from ultrasharp groove arrays. *Physical Review*
*B*, 97(11), p.115429.

>Zhu, W., Esteban, R., Borisov, A. et al. Quantum mechanical effects in plasmonic structures with
subnanometre gaps. *Nat Commun* 7, 11495 (2016). <https://doi.org/10.1038/ncomms11495>

**Reply to reviewer #4's comments 2 and 4**

As the reviewer correctly pointed out, “constant current” mode refers to an STM setting in which the
tip height is dynamically adjusted to maintain a fixed tunneling current at a given bias. As discussed
in the main text and in Supplementary Section 11, the influence of the tunneling current in our
experiments was clearly ruled out through our control experiments performed under a constant
tunneling current setpoint.

In the bias-dependent TE-SHG measurements presented in Fig. 3 in the main text, we varied
not only the applied bias but also the tunneling current setpoint in the range of 40–1750 pA, allowing
264 us to prevent the variation in the tip-substrate distance and maintain the field enhancement strength
(see Table S1 for the explicit setpoint values). While this experimental scheme may seemingly leave
the possibility of tunneling current contributions in the observed clear quadratic bias dependence and
$\sim 2000\%/V$ modulation of TE-SHG intensity, our control bias-dependent TE-SHG measurements
performed under a “fixed tunneling-current setpoint (500 pA)” also clearly revealed a quadratic
modulation of comparable magnitude (Supplementary Fig. S14b). The consistency of the two results
(Figs. 3 and S14) indicates that giant near-field nonlinear electrophotonic effects occur regardless of
the tunneling current variations, demonstrating that the contributions of tunneling-current-driven
processes, such as hot-carrier generation, inelastic scattering, impact ionization, photoluminescence,
or electroluminescence, are negligibly small.

Notably, in this control experiment under the constant current mode (Fig. S14b), increasing the
applied bias from 0.1 V to 1 V led to an elongation of the tip–substrate distance from ~ 4 Å to ~ 6 Å
(Fig. S14a). Based on conventional classical electrodynamic simulations, both the near-field
enhancement factor (K_{gap}) and the radiation efficiency (L_{gap}) are expected to decrease by several tens
of percents during this tip–substrate distance elongation [*J. Phys. Chem. B* **110**, 6692 (2006);

*Nanotechnology* **22**, 075204 (2011); *Faraday Discuss.* **178**, 151 (2015); *Nat. Commun.* **11**, 1021
(2020)]. Accordingly, the absolute TE-SHG intensity, proportional to $|K_{\text{gap}}|^4 |L_{\text{gap}}|^2$ (equation (S3)
in Supplementary Information), should decrease by nearly one order of magnitude during this
elongation. Nevertheless, the TE-SHG intensity markedly increased with the applied bias, and the
pronounced quadratic bias dependence with a modulation depth of $\sim 2000\%/V$ was still observed under
the constant current mode (Fig. S14b). This result indicates that the expected distance-dependent
variations in the field enhancement strength predicted by classical electromagnetic theory are
essentially suppressed. This deviation from classical behavior can be attributed to quantum plasmonic
quenching effects: at the gap distances of < 1 nm, the influences of quantum mechanical phenomena,
such as electron spill-out from the metal surface and the overlap of electronic wavefunctions across
the gap, begin to play roles and suppress the classically predicted field enhancement [*Nat. Commun.*
**3**, 825 (2012); *Nano. Lett.* **12**, 1333 (2012)]. Particularly, in the gap distance range of $\sim 4\text{--}7$ Å, these
quantum suppression effects and the classically expected enhancement nearly cancel each other,
making the overall field enhancement effectively independent of the tip–substrate distance (Fig. III)
[*Nano. Lett.* **12**, 1333 (2012)]. Note that the peak power densities of the near-IR (1500 nm and 1033
294 nm) and mid-IR (3280 nm) excitation pulses in our experiments are estimated to be 5×10^6 W/cm²
and 2×10^6 W/cm², respectively, which are involved within the intensity range investigated in this
calculation (Fig. III) [*Nano. Lett.* **12**, 1333 (2012)]. We thus consider that this theoretical result (Fig.
III) serves as a useful reference for estimating the possible contributions of quantum effects involved
in our present experimental conditions.

Therefore, in our fixed-current experimental conditions shown in Fig. S14, not only current-
induced effects but also gap-distance-dependent variations in field enhancement can be reasonably
excluded. This allows us to clearly attribute the observed TE-SHG modulation to the voltage-induced
nonlinearity represented by the $\chi^{(3)}E_{\text{DC}}$ term.

[REDACTED]

**Fig. III | Influences of quantum plasmonic quenching in subnanometer gaps** [*Nano. Lett.* **12**, 133
3 305 (2012)]. [REDACTED]

In our previous reply and earlier manuscript version, we assumed that quantum plasmonic
quenching effects are minimal under at the tip–substrate distance of $\geq 5 \text{ \AA}$. However, your insightful
comments led us to carefully reconsider this assumption. We now recognize that in the 4–7 Å range
employed in our experiments (Figs. 2–4 in the main text and Supplementary Fig. S14), quantum
plasmonic quenching effects play a significant role in stabilizing the field enhancement strength,
effectively making it almost insensitive to variations in the gap distance (4–7 Å). We have therefore
revised our manuscript to more accurately convey this understanding and added citations to the
following seven key references that identify the onset of quantum plasmonic suppression in
subnanometer gaps. We sincerely appreciate your comments.

[a] Zuloaga, J., Prodan, E. & Nordlander, P. Quantum Description of the Plasmon Resonances of
a Nanoparticle Dimer. *Nano Lett.* **9**, 887–891 (2009).

[b] Savage, K. J. *et al.* Revealing the quantum regime in tunnelling plasmonics. *Nature* **491**, 574–
577 (2012).

[c] Marinica, D. C., Kazansky, A. K., Nordlander, P., Aizpurua, J. & Borisov, A. G. Quantum
Plasmonics: Nonlinear Effects in the Field Enhancement of a Plasmonic Nanoparticle Dimer.
*Nano Lett.* **12**, 1333–1339 (2012).

[d] Esteban, R., Borisov, A. G., Nordlander, P. & Aizpurua, J. Bridging quantum and classical
plasmonics with a quantum-corrected model. *Nat. Commun.* **3**, 825 (2012).

[e] Zhu, W. & Crozier, K. B. Quantum mechanical limit to plasmonic enhancement as observed
by surface-enhanced Raman scattering. *Nat. Commun.* **5**, 5228 (2014).

[f] Zhu, W. *et al.* Quantum mechanical effects in plasmonic structures with subnanometre gaps.
*Nat. Commun.* **7**, 11495 (2016).

[g] Liu, S. *et al.* Inelastic Light Scattering in the Vicinity of a Single-Atom Quantum Point Contact
in a Plasmonic Picocavity. *ACS Nano* **17**, 10172–10180 (2023).

**Revisions**

Page 6, lines 158–173 in the main text: This voltage increase at a constant tunneling current (500 pA)
extended the tip–sample distance from $\sim 5 \text{ \AA}$ to $\sim 7 \text{ \AA}$ (Fig. 2b). Based on conventional classical
electrodynamic simulations, the near-field enhancement strength is expected to decrease by several
tens of percents during this $\sim 2\text{-}\text{Å}$ tip–substrate distance elongation^{54–59}. However, at the gap distances
of $< 1 \text{ nm}$, the influences of quantum mechanical phenomena, such as electron spill-out from the metal
surface and the overlap of electronic wavefunctions across the gap, begin to quench plasmon
excitations and suppress the increase in field enhancement^{53,56,57,60–63}. Particularly, in the gap distance
range of $\sim 4\text{--}7 \text{ \AA}$, these quantum suppression effects and the classically expected enhancement nearly

cancel each other, making the overall field enhancement effectively independent of the tip–substrate
distance^{53,56,57,60–63}. Therefore, the near-field enhancement strength should remain essentially
unaffected during the distance elongation from 5 Å to 7 Å. Despite such constant field enhancement,
the TE-SHG intensity significantly increased with the applied voltage (Fig. 2a). Importantly, this
distance expansion occurred when the bias voltage was raised from 0.1 V to 0.75 V at a constant
tunneling current (500 pA). With this voltage increase, the electrostatic field below the tip apex
increased from 2×10^8 V/m to 1×10^9 V/m. Therefore, both the TE-SHG intensity and the electrostatic
field increased concurrently, indicating the presence of substantial electrostatic field-induced effects
that boost the overall TE-SHG intensity.

Page 7, lines 196–198 in the main text: Therefore, we can reasonably conclude that the observed
electrical modulation behavior is predominantly governed by voltage-induced electrophotonic effects,
distinct from variations in the near-field enhancement or contributions from the tunneling current.

Page 11, lines 311–313 in the main text: While the tip–substrate distance varied from 6.7 Å to 8.2 Å
in this experiment, the distance dependence of the electric field enhancement strength should be
negligibly small due to the quantum plasmonic quenching effects^{53,56,57,60–63}.

Page 15, lines 365–369 in Supplementary Section 10: Moreover, in the case of bare Au (111) substrate,
the electron wavefunctions spilling out from the tip apex and substrate surface are expected to more
largely overlap compared with the SAM-covered configuration. Such overlap would enhance quantum
plasmonic quenching effects^{15–21}, leading to a weaker near-field enhancement and consequently a
smaller TE-SHG signal for the bare Au(111) substrate compared with the SAM-covered surface.

Pages 17–18, lines 412–452 in Supplementary Section 11: In the bias-dependent TE-SHG
measurements presented in Fig. 3 in the main text, we varied not only the applied bias but also the
tunneling current setpoint in the range of 40–1750 pA, allowing us to prevent the variation in the tip–
substrate distance and maintain the field enhancement strength (see Table S1 for the explicit setpoint
values). To check whether this current tuning influenced the observed clear quadratic bias dependence
and $\sim 2000\%/V$ modulation of TE-SHG intensity, we performed control bias-dependent TE-SHG
measurements under a constant tunneling current setpoint (Fig. S14). As shown in Fig. S14b, even in
this constant-current condition, the TE-SHG intensity exhibited quadratic bias dependence, and the
large modulation depth ($\sim 2000\%/V$) was still observed. These results indicate that the giant near-field
nonlinear electrophotonic effects occur regardless of the tunneling current variations, demonstrating
that the contributions of tunneling-current–driven processes, such as hot-carrier generation, inelastic
scattering, impact ionization, photoluminescence, or electroluminescence, are negligibly small.

It should be noted that in this control experiment under the constant current mode, increasing
the applied bias from 0.1 V to 1 V led to an elongation of the tip–substrate distance from ~ 4 Å to ~ 6

Å (Fig. S14a). Based on the previous classical electrodynamic simulations, it is expected that the near-
field enhancement strength significantly decreases during this tip–substrate distance elongation^{16,20,24–}
²⁷. Nevertheless, the TE-SHG intensity markedly increased with the applied bias, and the pronounced
quadratic bias dependence with a modulation depth of ~2000%/V was still observed under the constant
current mode (Fig. S14b). This result indicates that the expected distance-dependent variations in the
field enhancement strength predicted by classical electromagnetic theory are essentially suppressed.
This deviation from classical behavior can be attributed to quantum plasmonic quenching effects: at
the gap distances of < 1 nm, the influences of quantum mechanical phenomena, such as electron spill-
out from the metal surface and the overlap of electronic wavefunctions across the gap, begin to play
roles and suppress the classically predicted field enhancement^{15–21}. Particularly, in the gap distance
range of 4–7 Å, these quantum suppression effects and the classically expected enhancement nearly
cancel each other, making the overall field enhancement effectively independent of the tip–substrate
distance^{15–21}. Therefore, in the constant-current experiment shown in Fig. S14, not only current-
induced effects but also distance-dependent variations in field enhancement strength can be reasonably
excluded. This allows us to clearly attribute the observed TE-SHG modulation to the voltage-induced
nonlinearity represented by the $\chi^{(3)}E_{DC}$ term.

Notably, further reduction of the tip–substrate distance below ~4 Å leads to a regime where
quantum plasmonic quenching becomes dominant, resulting in a steep decrease in the electric field
enhancement factors^{15–21}. In contrast, the 4–7 Å regime represents a critical crossover region just
before this steep decline, where the electric field enhancement is maximized^{15–21}. Due to this
compensation, the overall field enhancement factor remains nearly constant at its maximum value
throughout this gap-size range, rendering it effectively independent of tip–substrate distance from 4 Å
to 7 Å. Thus, the 4–7 Å distance range not only offers optimal field enhancement conditions for both
TE-SHG and TE-SFG processes, but also provides an ideal regime for exclusively probing the intrinsic
bias dependence of the giant electrophotonic response, distinct from variations in the near-field
enhancement.

**Reviewer #4's comment 3**

3) I am having also slight concern with the terminology: Definition of “Ångström-Scale” vs. SAM
Thickness: The tip–substrate distance is often referred to as “5–7 Å,” but includes a ~6 Å SAM layer
(MBT). Therefore, the actual metal–metal distance may be larger. As a suggestion, I kindly
recommend the clarification for the definition of “angstrom-scale gap” defined from metal-to-metal,
tip apex to SAM, or through tunneling current equivalent?

**Reply to reviewer #4's comment 3**

Thank you for your important comment regarding the definition of the term “angstrom-scale gap” and
“tip–substrate distance” in our manuscript. We would like to clarify that throughout the main text,
these terms refer specifically to the metal–to–metal separation between the tip apex and the substrate

surface, rather than to the distance from the tip apex to the outer end of the SAM layer. As explained
in Supplementary Section 3, such metal-to-metal distances were measured by identifying the contact
point of the tip apex and the ~ 6 -Å-thick SAM layer through the measurement of the $I_t - d$ curve.

When we refer to a tip-substrate distance of ~ 5 Å (the light blue curve in Fig. 2a in the main
text), we assume that the tip apex slightly indents into the SAM layer, and thus the tip-to-substrate
separation is shorter than the thickness of SAM. Although such direct contact conditions may give rise
to additional signal enhancement arising from chemical bonding formation between the tip and sample
[*Angew. Chem. Int. Ed.* **62**, e202218799 (2023)], the terminal methyl group of MBT molecules is
chemically inert, and its interactions with the Au tip are too weak to form chemical bonds. Therefore,
such additional chemical enhancement should be negligibly small in our measurement conditions.
Indeed, even when the metal-to-metal distance reached ~ 5 Å where the tip-molecule contact was
expected, we did not observe any significant increase in signal intensity (the light blue curve in Fig.
2a in the main text). Therefore, the contribution of contact-induced chemical enhancement can be
disregarded in our observation. We have revised the manuscript to explicitly state this point, as well
as the definition of the term “angstrom-scale gap” and “tip-substrate distance” to avoid ambiguities
for future readers. Thank you again for your valuable comment.

**Revisions**

Page 5, lines 121–123 in the main text: The tip-substrate distance d is defined as the metal-to-metal
separation between the tip apex and the substrate surface.

Page 5, lines 130–131 in the main text: Then, the tip-substrate distance (d), defined as the metal-to-
metal separation between the tip apex and the substrate surface, was reduced from ~ 30 nm to ~ 5 Å
under the sample bias of 0.1 V.

Page 5, lines 136–139 in the main text: Moreover, while the tip apex is expected to slightly indent into
the SAM layer at a tip-substrate distance of ~ 5 Å, the potential influence of such contact, including
chemical enhancement effects⁵², should be negligibly small because the terminal methyl group of MBT
molecules is chemically inert (further details in Supplementary Section 3).

Page 4, lines 67–68 in Supplementary Section 3: The absolute gap distance between the tip apex and
metal surface (d) is an important experimental parameter for tip-enhanced nonlinear optical
measurements.

Page 4, lines 86–94 in Supplementary Section 3: It should be noted that such direct contact conditions
can sometimes give rise to additional signal enhancement effects arising from chemical bonding
formation between the tip and sample⁷. However, since the terminal methyl group of MBT molecules
is chemically inert, and its interactions with the tip are too weak to form chemical bonds, such

additional enhancement should be negligibly small in our measurement conditions. Indeed, even when
the metal-to-metal distance reached $\sim 5 \text{ \AA}$ where the tip-molecule contact was expected, we did not
observe any significant increase in signal intensity (the light blue curve in Fig. 2a in the main text).
Therefore, the contribution of contact-induced chemical enhancement can be disregarded in this study.

Page 6, lines 127–129 in Supplementary Section 4: In these experiments, it was critical to minimize
the voltage-dependent changes in the gap distance between the tip apex and metal surface (d) to reduce
the variation of field enhancement strength and isolate purely voltage-induced effects.

Reviewer #4's comment 5

5) I am also curious how the choice of crystalline Au influences the nonlinear modulation effects
reported here. The authors utilize flame-annealed Au(111), which provides a flat, atomically ordered
surface. However, what would they expect if an amorphous gold film or tip were used instead?

In particular, it would be helpful to know whether the electron scattering, plasmon damping,
or local field enhancement in amorphous gold would significantly reduce the efficiency or stability of
the observed TE-SHG and TE-SFG effects. Clarifying this would help evaluate how generalizable the
approach is to other material systems and fabrication methods.

Reply to reviewer #4's comment 5

Although we have not directly performed experiments using amorphous gold films or tips, we expect
that the reduced atomic ordering and increased density of local defects in amorphous gold would lead
to stronger electron scattering, enhanced plasmon damping, and thus lower the efficiency and stability
of the field enhancement effect compared to flame-annealed Au(111). However, we consider that the
voltage-induced giant quadratic modulation of TE-SHG and TE-SFG intensities reported here should
be observable even for amorphous gold. This is because these field-induced effects arise from an
intrinsic near-field electrophotonic response of the medium confined within the nanogap and can be
generally observed regardless of the materials. This medium-generalizability has been experimentally
confirmed by observing significant electrophotonic effects occur in both molecule-adsorbed and non-
adsorbed gold substrates, as described in the main text and Supplementary Section 10. Since this work
aims to present the first demonstration of giant electrophotonic modulation in TE-SHG and TE-SFG
processes, a systematic investigation of substrate morphology effects lies beyond the scope of the
present work. However, we agree that extending our research to include amorphous gold would be
valuable for further assessing the universality of angstrom-scale electrophotonic effects. We consider
that this is an important direction for future work. Thank you for your insightful comment.

Reviewer #4's comments 6 and 7

6) While the study offers compelling experimental evidence for giant near-field nonlinear modulation
in an ångström-scale plasmonic junction, the theoretical underpinnings remain underdeveloped. I note
that in Section 13 of the Supplementary Information, the authors introduce simulations to capture field
enhancement and its spectral evolution. However, this section is quite limited in detail.

To enhance the clarity and reproducibility of this part of the work, the authors should explicitly state
the underlying mechanism behind their numerical method used (FDTD), the dielectric functions
employed for gold and MBT, and whether nonlocal or quantum-corrected models were considered.
Given the sub-nanometer nature of the gap, such modeling choices are highly consequential.

More broadly, the manuscript would benefit significantly from deeper computational or analytical
exploration of:

- · The near-field distribution and its evolution under applied bias,
- · The interaction between light and tunneling carriers in such confined geometries,
- · The voltage-dependent transition from $\chi^{(2)}$ - to $\chi^{(3)}$ -dominated nonlinearities,
- · And the gap-size dependence of TE-SHG and TE-SFG efficiencies.

In addition, the physical origin of the observed nonlinear peaks remains insufficiently explored. A
decomposition of the nonlinear scattering response into electromagnetic modes—such as via quasi-
normal mode analysis or pole fitting of the scattering matrix—could provide insights into whether
dipolar, quadrupolar, or higher-order gap plasmons are involved.

Incorporating these theoretical and mechanistic insights would not only support the authors’
experimental claims but also deepen the conceptual impact of this important work.

7) Additionally, I noticed that Fig. S15 in the current Supplementary Information appears closely
related to Fig. 4 from the authors' prior publication (Takahashi et al., *J. Phys. Chem. Lett.* 2023).
However, it is unclear what modeling assumptions have changed between the two—particularly whether
the current work accounts for the spacer layer (e.g., MBT) when defining the tip-substrate gap in the
simulations. Could the authors clarify whether the SAM thickness was explicitly included in the field
enhancement calculations presented in Fig. S15? It would also be helpful to briefly explain how the
present modeling differs from the earlier work—both in physical assumptions and numerical parameters.

**Reply to reviewer #4’s comments 6 and 7**

First of all, we would like to clarify that the spectra in Fig. S16 in our revised Supplementary
Information are not newly obtained results in this work, but rather are reproduced from Fig. 4 in our
prior publication [*J. Phys. Chem. Lett.* **14**, 6919–6926 (2023)]. Therefore, the FDTD calculations
presented in Supplementary Information did not incorporate the presence of MBT layer between the
tip and substrate; instead, the tip–substrate gap was assumed to be vacuum. We would like to
emphasize that those results were presented solely as a reference to facilitate a clear understanding of
the broadband IR-to-visible near-field enhancement that is essential for the TE-SHG and TE-SFG
processes. To avoid misunderstanding, we have revised the Supplementary Information to more clearly

convey this point. We apologize for any confusion caused by the original presentation.

Although the simulation results presented in Supplementary Fig. S16 are reproduced from our
prior work, we fully agree that explicitly stating the simulation conditions, including the dielectric
functions and the treatment of quantum effects, is of great importance for enhancing the clarity and
reproducibility of our simulations for future readers. To this end, we have revised Supplementary
Information to explicitly state that (i) the values of gold refractive index were adopted from the
experimental work presented by Olmon *et al.* [*Phys. Rev. B* **86**, 235147 (2012)] and (ii) nonlocal or
quantum-corrected models were not explicitly considered in our simulations. We also provided
additional information that allows potential readers to reproduce our calculation, including the
simulation software that we used (Lumerical FDTD, Ansys), configurations of the simulation space,
and detailed description of optical sources and field monitors. Please see the revisions listed below.

We would also like to note that while the reviewer suggested addressing the underlying
mechanism behind the FDTD method, this technique is already a well-established tool for
electromagnetic field simulations and has been extensively utilized in previous studies [for example,
*Opt. Express* **21**, 25271–25276 (2013); *Opt. Express* **26**, 27668–27682 (2018); and *Nat. Commun.* **12**,
3465 (2021)]. A detailed description of its mechanism would therefore be redundant and beyond the
scope of the present work. For readers who may be interested in the fundamental concepts and
underlying mechanisms of the FDTD method, we have added the citation to relevant literature that
provides comprehensive descriptions [*IEEE Trans. Antennas Propagat.* **14**, 302 (1966); *IEEE Access*
**7**, 63852 (2019); *Nat. Rev. Methods Primers* **3**, 75 (2023)]. We believe that these revisions will both
clarify our methodological framework and improve the reproducibility of the presented results, thereby
ensuring transparency and facilitating future research building upon this work.

In addition, to clearly show the influence of the presence of the SAM layer on the field
enhancement spectra, we performed additional electromagnetic field simulations incorporating 6-Å-
thick SAM layer within the tip–substrate gap with 1-nm distance (Fig. IVa). The refractive index of
the SAM layer was set to be 1.2, which represents the typical value that has been used to calculate the
optical responses of interfacial SAM layers [*Phys. Rev. B* **59**, 12632 (1999); *J. Chem. Phys.* **148**,
134701 (2018); *J. Chem. Phys.* **159**, 164201 (2023)]. As shown in Figs. IVb and c, the presence of the
SAM layer does not significantly alter the spectral profiles of both field enhancement factor (Fig. IVb)
and radiation efficiency (Fig. IVc). Therefore, although our original discussion on the field
enhancement mechanisms was provided based on the vacuum gap, the same conclusion can be deduced
even in the presence of the SAM layer within the gap. We added the above discussion to
Supplementary Section 13. Please see the revisions listed below.

Fig. IV | Theoretical calculation of the near-field properties in the presence of the MBT layer within the gap. **a.** Schematic representation of the gap region including the MBT layer with a thickness of 6 Å and a refractive index of 1.2 (the light blue region). The metal-to-metal separation between the tip apex and the substrate surface was 1 nm. **b, c.** The red curves represent the spectra of **(b)** the field enhancement factor ($|K_{\text{gap}}|^2$) and **(c)** the radiation efficiency ($|L_{\text{gap}}|^2$) calculated under the MBT-introduced gap configuration shown in panel **(a)**. A rounded cone tip with a 30° opening angle, 50 nm radius of curvature, and 15000-nm length was adopted in the calculation. The calculated results obtained without incorporating the MBT layer (the blue curves in **b** and **c**) are also shown as a reference. Note that the blue curves are identical to the data at $l = 15000$ nm shown in Supplementary Fig. S16.

Finally, the reviewer suggested that we provide additional discussion from the theoretical and mechanistic perspective in order to more strongly support our experimental claims and deepen the conceptual impact of our achievements. In particular, the following five points were raised as desirable computational or analytical explorations:

- (a) The near-field distribution and its evolution under applied bias**
- (b) The interaction between light and tunneling carriers in such confined geometries**
- (c) The voltage-dependent transition from $\chi^{(2)}$ - to $\chi^{(3)}$ -dominated nonlinearities**
- (d) The gap-size dependence of TE-SHG and TE-SFG efficiencies**
- (e) The physical origin of the observed nonlinear peaks—contributions of dipolar, quadrupolar, or higher-order gap plasmons**

In the following, we provide point-by-point responses to these five issues.

(a) The near-field distribution and its evolution under applied bias

The spatial distributions of the oscillating near-field and electrostatic field calculated through the FDTD method have already been provided in Figs. I and II, respectively. Although the reviewer suggested further investigating applied bias dependence of the near-field enhancement represented by $|K_{\text{gap}}|^2$ and $|L_{\text{gap}}|^2$ and its spatial distribution, we would like to respectfully note that the influence of static bias has already been experimentally examined in a previous study [*Nat. Nanotechnol.* **15**, 105 (2020)]. In this prior work, by measuring tip-enhanced Raman scattering (TERS) spectra while varying the applied STM bias from 0.1 V to 1.0 V under a constant tip-substrate distance, the TERS intensity was found to be essentially independent of the applied bias. This result clearly demonstrates that the near-field enhancement factor remains unaffected by the applied bias (at least up to 1 V). Therefore, the bias-free FDTD calculations results employed in our analysis can be regarded as

sufficiently reliable and applicable even under biased conditions. Therefore, within the scope of the
present work, we consider that investigation of the bias-dependent evolution of near-field enhancement
would not be essential. We have revised the manuscript to convey this point more clearly. We
appreciate your important comment, which helped us refine the discussion.

**(b) The interaction between light and tunneling carriers in such confined geometries,**

We believe that this issue has already been thoroughly addressed in our responses to the reviewer's
comments 2 and 4. For further details, please kindly refer to our replies to those comments.

**(c) The voltage-dependent transition from $\chi^{(2)}$ - to $\chi^{(3)}$ -dominated nonlinearities**

As the reviewer pointed out, the overall near-field nonlinear optical response exhibits a bias-dependent
crossover between the $\chi^{(2)}$ and $E_{\text{DC}}\chi^{(3)}$ contributions. In the absence of the applied electrostatic
field, the near-field nonlinear optical responses are solely governed by the intrinsic $\chi^{(2)}$ effect, with
no contributions from $\chi^{(3)}$. As field strength increases, the field-induced $\chi^{(3)}$ effect becomes more
significant and eventually surpasses the contribution of $\chi^{(2)}$, making a transition from the $\chi^{(2)}$ -
dominated regime to the $E_{\text{DC}}\chi^{(3)}$ -dominated regime. This crossover can be quantitatively
characterized by defining a threshold field ($E_{\text{DC},0}$), at which the contributions of $\chi^{(2)}$ and $E_{\text{DC}}\chi^{(3)}$
become equal: $|\chi^{(3)}E_{\text{DC},0}| = |\chi^{(2)}|$. Combining this relation and the fitting results presented in the
main text ($|\chi^{(3)}|^2/|\chi^{(2)}|^2 = (10.3 \pm 0.4) \text{ nm}^2/\text{V}^2$), the value of $|E_{\text{DC},0}|$ is calculated as $3.1 \times$
10^8 V/m . Assuming the gap distance of $d = 7 \text{ \AA}$, the corresponding applied voltage is estimated to
be $\pm 0.22 \text{ V}$. Thus, when applied voltage exceeds this value, the system transitions from the $\chi^{(2)}$ -
dominated regime to the $E_{\text{DC}}\chi^{(3)}$ -dominated regime. We have revised the main text to briefly include
this explanation. We appreciate the reviewer's insightful comment, which led to this important
clarification.

**(d) The gap-size dependence of TE-SHG and TE-SFG efficiencies**

The gap-size dependence of TE-SHG efficiencies has already been addressed in detail in our reply to
the reviewer's comments 2 and 4. Importantly, the $4\text{--}7 \text{ \AA}$ gap distance range employed in this study
represents a crossover regime where the classically expected enhancement are counterbalanced by the
quantum suppression effects, making the overall field enhancement nearly maximized (Fig. III). Due
to this compensation, the overall field enhancement factor remains nearly constant at its maximum
value throughout this gap-size range, rendering it effectively independent of tip-substrate distance
between 4 and 7 \AA . Therefore, this distance range provides optimal field enhancement conditions for
both TE-SHG and TE-SHG processes, in terms of both strength and stability. The revisions
corresponding to this comment have been provided as our responses to the reviewer's comments 2 and
4. Thank you for your comments.

**(e) The physical origin of the observed nonlinear peaks—contributions of dipolar, quadrupolar,**

**or higher-order gap plasmons**

In general, the near-field nonlinear optical processes can be understood as consisting of three steps:
(e-i) field enhancement within the tip–substrate gap, (e-ii) excitation of nonlinear polarization by the
enhanced field, and (e-iii) radiation from the induced nonlinear polarization. Based on the
comprehensive discussion provided in the following, we can conclude that in the present study, all of
these processes are governed predominantly by dipolar contributions.

**(e-i) Field enhancement.** The incident laser is confined within the tip–substrate gap (Fig. I) and
undergoes strong near-field enhancement by several orders of magnitude. This enhanced near-field
arises from oscillating charges distributed across the tip and substrate. To elucidate the physical origin
of this enhancement, we calculated the charge density distribution in the gap region under the mid-IR
excitation by applying Gauss’s law to the spatial electric field distribution shown in Fig. I. Figure Va
displays the charge density distribution on the XZ plane ($Y = 0$ nm) under the 3280-nm excitation,
showing that charge neutrality is preserved within the bulk due to screening effects and charges are
localized at the metal surfaces. Similar charge distribution is predicted for the near-IR (1033 nm)
excitation conditions (Fig. Vb). Notably, under these mid- and near-IR excitation conditions, charges
of opposite signs are simply distributed across the tip and substrate without forming any multipole-
like complex charge distribution. Therefore, the plasmon mode generated under the IR irradiation can
be regarded as strongly dipolar in nature.

**Fig. V | Charge distribution across the tip and substrate on the XZ plane ($Y = 0$ nm) under the**
**IR excitation.** These charge densities were obtained by applying Gauss’s law ($\nabla \cdot E = \rho/\epsilon$) to the
spatial electric field distribution. Panels **a** and **b** represent the Fourier component corresponding to
3280-nm and 1033-nm excitation, respectively. The bulk region is electrically neutral due to the
screening effects, and charges with opposite signs are accumulated at the metal surfaces.

**(e-ii) Excitation of nonlinear polarizations.** The enhanced fields induce nonlinear polarizations
within the molecules. Importantly, as discussed in our reply to reviewer’s comment 1, the lateral size
of the field enhancement area is approximately 20 nm under the current ~50-nm-scale tip apex (Fig.
I). This scale of the field enhancement is nearly two orders of magnitude larger than the size of a single
MBT molecule (~6 Å). We recently demonstrated that under such conditions, where the near-field area
is far larger than the molecular size, the induced second-order nonlinear polarization within the

molecules can be well approximated as dipoles ($P^{(2)}$), with negligible contributions from quadrupoles
($Q^{(2)}$) or higher-order multipoles [arXiv:2509.09179 [physics.optics]].

**(e-iii) Radiation process.** The gap-mode plasmon contributes to amplifying the radiation from the
induced nonlinear dipolar polarizations. Importantly, the center wavelengths of TE-SHG and TE-SFG
outputs in this study were 750 nm and 785 nm, respectively, which are located in the visible region
but lie in its longer-wavelength edge, relatively close to the near-infrared region. Prior studies
demonstrated that gap plasmons in this wavelength range can be well approximated as dipolar modes
[*Nat. Commun.* **11**, 1021 (2020)]. Therefore, the radiation process is mediated predominantly by
dipolar gap-mode plasmons.

To more clearly convey the dominance of dipolar polarizations in steps (i)–(iii), we have
revised our manuscript to include the above discussion, together with Fig. V. Thank you again for your
insightful comment.

**Revisions for the former half of Comment 6 and for Comment 7**

Page 21, lines 515–540 in Supplementary Section 13: The procedure for electromagnetic field
simulations was described elsewhere^{22,28}. Briefly, the finite-difference time-domain (FDTD)
method^{29–31} was adopted with commercial software (Lumerical FDTD, Ansys). The system
investigated in the simulation consists of a gold tip positioned above a gold substrate in vacuum,
representing the nanogap in our STM. The refractive index of gold was taken from the experimental
values of Olmon *et al.*³² Perfectly matched layer boundary conditions were used in all simulations to
absorb all outgoing waves and eliminate light reflection. To evaluate the spectral properties of incident
field enhancement (K_{gap}), we placed a monitor at the midpoint between the tip apex and the substrate
surface to measure the electromagnetic field strength. A *p*-polarized Gaussian beam source with a
waist of 2 μm was used to illuminate the nanogap at an incident of 55°.

To evaluate the radiation efficiency (L_{gap}), an oscillating dipole source perpendicular to the
gold substrate, representing the nonlinear polarizations generated within the gap, was placed at the
same position as the monitor for K_{gap} . The selection of the dipole source is verified based on our
recent demonstration¹⁰: we experimentally and theoretically revealed that when the tip apex size is on
the order of ~50 nm as in the present study (Fig. 1b in the main text), the generated nonlinear
polarizations are highly dipole in nature, with no significant contributions from quadrupoles or higher-
order multipoles¹⁰. Radiated electromagnetic field from the dipole was monitored at a position where
the lateral and vertical distances from the dipole were 3000 and 2100 nm, respectively, corresponding
to the reflection angle of 55° employed in our experiments. Note that nonlocal or quantum-corrected
models^{15–21} were not explicitly considered in our simulations. Moreover, since the near-field
enhancement factor within the tip–substrate gap was revealed to be essentially unaffected by the STM
bias of ~1 V applied across a tip–substrate junction maintained at constant distance³³, the influence of

the applied bias was not incorporated in our FDTD simulation.

In the following, we review previously reported spectral characteristics of the field
enhancement factor and radiation efficiency^{22,28} to facilitate a clear understanding of the broadband
IR-to-visible near-field enhancement that is essential for the TE-SHG and TE-SFG processes.

Page 23, lines 615–623 in Supplementary Section 13: Finally, to show the influence of the presence of
the SAM layer on the field enhancement spectra, we performed additional electromagnetic field
simulations incorporating 6-Å-thick SAM layer within the tip–substrate gap with 1-nm distance (Fig.
S18a). The refractive index of the SAM layer was set to be 1.2, which represents the typical value that
has been used to calculate the optical responses of interfacial SAM layers^{51–53}. As shown in Figs. S18b
and c, the presence of the SAM layer does not significantly alter the spectral profiles of both field
enhancement factor (Fig. S18b) and radiation efficiency (Fig. S18c). Therefore, although the above
discussion on the field enhancement mechanisms is based on the vacuum gap, the same conclusion
can be deduced even in the presence of the SAM layer within the gap.

**Revisions for (a)**

Page 7, lines 178–180 in the main text: Moreover, as long as the tip–substrate distance is kept constant,
variations in the STM bias voltage on the order of 1 V essentially unaffected the near-field enhancement
factor within the gap⁶⁴.

Page 21, lines 534–537 in Supplementary Section 13: Moreover, since the near-field enhancement
factor within the tip–substrate gap was revealed to be essentially unaffected by the STM bias of ~1 V
applied across a tip–substrate junction maintained at constant distance³³, the influence of the applied
bias was not incorporated in our FDTD simulation.

**Revisions for (c)**

Page 10, lines 273–277 in the main text: From these relative values, we can estimate the threshold field
($E_{DC,0}$), at which $\chi^{(2)}$ and $\chi^{(3)}E_{DC}$ effects equally contributes to the nonlinear optical generation
($|\chi^{(3)}E_{DC,0}| = |\chi^{(2)}|$), to be 3.1×10^8 V/m. Assuming the gap distance of $d = 7$ Å, the
corresponding applied voltage is estimated to be ± 0.22 V. Thus, when applied voltage exceeds this
value, the contributions from $\chi^{(3)}E_{DC}$ becomes dominant over that from $\chi^{(2)}$.

**Revisions for (e)**

Page 9, lines 238–241 in the main text: Note that our recent work demonstrated that, when the tip apex
size is on the order of ~50 nm as in the present study (Fig. 1b), the generated nonlinear polarizations
are predominantly dipolar in nature, with negligible contributions from quadrupolar or higher-order
multipolar components⁷⁰.

Page 21, lines 527–531 in Supplementary Section 13: The selection of the dipole source is verified
based on our recent demonstration¹⁰: we experimentally and theoretically revealed that when the tip
apex size is on the order of ~50 nm as in the present study (Fig. 1b in the main text), the generated
nonlinear polarizations are highly dipole in nature, with no significant contributions from quadrupoles
or higher-order multipoles¹⁰.

Page 22, lines 572–576 in Supplementary Section 13: Note that the center wavelengths of TE-SHG
and TE-SFG outputs in this study (750 nm and 785 nm, respectively) are covered by the gap-mode
resonance band in the visible region, though they are located in the longer-wavelength edge. Prior
studies demonstrated that gap-mode plasmons in this longer wavelength range can be well
approximated as dipolar modes²⁷. Therefore, the radiation process is mediated predominantly by
dipolar gap-mode plasmons.

Page 27, lines 705–714 in Supplementary Section 14: Finally, to elucidate the physical origin of the
near-field enhancement at the tip–substrate gap, we calculated the charge density distribution in the
gap region by applying Gauss’s law ($\nabla \cdot E \propto \rho$) to the spatial electric field distribution shown in Fig.
S19. Figure S21a displays the charge density distribution on the XZ plane ($Y = 0$ nm) under the 3280-
765 nm excitation, showing that charge neutrality is preserved within the bulk due to screening effects and
766 charges are localized at the metal surfaces. Similar charge distribution is predicted for the near-IR
(1033 nm) excitation conditions (Fig. S21b). Notably, under these mid- and near-IR excitation
conditions, charges of opposite signs are simply distributed across the tip and substrate without
forming any multipole-like complex charge distribution. Therefore, the plasmon mode generated under
the near-IR irradiation can be regarded as strongly dipolar in nature.

Reviewer #4’s comment 8

Page2, Lines35-39: Slightly, less critical but still important point is that regarding the very first part of
introduction, the Authors stated that “Plasmonic functionalities are determined by the intrinsic optical
properties of materials and the static geometry of the plasmonic structures (e.g., size, shape, and
surrounding environment), making it difficult to adjust the plasmonic properties after the geometry of
the plasmonic nanostructure is fixed.” While I understand the statement but it is misleading the
potential readers from the point of when the geometry fixed it is hard to tune/control the optical
properties. There are several studies counter actively showing the tunability of the intrinsic optical
properties by changing the size and hybridizing the material content while keeping/maintaining the
shape of the determined geometry.

Single material:

[1] Campos, A., Troc, N., Cottancin, E. et al. Plasmonic quantum size effects in silver nanoparticles
are dominated by interfaces and local environments. Nat. Phys. 15, 275 - 280 (2019).

<https://doi.org/10.1038/s41567-018-0345-z>

[2] Ringe, Emilie, et al. "Plasmon length: a universal parameter to describe size effects in gold
nanoparticles." The journal of physical chemistry letters 3.11 (2012): 1479-1483.

[3] Henry, A.I., Bingham, J.M., Ringe, E., Marks, L.D., Schatz, G.C. and Van Duyne, R.P., 2011.
Correlated structure and optical property studies of plasmonic nanoparticles. The Journal of Physical
Chemistry C, 115(19), pp.9291-9305.

For the hybrid/colloidal formation:

[4] Rossner, Christian, Tobias AF König, and Andreas Fery. "Plasmonic properties of colloidal
assemblies." Advanced Optical Materials 9.8 (2021): 2001869.

[5] Kilic, Ufuk, et al. "Broadband enhanced chirality with tunable response in hybrid plasmonic helical
metamaterials." Advanced Functional Materials 31.20 (2021): 2010329.

[6] Kılıç, U., Mock, A., Feder, R., Sekora, D., Hilfiker, M., Korlacki, R., Schubert, E., Argyropoulos,
C. and Schubert, M., 2019. Tunable plasmonic resonances in Si-Au slanted columnar heterostructure
thin films. Scientific reports, 9(1), p.71.

I thereby kindly recommend that the author shines a light on the path of potential readers by showing
the efforts being made and are indeed exist some studies available in the literature that some geometries
can provide the opportunity still spectrally and amplitude-wise tailor the optical properties.

**Reply to reviewer #4's comment 8**

We thank the reviewer for your important comment regarding our introductory statement. First of all,
we would like to clarify that our original intention was to convey the fundamental challenge in tuning
the optical properties of plasmonic nanostructures at their post-fabrication stage, where all structural
parameters including size, shape, constituent materials, and surrounding environment are fully fixed.
However, considering the reviewer's comment and carefully reviewing the cited literature, we
recognized that our original phrasing could make readers misunderstand that plasmonic nonlinear
optical responses allow no possibility for modulation. To avoid such misinterpretation, we revised our
introduction to highlight previous efforts to tune the plasmonic properties by changing the size, shape,
and material content, as well as citations to the literatures raised by the reviewer. In addition, we have
refined the question-raising sentences to more clearly convey the challenge of adjusting plasmonic
properties at the post-fabrication stage, thereby providing a smooth transition to our discussion of
electrical approaches to overcome the limitation in the tunability. Thank you for your insightful
comment.

**Revisions**

Page 2, lines 34–41 in the main text: Nonetheless, active control of plasmonic nonlinear optical effects
is **often** constrained by limited tunability. Plasmonic functionalities are **primarily governed** by the
intrinsic optical properties of **constituent** materials and the static geometry of the plasmonic structures
(e.g. size, shape, **material composition**, and surrounding environment). **Once these structural**

parameters are fully fixed, the plasmonic response becomes uniquely determined, rendering post-
fabrication adjustment inherently challenging. Consequently, although considerable efforts have been
devoted to control plasmonic properties by tuning the size, shape, and material content^{26–31}, efficient
post-fabrication tuning of plasmonic nonlinear optical properties still remains an open challenge.

**Reviewer #4's comment 9**

Lastly, the manuscript's novelty is enhanced by ambient-pressure operation. However, reviewer 3's
concern about dielectric breakdown remains partially open. I think that authors kindly include the gas
breakdown discussion in the main text or supplementary (it is currently only in their response letter),
perhaps citing the limited gas volume ($<10^{-20}$ cm³) and impossibility of avalanche ionization as
justification.

**Reply to reviewer #4's comment 9**

We thank the reviewer for acknowledging that the demonstration of angstrom-scale electrophotonic
effects under ambient conditions highlights the novelty of our work. In accordance with the reviewer's
suggestion, we revised the main text and Supplementary Information to include the discussion on the
impossibility of dielectric breakdown. Please refer to the revisions listed below. We believe that these
revisions will facilitate a proper understanding for future readers regarding the absence of dielectric
breakdown and associated tip/substrate damage within the angstrom-scale gap even under ambient
conditions, further ensuring the reliability of our results. Thank you for your valuable comment.

**Revisions**

Page 13, lines 353–357 in the main text: In addition, the possibility of the dielectric breakdown caused
by an intense electrostatic field within the gap ($\sim 10^9$ V/m) can also be reasonably ruled out in our
ambient experimental conditions, because of the extremely rare electron–gas collision within our
angstrom-scale gap (further details in Supplementary Section 9). This allows us to safely disregard any
intense-field-induced modification of the tip or substrate even under ambient experimental conditions.

Page 13, lines 310–329 in Supplementary Section 9: Notably, the absence of the structural
modification of the tip and substrate indicates that dielectric breakdown within the gap and associated
breakdown-induced damage to either the sample or the tip can be ruled out, even under the application
of a strong electrostatic field ($\sim 10^9$ V/m) across the gap. The impossibility of the dielectric breakdown
can be correctly interpreted by considering the probability of the electron–gas collision within our
angstrom-scale gap. Generally, the dielectric breakdown is induced by field emission of electrons from
an electrode into gas phase, avalanche ionization of gas molecules, and ballistic acceleration of charge
carriers within the interelectrode gap, leading to physical damage of the electrodes¹⁴. Thus, the
collision process between electrons and gas molecules plays a key role in the emergence of dielectric

breakdown. However, since the angstrom-scale tip–substrate distance employed in our experiments
ensures the overlap of electronic wavefunctions at the tip and substrate surfaces, the interelectrode
electron transfer occurs predominantly via the tunneling process, rather than ballistic field emission of
electrons into the gas phase. This prevents electron–gas collisions that would otherwise trigger
avalanche ionization of gas molecules, resulting in the absence of the dielectric breakdown. Moreover,
considering the number density of gas molecules under ambient conditions ($2.69 \times 10^{19} \text{ cm}^{-3}$), the
extremely small volume of the angstrom-scale gap ($\sim 10^{-20} \text{ cm}^3$) allows for the presence of at most a
few gas molecules within the gap. This is insufficient to sustain the cascade ionization processes,
making gas-phase electric breakdown physically impossible even under ambient experimental
conditions. Therefore, the possibility of electric discharges within our angstrom-scale gap structure
can be clearly ruled out, and any discharge-induced modification of the tip or substrate can be safely
disregarded.

Reviewer #5

**Reviewer #5's general comments**

The manuscript by S.Takahashi et al. presents a study on the tip-enhanced nonlinear optical processes
on plasmonic thin films. They experimentally demonstrate the TE-SHG, TE-SFG and so on by utilizing
an angstrom-scale plasmonic gap between a metallic tip and a flat metal substrate in a scanning
tunneling microscope. They have found that such plasmonic gaps can experience strong near-field
enhancement, which leads to a modulation factor up to 2000%/V in the EFISH process. Their results
are detailed and well-organized, and may open new avenue for designing angstrom-level photonic
devices. However, I still have several questions and comments about this manuscript before
recommend it for publication. Below are my specific comments:

**Reply to reviewer #5's general comments**

We sincerely thank the reviewer for evaluating our results as detailed, well-organized, and potentially
impactful for the development of angstrom-level photonic devices. In response to the reviewer's
questions and comments, we provide point-by-point replies and associated revisions to our manuscript
in the following. We believe that these revisions will help us more clearly convey the precise physical
interpretation, reliability, and impact of our achievements to broad readership of *Nature*
*Communications*.

**Reviewer #5's comments 1 and 2**

1. The authors have measured the relative value of the second order and third order nonlinear
susceptibilities. As a general question in the area of nonlinear optics, what is the SHG conversion
efficiency? Is there a solution to improve this efficiency?

2. The authors have calculated the enhancement spectra of Au tip, but it seems that this resonance is
not close to the fundamental wavelength. Is it possible to manipulate the plasmonic resonance of the
Au tip to get a larger enhancement factor?

**Reply to reviewer #5's comments 1 and 2**

Thank you for your insightful comments. In response to the reviewer's comment 1, we calculated the
conversion efficiencies of TE-SHG and TE-SFG from the SAM-adsorbed gold substrate and plotted
them against the applied bias (Fig. VI). Based on the FF-SHG intensity spectrum shown in
Supplementary Fig. S7, the conversion efficiency of FF-SHG is estimated to be on the order of 10^{-14} .
In contrast, the TE-SHG and TE-SFG exhibited higher efficiencies reaching the order of 10^{-11} and 10^{-12} ,
respectively (Fig. VI), indicating that the nonlinear optical efficiency is greatly enhanced by near-
field electric field confinement. Importantly, this enhancement occurs despite the extremely small
number of molecules involved in the TE-SHG/TE-SFG processes: based on the ~20-nm-scale field

enhancement area (Fig. S19 in our revised Supplementary Information) and the previously reported
 molecular density of the MBT SAM ($\sim 4 \times 10^{14} \text{ cm}^{-2}$) [*J. Phys. Chem. C* **111**, 6335–6342 (2007)], only
 $\sim 10^3$ molecules are estimated to contribute to TE-SHG/TE-SFG processes. In contrast, the FF-SHG
 process should involve at least $\sim 10^9$ molecules within the micrometer-scale optical focus spot. Thus,
 the per-molecule efficiency in the near-field scheme (10^{-14} – 10^{-15}) is much more significant than that
 in the far-field condition ($\sim 10^{-23}$), clearly demonstrating the crucial role of the near-field enhancement
 (10^9 – 10^8) in the nonlinear optical generation processes. We revised the manuscript to include the above
 discussion, together with Fig. VI. It should be noted that $\sim 10^3$ molecules are still too few to satisfy
 macroscopic phase-matching conditions, and the TE-SHG/TE-SFG signals under the present
 experimental conditions can therefore be approximated as originating from single-dipole radiation
 [arXiv:2509.09179 [physics.optics]].

 **Fig. VI | Conversion efficiencies of TE-SHG (a) and TE-SFG (b) processes plotted against the**
 **sample bias.** The conversion efficiency data presented in panel (a) were obtained by converting the
 photon counts used to evaluate the modulation depth in Fig. 3 in the main text into optical power and
 normalizing them by the incident power. The data in panel (b) were calculated in the same manner
 from the measurements shown in Fig. 4e in the main text.

 The reviewer also raised the concern that the TE-SHG excitation wavelength (1500 nm) is
 detuned from the gap mode plasmon resonance in our Au tip–substrate gap. As theoretically discussed
 in Supplementary Section 13, however, strong field enhancements occur not only in the visible range
 but also in the IR region because the micrometer-scale tip shaft serves as an optical antenna that
 efficiently captures incident IR light [*J. Phys. Chem. Lett.* **14**, 6919 (2023)]. As shown in the blue
 curve at the bottom of Fig. S16e, such antenna effects of the tip shafts are operative across broad
 wavelength regions that encompass near- and mid-IR regions, efficiently covering the excitation
 wavelengths of TE-SHG (1500 nm) and TE-SFG (1033 nm and 3280 nm) conducted in this work.
 Furthermore, the presence of this strong IR enhancement effect was not only theoretically discussed
 but also experimentally corroborated in our recent work [*J. Phys. Chem. Lett.* **14**, 6919 (2023)]: we
 measured the excitation wavelength dependence of the TE-SHG intensity and confirmed the presence

of the broad enhancement effects in the IR range. Therefore, even if the excitation wavelengths of TE-
SHG and TE-SFG are not close to the gap-mode plasmon resonance in the visible range, they are still
sufficiently enhanced through the antenna effect specific to tip shafts without relying on the gap-mode
plasmon resonance.

On the other hand, the gap-mode plasmon plays a crucial role in amplifying the radiation from
the induced nonlinear polarization in the visible region. As shown in Fig. S16f, the radiation efficiency
from the tip–substrate gap is maximized in the visible region, where the gap-mode plasmon resonance
is present. Thus, the gap-mode plasmon can efficiently mediate the conversion of local oscillating
polarization within the gap into far-field radiation in the visible range. Since the output wavelengths
of TE-SHG and TE-SFG are located at this visible plasmonic resonance region, the gap-mode plasmon
plays a decisive role in enhancing the output radiation efficiency and contributes to the generation of
appreciable second-order nonlinear optical signals.

The overall signal intensity of TE-SHG (I_{TESHG}) is thus proportional to the product of the input
field enhancement factor (K_{gap}) at ω and output radiation efficiency (L_{gap}) at 2ω : $I_{\text{TESHG}}(2\omega) \propto$
$|K_{\text{gap}}(\omega)|^4 |L_{\text{gap}}(2\omega)|^2$ (equation (S3) in Supplementary Information). The intensity of TE-SFG
(I_{TESFG}) is described in a similar manner: $I_{\text{TESFG}}(\omega_{\text{SFG}}) \propto |K_{\text{gap}}(\omega_1)|^2 |K_{\text{gap}}(\omega_2)|^2 |L_{\text{gap}}(\omega_{\text{SFG}})|^2$
(equation (S4) in Supplementary Information). Therefore, to further improve the conversion
efficiencies, it is essential to tailor the enhancement spectrum such that the value of $|K_{\text{gap}}K_{\text{gap}}L_{\text{gap}}|^2$
is maximized. As clarified in our previous work [*J. Phys. Chem. Lett.* **14**, 6919 (2023)], the spectral
profiles of K_{gap} and L_{gap} are highly sensitive to the tip geometry. Specifically, the nanometer-scale
curvature of the tip apex primarily determines the gap-mode plasmon resonance at the visible region,
whereas the micrometer-scale surface geometry of the tip shaft governs the broader field enhancement
at the IR region through the antenna effect [*J. Phys. Chem. Lett.* **14**, 6919 (2023)]. Therefore, to
improve the field enhancement strengths at both the visible and IR regions, simultaneous control of
these different-scale structures through nanoscale adjustment of the apex curvature [*Appl. Phys. Lett.*
**109**, 153110 (2016)] and the introduction of grating patterns on the micrometer-scale tip shaft [*J. Phys.*
*Chem. Lett.* **3**, 945 (2012); *J. Opt.* **16**, 114003 (2014)] is essential, leading to further improvement of
the TE-SHG/TE-SFG efficiencies. Such fine control of tip geometries would be achieved by exploiting
more sophisticated tip processing techniques, such as field-directed sputter sharpening for the tip apex
[*Nat. Commun.* **3**, 935 (2012)] and focused ion beam processing for the tip shaft [*Nano Lett.* **19**, 3597
(2019)]. Although implementing these strategies lies beyond the scope of the present work, we believe
that this represents an important research direction for our future work, potentially leading to novel
strategies to improve the conversion efficiency of near-field nonlinear optical processes. We appreciate
the reviewer's insightful suggestions.

**Revisions**

Page 13, lines 357–363 in the main text: Furthermore, the near-field tip-enhancement scheme

constructed under room temperature achieves markedly higher SHG and SFG conversion efficiencies
than conventional far-field excitation, despite the much smaller number of molecules involved in the
signal generation process in the near-field scheme (further details in Supplementary Section 7). These
findings highlight the promising potential of the observed nonlinear electoplasmonic phenomena and
pave the way for the development of photonic devices that combine substantial modulation depth,
operation under ambient conditions, and higher conversion efficiency.

Page 11, lines 250–265 in Supplementary Section 7: In this section, we discuss the conversion
efficiencies of TE-SHG and TE-SFG responses of the SAM-adsorbed gold substrate (Fig. S10). Based
on the FF-SHG intensity spectrum shown in Fig. S7, the conversion efficiency of FF-SHG is estimated
to be on the order of 10^{-14} . In contrast, the TE-SHG and TE-SFG exhibited higher efficiencies on the
order of 10^{-11} and 10^{-12} , respectively (Fig. S10), indicating that the nonlinear optical efficiency is
enhanced by near-field electric field confinement. Importantly, this enhancement occurs despite the
extremely small number of molecules involved in the TE-SHG/TE-SFG processes: based on the ~10-
993 nm-scale field enhancement area (Fig. S19) and the previously reported molecular density of the MBT
SAM ($\sim 4 \times 10^{14} \text{ cm}^{-2}$)⁹, only $\sim 10^3$ molecules are estimated to contribute to TE-SHG/TE-SFG process.
In contrast, the FF-SHG process should involve at least $\sim 10^9$ molecules within the micrometer-scale
optical focus spot. Thus, while the absolute efficiencies are low, the per-molecule efficiency in the
near-field scheme (10^{-14} – 10^{-15}) is significantly higher than that in the far-field condition ($\sim 10^{-23}$),
clearly demonstrating the crucial role of the near-field enhancement (10^9 – 10^8) in the nonlinear optical
generation processes. It should be noted that $\sim 10^3$ molecules are still too few to satisfy macroscopic
phase-matching conditions, and the TE-SHG/TE-SFG signals under the present experimental
conditions can therefore be approximated as originating from single-dipole radiation¹⁰.

Page 23, lines 599–614 in Supplementary Section 13: To further improve the TE-SHG and TE-SFG
efficiencies, it is essential to tailor the enhancement spectrum such that the value of $|K_{\text{gap}}K_{\text{gap}}L_{\text{gap}}|^2$
terms in equations (S3) and (S4) are maximized. As discussed above and demonstrated in more detail
in our previous work²², the spectral profiles of K_{gap} and L_{gap} are highly sensitive to both nanometer-
and micrometer-scale tip geometry. Specifically, the nanometer-scale curvature of the tip apex
primarily determines the gap-mode plasmon resonance in the visible region, whereas the micrometer-
scale surface geometry of the tip shaft governs the broader field enhancement at the IR region through
the antenna effect²². Therefore, to improve the field enhancement strengths at both the visible and IR
regions, simultaneous control of these different-scale structures through nanoscale adjustment of the
apex curvature⁴⁶ and the introduction of grating patterns on the micrometer-scale tip shaft^{47,48} is
essential, leading to further improvement of the TE-SHG/TE-SFG efficiencies. Such fine control of
tip geometries would be achieved by exploiting more sophisticated tip processing techniques, such as
field-directed sputter sharpening for the tip apex⁴⁹ and focused ion beam processing for the tip shaft⁵⁰.
We believe that implementing these strategies to improve the overall near-field nonlinear optical

efficiencies represents an important research direction for our future work, potentially leading to novel
strategies to improve the conversion efficiency of near-field nonlinear optical processes.

**Reviewer #5's comment 3**

3. In the TE-SHG measurement, the SHG intensity first increases and then decreases while the gap
becomes larger. I suggest the authors discuss more about this phenomenon. Where is the maximum
intensity position? Is it related to the coupling effect or the interference of the plasmonic resonances?

**Reply to reviewer #5's comment 3**

Thank you for your important comment. As the reviewer correctly pointed out, the TE-SHG spectra in
Fig. 2a in the main text exhibit a non-monotonic dependence on the tip–substrate distance. In this
experiment, the gap distance was varied through two different procedures. (i) Increasing the applied
bias from +0.1 V to +0.75 V under the constant current mode expanded the gap distance from ~5 Å to
~7 Å, in accordance with the relation of $V = I_t G_0 \exp(-2\kappa d)$ (equation (S1) in Supplementary
Information). (ii) Then, the gap distance was further expanded to ~30 nm by mechanically retracting
the piezoelectric stage of the STM unit. As the reviewer pointed out, the TE-SHG intensity increased
in step (i), whereas it decreased in step (ii). These behaviors can be rationalized by considering the
fundamental mechanisms governing the tip-enhanced signal intensity.

The overall TE-SHG intensity (I_{TESHG}) is determined by the equation (S2) in Supplementary
Section 13: $I_{\text{TESHG}}(2\omega) \propto |\chi^{(2)} + \chi^{(3)} E_{\text{DC}}|^2 |K_{\text{gap}}(\omega)|^4 |L_{\text{gap}}(2\omega)|^2$. Thus, the observed intensity
variation is determined by two contributions: the electric field induced second harmonic (EFISH)
effect $|\chi^{(2)} + \chi^{(3)} E_{\text{DC}}|^2$, and the near-field enhancement factors $|K_{\text{gap}}(\omega)|^4 |L_{\text{gap}}(2\omega)|^2$. The
intensity decrease observed in step (ii) straightforwardly resulted from the significant reduction of field
enhancement factors ($|K_{\text{gap}}(\omega)|^4 |L_{\text{gap}}(2\omega)|^2$) due to the increased tip–substrate gap distance (~30
1042 nm). This is supported by our FDTD simulations based on classical electrodynamic models performed
to predict the near-field properties within the gap in step (ii) (Supplementary Fig. S22). In contrast, the
intensity increase observed in step (i) during the small gap-distance change from ~5 Å to ~7 Å cannot
be attributed solely to the classical field enhancement effects. This is because in this angstrom-scale
distances (4–7 Å), the classically predicted increase in field enhancement with decreasing gap size is
counteracted by quantum plasmonic quenching effects, making an overall field enhancement factor
nearly independent of the tip–substrate distance (Fig. VII) [*Nano. Lett.* **12**, 1333 (2012)]. In these
conditions, the observed bias dependent variation in the TE-SHG intensity is primarily determined by
the EFISH effects ($|\chi^{(2)} + \chi^{(3)} E_{\text{DC}}|^2$). Importantly, in the constant current mode in Fig. 2a, the gap-
distance expansion from ~5 Å to ~7 Å was accompanied by a voltage increase from +0.1 V to +0.75
1052 V, resulting in a corresponding increase in the electrostatic field below the tip apex (E_{DC}) from 2×10^8
V/m to 1×10^9 V/m. This substantial increase in E_{DC} strengthened the EFISH contribution, yielding
a higher TE-SHG intensity at 7 Å (Fig. 2a in the main text).

[REDACTED]

**Fig. VII | Influences of quantum plasmonic quenching in subnanometer gaps** [*Nano. Lett.* **12**, 10
57 1333 (2012)]. [REDACTED]

Notably, further reduction of the tip–substrate distance below $\sim 4 \text{ \AA}$ leads to a regime where
quantum plasmonic quenching becomes dominant, resulting in a steep decrease in the electric field
enhancement factors (Fig. VII) [*Nano. Lett.* **12**, 1333 (2012)]. In contrast, the $4\text{--}7 \text{ \AA}$ regime represents
a critical crossover region just before this steep decline, where the electric field enhancement is
maximized and remains largely insensitive to the gap-distance variations. Thus, we consider that our
TE-SHG/TE-SFG experiments under this $4\text{--}7 \text{ \AA}$ regime (Figs. 2–4 in the main text) were conducted
under conditions that not only offer optimal field enhancement but also provide ideal conditions for
directly probing the intrinsic bias dependence of the giant electrophotonic response.

To more clearly convey the underlying mechanisms of the nonmonotonic distance dependence
in Fig. 2a, we have revised the manuscript to include the above discussion. Thank you again for your
helpful comment.

**Revisions**

Page 6, lines 158–173 in the main text: This voltage increase at a constant tunneling current (500 pA)
extended the tip–sample distance from $\sim 5 \text{ \AA}$ to $\sim 7 \text{ \AA}$ (Fig. 2b). Based on conventional classical
electrodynamic simulations, the near-field enhancement strength is expected to decrease by several

tens of percents during this ~ 2 -Å tip–substrate distance elongation^{54–59}. However, at the gap distances
of < 1 nm, the influences of quantum mechanical phenomena, such as electron spill-out from the metal
surface and the overlap of electronic wavefunctions across the gap, begin to quench plasmon
excitations and suppress the increase in field enhancement^{53,56,57,60–63}. Particularly, in the gap distance
range of ~ 4 – 7 Å, these quantum suppression effects and the classically expected enhancement nearly
cancel each other, making the overall field enhancement effectively independent of the tip–substrate
distance^{53,56,57,60–63}. Therefore, the near-field enhancement strength should remain essentially
unaffected during the distance elongation from 5 Å to 7 Å. Despite such constant field enhancement,
the TE-SHG intensity significantly increased with the applied voltage (Fig. 2a). Importantly, this
distance expansion occurred when the bias voltage was raised from 0.1 V to 0.75 V at a constant
tunneling current (500 pA). With this voltage increase, the electrostatic field below the tip apex
increased from 2×10^8 V/m to 1×10^9 V/m. Therefore, both the TE-SHG intensity and the electrostatic
field increased concurrently, indicating the presence of substantial electrostatic field-induced effects
that boost the overall TE-SHG intensity.

Page 7, lines 176–183 in the main text: At this tip–substrate distance, the classically predicted near-
field enhancement and the quantum plasmonic quenching effects arrive at a nearly optimal balance,
resulting in almost maximum near-field enhancement values^{53,56,57,60–63}. Moreover, as long as the tip–
substrate distance is kept constant, variations in the STM bias voltage on the order of 1 V essentially
unaffected the near-field enhancement factor within the gap⁶⁴. Therefore, fixing the tip–substrate distance
at ~ 7 Å can be regarded as the optimal condition under which the field enhancement can be maintained
at its nearly maximum value regardless of the applied bias voltage. This enables us to exclusively
examine the influence of electrostatic fields on the TE-SHG process, while eliminating the
contributions of variations in the near-field enhancement.

Pages 17–18, lines 443–452 in Supplementary Section 11: Notably, further reduction of the tip–
substrate distance below ~ 4 Å leads to a regime where quantum plasmonic quenching becomes
dominant, resulting in a steep decrease in the electric field enhancement factors^{15–21}. In contrast, the
4 – 7 Å regime represents a critical crossover region just before this steep decline, where the electric
field enhancement is maximized^{15–21}. Due to this compensation, the overall field enhancement factor
remains nearly constant at its maximum value throughout this gap-size range, rendering it effectively
independent of tip–substrate distance from 4 Å to 7 Å. Thus, the 4 – 7 Å distance range not only offers
optimal field enhancement conditions for both TE-SHG and TE-SFG processes, but also provides an
ideal regime for exclusively probing the intrinsic bias dependence of the giant electrophotonic response,
distinct from variations in the near-field enhancement.

Reviewer #5's comment 4

4. P-polarized FWs are used in the measurement. Is it possible to measure the polarization states of the
SHG and SFG signals?

**Reply to reviewer #5's comment 4**

First of all, we would like to clarify that the TE-SHG/TE-SFG signals were detected without the
polarization filtering in order to avoid the signal attenuation by polarizers and maximize the signal-to-
noise ratio of the TE-SHG/SFG spectra. Therefore, the polarization states of the TE-SHG and TE-SFG
signals were not directly measured in our experiments. However, based on the currently available data,
we can reasonably conclude that the observed TE-SHG/SFG signals are *p*-polarized.

Figure S19 in our revised Supplementary Information displays the surface-normal and surface-
parallel components of the electric near-field within the tip–substrate gap, which were calculated
through the FDTD simulation for a tip with a curvature radius of 50 nm. This tip radius is comparable
to the size of the tip apex used in our experiments (Fig. 1b in the main text). As shown in Fig. S19, the
surface-normal (*Z*-directed) near-field (E_z) component is more than one order of magnitude stronger
than the surface-parallel field component, indicating that only E_z dominantly contributes to nonlinear
optical processes within the gap. In this case, among the 27 tensor components of the second-order
nonlinear susceptibility, only $\chi_{xzz}^{(2)}$, $\chi_{yzz}^{(2)}$, or $\chi_{zzz}^{(2)}$ can in principle contribute to the second-order
nonlinear polarization within the tip–substrate gap. When the cylindrical symmetry about the central
axis of the tip is satisfied, the contributions of $\chi_{xzz}^{(2)}$ and $\chi_{yzz}^{(2)}$ are eliminated, leaving $\chi_{zzz}^{(2)}$ as the
dominant component [*Int. Rev. Phys. Chem.* **24**, 191–256 (2005)]. Therefore, the generated second-
order nonlinear polarization is mainly oriented along the *Z*-axis, and the corresponding emitted field
is *p*-polarized.

We would like to note that, consistent with this argument, the dominance of $\chi_{zzz}^{(2)}$ was also
confirmed experimentally in our recent work [arXiv:2509.09179 [physics.optics]]. By measuring the
vibrationally resonant TE-SFG for methyl vibrations in MBT molecules, we revealed that the methyl
antisymmetric stretching modes exhibited negative $\text{Im}(\chi^{(2)})$ spectra. Based on the considerations of
the explicit values of hyperpolarizability tensors and angular distributions of MBT molecules, $\chi_{zzz}^{(2)}$
associated with the methyl antisymmetric modes exhibits negative imaginary part, whereas other
hyperpolarizability tensor components, such as $\chi_{xxz}^{(2)}$ and $\chi_{yyz}^{(2)}$, possess positive imaginary parts.
Consequently, the observation of the negative $\text{Im}(\chi^{(2)})$ indicates that $\chi_{zzz}^{(2)}$ component
predominantly contributes to the generation of the tip-enhanced second-order nonlinear polarization.
Therefore, not only theoretical analysis but also our experimental results [arXiv:2509.09179
[physics.optics]] consistently corroborate the predominance of $\chi_{zzz}^{(2)}$, indicating that near-field
nonlinear signal radiation is dominantly *p*-polarized.

Similar to the second-order case, the third-order polarization is also oriented along the *Z*-axis.
In addition to the oscillating near-field induced by the incident excitation light, the electrostatic field
generated through the voltage application across the tip–substrate gap is also dominated by its *Z*-

component (Fig. S20 in our revised Supplementary Information). Therefore, by analogy with the
second-order susceptibility, we conclude that only $\chi_{ZZZZ}^{(3)}$ component is active among all third-order
susceptibility tensor components. This ensures that the third-order nonlinear polarization in EFISH is
also aligned along the Z-axis and that the corresponding emitted field is *p*-polarized.

To address the reviewer's concern, we have revised the manuscript to include discussion of the
polarization state of the TE-SHG and TE-SFG signals, as well as the fact that no polarization selection
was applied in our measurements. We thank the reviewer for your helpful comment.

**Revisions**

**Page 9, lines 243–246 in the main text:** Since these intragap fields are dominated by surface-normal
(Z-directed) components (Supplementary Figs. S19 and S20), among the multiple tensor elements of
$\chi^{(2)}$ and $\chi^{(3)}$, only $\chi_{ZZZ}^{(2)}$ and $\chi_{ZZZZ}^{(3)}$ components dominantly contribute to the generation of TE-
SHG signals (further details in Supplementary Section 14).

**Page 15, lines 438–440 in the main text:** Those two kinds of signals were separately input into a bundle
fiber branched into two halves without polarization selection and directed into a spectrometer

**Pages 26–27, lines 675–704 in Supplementary Section 14:** Based on the spatial distributions and
relative intensities of the surface-normal and surface-parallel components, we can directly identify the
main nonlinear susceptibility tensor components contributing to the near-field nonlinear generation
and the resultant polarization state of emitted light. As shown in Fig. S19, the surface-normal (Z-
directed) near-field component (E_Z) is more than one order of magnitude stronger than the surface-
parallel field component, indicating that only E_Z dominantly contributes to nonlinear optical
processes within the gap. In this case, among the 27 tensor components of the second-order nonlinear
susceptibility, only $\chi_{XZZ}^{(2)}$, $\chi_{YZZ}^{(2)}$, or $\chi_{ZZZ}^{(2)}$ can in principle contribute to the second-order nonlinear
polarization within the tip–substrate gap. When the cylindrical symmetry about the central axis of the
tip is satisfied, the contributions of $\chi_{XZZ}^{(2)}$ and $\chi_{YZZ}^{(2)}$ are eliminated, leaving $\chi_{ZZZ}^{(2)}$ as the dominant
component⁵⁵. Therefore, the generated second-order nonlinear polarization is mainly oriented along
the Z-axis, and the corresponding emitted field is *p*-polarized.

**We would like to note that, consistent with this argument, the dominance of $\chi_{ZZZ}^{(2)}$ was also**
**confirmed experimentally in our recent work¹⁰. By measuring the vibrationally resonant TE-SFG for**
**methyl vibrations in MBT molecules, we revealed that the methyl antisymmetric stretching modes**
**exhibited negative $\text{Im}(\chi^{(2)})$ spectra. Based on the considerations of the explicit values of**
**hyperpolarizability tensors and angular distributions of MBT molecules, $\chi_{ZZZ}^{(2)}$ associated with the**
**methyl antisymmetric modes exhibits negative imaginary part, whereas other hyperpolarizability**
**tensor components, such as $\chi_{XXZ}^{(2)}$ and $\chi_{YYZ}^{(2)}$, are characterized by positive imaginary parts. Therefore,**
**the observation of the negative $\text{Im}(\chi^{(2)})$ indicates that $\chi_{ZZZ}^{(2)}$ component predominantly contributes**

to the generation of the tip-enhanced second-order nonlinear polarization. Therefore, not only
theoretical analysis but also our experimental results¹⁰ consistently corroborates the predominance of
$\chi_{ZZZ}^{(2)}$, indicating that near-field nonlinear signal radiation is dominantly *p*-polarized.

Similar to the second-order case, the third-order polarization is also oriented along the *Z*-axis.
In addition to the oscillating near-field induced by the incident excitation light (Fig. S19), the
electrostatic field generated through the voltage application across the tip–substrate gap is also
dominated by its *Z*-component (Fig. S20). Therefore, by analogy with the second-order susceptibility,
only $\chi_{ZZZ}^{(3)}$ component is active among all third-order susceptibility tensor components. This ensures
that the third-order nonlinear polarization in EFISH is also aligned along the *Z*-axis and that the
corresponding emitted field is *p*-polarized.

**Reviewer #5's comment 5**

5. Minor comments: a. the illustration of the backward/forward scattering schemes should be added;
b. the font size in figures should be optimized.

**Reply to reviewer #5's comment 5**

a. In accordance with the reviewer's comment, we modified Figs. 1c and 4b in the main text to include
the illustration of both backward- and forward-scattering schemes.

b. In accordance with the reviewer's comment, we modified the font size in figures in the main text
and Supplementary Information to enhance the visibility for the readers in the future.

Thank you for the reviewer's careful attention to details of our manuscript and constructive
feedback.

Responses to reviewers' comments

The authors acknowledge the reviewers' fruitful comments and suggestions on our manuscript (manuscript number: NCOMMS-25-44074-A). Please find below a detailed response to each comment. The reviewers' comments and corresponding changes in the revised manuscript are highlighted in blue letters and yellow backgrounds, respectively.

Reviewer #3

Reviewer #3's comments

The authors addressed reasonably my concerns.

Reply to reviewer #3's comments

We thank the reviewer for acknowledging the minimality of electric discharge effects under ambient conditions. We also sincerely appreciate the reviewer's valuable and constructive comments throughout this review process, which have helped us to significantly improve the clarity and impact of our manuscript.

Reviewer #4

**Reviewer #4's general comments**

As I noted in my previous review preamble, the work submitted by Takahashi et al. is timely and of
broad interest to the nonlinear optics, nanophotonics, and scanning-probe communities. In my view,
the work is merit to get published in Nature Communications. However I have few more response to
the authors review and I believe that once the following points are also addressed, I can recommend
for its publication in Nature Communications. Below, I further outline some points and suggestions
below;

**Reply to reviewer #4's general comments**

We sincerely appreciate the reviewer's positive and encouraging evaluation of our work, and for
recognizing that our study merits publication in *Nature Communications*. In response to the reviewer's
additional comments, we provide point-by-point replies and the corresponding revisions below. We
believe that these revisions further improve the clarity and technical transparency of the manuscript.

**Reviewer #4's comment 1**

Plasmonic Tunability and Field Modulation: While the authors made substantial improvements in
addressing physical modeling and parasitic effects, one point appears to remain unaddressed: the
introductory statement implying that fixed-geometry plasmonic systems are untunable. Given the
extensive literature on active modulation via hybridization and field-driven effects (including EFISH
itself), I recommend revising the statement to better reflect the current state of tunability in plasmonic
nanostructures.

**Reply to reviewer #4's comment 1**

In accordance with the reviewer's comments, we carefully re-examined the relevant literature
including both experimental reports and review articles on active plasmonics (Refs. 32–48 in the main
text), and have revised the introductory statements to more explicitly convey the extensive efforts
devoted to achieving active modulation and control of plasmonic nanostructures. We believe that these
revisions ensure that the current state of tunability in plasmonic systems and their optical responses is
more accurately clarified. We sincerely appreciate the reviewer's insightful comment.

**Revisions**

Page 2, lines 34–44 in the main text: These plasmonic functionalities are fundamentally governed by
the intrinsic optical properties of constituent materials and the static geometry of the plasmonic
structures, such as size, shape, material composition, and surrounding environment^{26–31}. Although
these static structural properties are typically difficult to adjust at their post-fabrication stage, designing
active plasmonic systems that respond to external stimuli, such as light, heat, mechanical force, or

electric field, provides a route toward active modulation of both linear^{32–39} and nonlinear^{40–46}
 plasmonic responses. Since such tunable plasmonic platforms are essential for advanced nanophotonic
 applications^{47,48}, approaches that allow more flexible and efficient active control of plasmonic
 responses are highly desirable.

One promising strategy for active control of plasmonic responses is to directly exploit the
 metallic nature of plasmonic systems.

**Reviewer #4’s comment 2**

2) Electrostatic Field Distribution: Clarification and Visual Support: The authors mention that the STM
 tip–substrate junction exhibits a non-uniform electrostatic field due to its curved apex geometry, in
 contrast to a parallel-plate capacitor. To enhance clarity and pedagogical value especially for readers
 outside the STM community, I recommend adding a simple schematic and FEM simulation illustrating
 this spatial field distribution as shown for their tip-substrate case. Such a visual comparison would
 strengthen the EFISH argument and help contextualize how Å-scale tip displacements can significantly
 modulate the local nonlinear response.

**Reply to reviewer #4’s comment 2**

In accordance with the reviewer’s comments, we have replaced Fig. S20 in our previous version of
 Supplementary Information with Fig. I shown below, which includes schematic illustrations of
 electrostatic field distributions in a tip–substrate nanogap (Fig. Ic) and in a parallel-plate capacitor (Fig.
 Id) under an applied bias. We agree that this visual comparison will help us more clearly convey the
 origin of the inherently non-uniform electrostatic field distribution within the tip–substrate gap with a
 curved tip geometry. Thank you for your comment.

**Fig. I | Spatial distributions of the electrostatic field E_{DC} across the tip-substrate nanogap.** (a,
 b) The surface-normal (a) and surface-parallel (b) electrostatic field components plotted in the XZ
 plane ($Y = 0$ nm). These images are identical to those shown in Fig. S20 in our previous version of
 Supplementary Information. (c, d) Schematic illustrations of electrostatic field distributions within a

tip–substrate nanogap and (c) a parallel-plate capacitor (d) under an applied bias V . The orange arrows
indicate the electric flux lines within the gaps, illustrating non-uniform field distribution in the tip–
substrate nanogap arising from the curved tip geometry (c) and uniform field distribution in the
parallel-plate geometry (d).

**Revision**

Page 28, lines 724–730 in Supplementary Section 14: Furthermore, by using the image charge
method⁵⁴, we also revealed that the electrostatic field across the gap ($E_{DC}(\mathbf{r})$) also exhibits a similar
spatial distribution (Figs. S22a and b). This spatial non-uniformity of $E_{DC}(\mathbf{r})$ (Figs. S22a and b) can
be attributed to the curved geometry of the tip apex (Fig. S22c), unlike a parallel-plate capacitor, where
the electrostatic field strength between electrodes is spatially uniform (Fig. S22d). Consequently, both
$E_{gap}(\mathbf{r})(= K_{gap}(\mathbf{r})E_0)$ and $E_{DC}(\mathbf{r})$ exhibit spatial variations, giving rise to position-dependent
nonlinear polarizations within the nanoscale region beneath the tip apex.

**Reviewer #4's comment 3-1**

3) Clarification of Enhancement Factors and Reference Baselines: I have to be critical on the
enhancement calculations or measurements. Throughout the manuscript, the authors cite various
enhancement factors and field strengths (e.g., 2000%/V TE-SFG modulation, 10^8 – 10^9 V/m fields, and
>40× enhancement in air). However, the methodology used to define and calculate these enhancement
factors is unclear. Could the authors explicitly clarify:

What is the reference intensity or configuration used for computing each enhancement value (e.g., zero
bias, retracted tip, or no-gap control)?

Is the reported 2000%/V slope extracted from normalized intensity fits (e.g., Fig. 4e)? If so, what was
the normalization basis?

**Reply to reviewer #4's comment 3-1**

We appreciate the reviewer's valuable questions regarding the reference intensity of the reported
2000%/V TE-SHG/TE-SFG modulation. As described in the original manuscript, the relative
modulations of TE-SHG/TE-SFG intensities plotted in Figs. 3 and 4e are defined as $\Delta I(V)/I(V = 0)$,
where $I(V = 0)$ denotes the TE-SHG/TE-SFG intensity at $V = 0$ V, and $\Delta I(V) = I(V) - I(V = 0)$
represents the bias-induced intensity change in the nonlinear optical signals. Therefore, the reference
intensity mentioned by the reviewer corresponds to the zero-bias limit of the TE-SHG/TE-SFG
intensity $I(V = 0)$. Under our STM-based experimental conditions, however, direct measurement of
$I(V = 0)$ is technically difficult. This is because a net tunneling current becomes zero at $V = 0$ V,
preventing the operation of the feedback loop of STM to maintain an angstrom-scale tip–substrate gap
distance. To circumvent this limitation, we approximated $I(V = 0)$ as the average TE-SHG/TE-SFG
intensities obtained at small positive and negative biases (± 0.1 V): $I(V = 0) \approx [I(V = +0.1 \text{ V}) +$
$I(V = -0.1 \text{ V})]/2$. This approximation is justified because the measured bias dependence of $I(V)$ is

well described by a parabolic function, and the intensities at ± 0.1 V are much smaller than those at ± 1
123 V. Consequently, the variation of $I(V)$ within the ± 0.1 V range is sufficiently small, and the averaged
value not only provides a reasonable estimate of the zero-bias limit but also has almost no influence
on the quantitative evaluation of the relative modulation depth $\Delta I(V)/I(V = 0)$. Therefore, this
averaged value was adopted as the reference intensity for quantitatively evaluating the relative
modulation.

As shown by the blue open circles in Figs. 3 and 4e, the relative intensity modulation
($\Delta I(V)/I(V = 0)$) obtained by using this zero-bias limit intensity reaches $\sim 2000\%$ at approximately
± 1 V. We would like to emphasize that this modulation depth value ($\sim 2000\%/V$) was directly estimated
from the experimentally measured TE-SHG/TE-SFG intensities, rather than from fitting analysis based
on Eq. 1 in the main text. Note that the curve fittings presented in Figs. 3 and 4e were performed solely
to quantify the relative contributions of $\chi^{(2)}$ and $\chi^{(3)}E_{DC}$ terms governing the intensity profiles, and
not used to quantitatively determine the modulation depth of $2000\%/V$. We have revised our
manuscript to more explicitly clarify the above explanation. Thank you for your comments.

**Revision**

Page 7, lines 194–197 in the main text: More remarkably, the experimentally obtained relative
modulation in the TE-SHG signal intensity ($\Delta I_{\text{TESHG}}(V)/I_{\text{TESHG}}(V = 0)$) reached $\sim 2000\%$ at $V = \pm 1$
V (the blue open circles in Fig. 3), indicating the achievement of $\sim 2000\%/V$ giant modulation depth.

Page 8, lines 232–234 in the main text: The reference intensity $I_{\text{TESHG}}(V = 0)$ was approximated as
the average TE-SHG intensities obtained at small positive and negative biases (± 0.1 V):
$I_{\text{TESHG}}(V = 0) \approx [I_{\text{TESHG}}(V = +0.1 \text{ V}) + I_{\text{TESHG}}(V = -0.1 \text{ V})]/2$.

Page 14, lines 383–385 in the main text: Similar to the case of bias-dependent TE-SHG measurements
(Fig. 3), the reference intensity $I_{\text{TESFG}}(V = 0)$ was approximated as the average TE-SFG intensities
at $V = -0.1$ V and $V = 0.1$ V.

Page 14, lines 392–394 in the main text: the experimentally obtained $\Delta I_{\text{TESFG}}(V)$ (the blue open
circles in Fig. 4e) exhibited a quadratic dependence on the applied voltage

Page 18, lines 515–528 in the main text: The relative modulations of TE-SHG/TE-SFG intensities
plotted in Figs. 3 and 4e are defined as $\Delta I(V)/I(V = 0)$, where $I(V = 0)$ denotes the TE-SHG/TE-
SFG intensity at $V = 0$ V, and $\Delta I(V) = I(V) - I(V = 0)$ represents the bias-induced intensity
change in the nonlinear optical signals. Under our STM-based experimental conditions, however,
direct measurement of $I(V = 0)$ is technically difficult. This is because a net tunneling current
becomes zero at $V = 0$ V, preventing the operation of the feedback loop of STM to maintain an
angstrom-scale tip–substrate gap distance. To circumvent this limitation, we approximated $I(V = 0)$

as the average TE-SHG/TE-SFG intensities obtained at small positive and negative biases (± 0.1 V):
 $I(V = 0) \approx [I(V = +0.1 \text{ V}) + I(V = -0.1 \text{ V})]/2$. This approximation is justified because the
 measured bias dependence of $I(V)$ is well described by a parabolic function, and the intensities at
 ± 0.1 V are much smaller than those at ± 1 V. Consequently, the variation of $I(V)$ within the ± 0.1 V
 range is sufficiently small, and the averaged value not only provides a reasonable estimate of the zero-
 bias limit but also has almost no influence on the quantitative evaluation of the relative modulation
 depth $\Delta I(V)/I(V = 0)$. Therefore, this averaged value was adopted as the reference intensity for
 quantitatively evaluating the relative modulation.

**Reviewer #4's comment 3-2**

Have the authors performed any field-distribution simulations (e.g., FEM or FDTD) to validate that
 the field enhancement scales consistently with the tip–substrate separation and geometry?
 Could they also provide a sensitivity analysis showing how ± 0.5 Å variations in gap size or voltage
 influence the enhancement trends?
 Providing these clarifications would significantly strengthen the quantitative credibility of the reported
 modulation factors and ensure reproducibility by the broader community.

**Reply to reviewer #4's comment 3-2**

Thank you for your valuable comments. In the earlier versions of the manuscript, the fitting analysis
 of the bias-dependent TE-SHG intensities (Fig. 3 in the main text) was performed under the assumption
 that the intragap electrostatic field E_{DC} is given by the ratio between the STM bias voltage V and
 the minimum tip–substrate separation d : $E_{\text{DC}} = V/d$. However, the reviewer's comments made us
 reconsider the validity of this treatment, and we now recognize that such description is oversimplified
 assumption because the inherently non-uniform spatial distribution of the electrostatic field, which
 arises from the curved tip geometry (Fig. I), was not taken into account. In the actual tip–substrate gap,
 the electrostatic field E_{DC} , as well as the field enhancement factor K_{gap} and the radiation efficiency
 L_{gap} , exhibits spatial distributions (Supplementary Section 14), and thus the TE-SHG electric field
 emitted from individual nonlinear polarizations depends on their absolute spatial position. In such
 conditions, the experimentally observed TE-SHG intensity should be described as the coherent sum of
 these position-dependent contributions.

Specifically, the overall TE-SHG intensity (I_{TESHG}), including the contributions from both the
 bias-independent second-order signal and the E_{DC} -induced third-order signal, is given by the total
 intensity of SHG fields emitted from N molecules within the gap:

$$194 \quad I_{\text{TESHG}} \propto N^2 \left| \chi^{(2)} \langle L_{\text{gap}}(\mathbf{r}_n) K_{\text{gap},z}(\mathbf{r}_n)^2 \rangle + \chi^{(3)} \left\langle \frac{L_{\text{gap}}(\mathbf{r}_n) K_{\text{gap},z}(\mathbf{r}_n)^2}{d(\mathbf{r}_n)} \right\rangle V \right|^2 I_0^2, \quad (\text{R1})$$

where $\langle \dots \rangle$ denotes spatial averaging over molecules present within the gap region; \mathbf{r}_n is the position

of n -th molecule; $d(\mathbf{r}_n)$ represents the position-dependent tip–substrate distance determined by the
 curvature radius of the tip apex; $K_{\text{gap},z}(\mathbf{r}_n)$ is an enhancement factor of Z -directed field; and
 $L_{\text{gap}}(\mathbf{r}_n)$ is efficiency of radiation from a nonlinear polarization located at the position \mathbf{r}_n . This
 equation can be simplified by introducing an effective tip-substrate distance d_{eff} , which satisfies the
 relation

$$201 \quad \left\langle \frac{L_{\text{gap}}(\mathbf{r}_n)K_{\text{gap},z}(\mathbf{r}_n)^2}{d(\mathbf{r}_n)} \right\rangle = \frac{\langle L_{\text{gap}}(\mathbf{r}_n)K_{\text{gap},z}(\mathbf{r}_n)^2 \rangle}{d_{\text{eff}}}. \quad (\text{R2})$$

Substituting this expression into Eq. R1 yields

$$203 \quad I_{\text{TESHG}} \propto N^2 \left| \chi^{(2)} + \chi^{(3)} \left(\frac{V}{d_{\text{eff}}} \right) \right|^2 \left| \langle L_{\text{gap}}(\mathbf{r}_n)K_{\text{gap},z}(\mathbf{r}_n)^2 \rangle \right|^2 I_0^2. \quad (\text{R3})$$

By defining V/d_{eff} term as $E_{\text{DC,eff}}$ and omitting N^2 and $|\langle L_{\text{gap}}(\mathbf{r}_n)K_{\text{gap},z}(\mathbf{r}_n)^2 \rangle|^2 I_0^2$ terms for
 simplicity, we arrive at the simplified expression for EFISH similar to that given in Eq. 1 in the main
 text. In this refined formulation, d_{eff} represents an effective spatial average of the tip–substrate
 distance within the gap, weighted by the near-field intensity distribution and the curvature of the tip
 apex (Eq. R2). Thus, the expression of $E_{\text{DC,eff}} = V/d_{\text{eff}}$ effectively incorporates the influences of
 spatial field inhomogeneity. Importantly, even within these revised formalisms, the fundamental
 concept that the nonlinear optical signal quadratically depends on the applied voltage V remains
 unchanged because $E_{\text{DC,eff}}$ is still proportional to V ($E_{\text{DC,eff}} = V/d_{\text{eff}}$). Therefore, our fitting
 analysis of the bias-dependent TE-SHG intensity based on a quadratic function (Fig. 3 in the main
 text) is still justified.

Based on the scanning electron micrograph of the tip apex structure (Fig. 1b in the main text)
 and the field distribution analyses shown in Supplementary Section 14, we have estimated the explicit
 value of d_{eff} . The bias-dependent TE-SHG/TE-SFG measurements (Figs. 3, 4d, and 4e in the main
 text) were performed by using a tip with ~ 50 -nm apex curvature (Fig. 1b in the main text), and the tip–
 substrate distance d employed in those experiments was ~ 7 Å. In such conditions, the effective tip–
 substrate distance d_{eff} , calculated by substituting the spatial distributions of $K_{\text{gap},z}(\mathbf{r}_n)$ (Figs. S20a
 and b) and the source position dependence of $L_{\text{gap}}(\mathbf{r}_n)$ (Fig. S23b) into Eq. R2, is approximately 9.7
 Å. This value is ~ 2.7 Å larger the actual tip–substrate distance $d = 7$ Å, which was used in the fitting
 analysis of the bias-dependent TE-SHG intensity profiles (Fig. 3) in the original manuscript. Thus, the
 estimates of the relative values of $\chi^{(2)}$ and $\chi^{(3)}$ are slightly modified. We have revised the
 manuscript to explicitly include these updated estimates. Please see the revisions listed below.

Furthermore, based on the revised discussion described above, we have also performed a
 sensitivity analysis showing how ± 0.5 -Å gap distance variations influence field-induced modulation
 behaviors. Specifically, we also calculated d_{eff} values for $d = 6.5$ Å and 7.5 Å conditions and
 obtained $d_{\text{eff}} = 9.2$ Å and 10.2 Å, respectively. Therefore, when the tip–substrate distance d
 fluctuates by ± 0.5 Å around 7 Å (Supplementary Fig. S11d), the effective tip–substrate distance d_{eff}
 also undergoes ± 0.5 -Å fluctuations around its average value of 9.7 Å. To examine the influence of

such fluctuations in d_{eff} , we numerically calculated relative modulation of the TE-SHG intensity
 ($\Delta I(V)/I(V=0)$) for three effective gap distances of $d_{\text{eff}} = 9.2, 9.7,$ and 10.2 \AA (Fig. II) by using
 the following equation:

$$234 \quad \frac{\Delta I_{\text{TESHG}}(V)}{I_{\text{TESHG}}(V=0)} = \frac{|\chi^{(3)}|^2}{|\chi^{(2)}|^2} \left(\frac{V}{d_{\text{eff}}}\right)^2 + \frac{2\text{Re}(\chi^{(3)}\chi^{(2)*})}{|\chi^{(2)}|^2} \left(\frac{V}{d_{\text{eff}}}\right). \quad (\text{R4})$$

As shown in Fig. II, within $(9.7 \pm 0.5) \text{ \AA}$ distance range, the distance-dependent variations in
 modulation depth are insignificant compared with the overall $\sim 2000\%$ modulation. This is because the
 amplitude of the distance variation ($\pm 0.5 \text{ \AA}$) is sufficiently smaller than the average value of d_{eff} (9.7
 \AA), leading to only minor variations in the coefficient of V^2 term in Eq. R4 ($|\chi^{(3)}|^2/|\chi^{(2)}|^2 d_{\text{eff}}^2$).
 Therefore, sub-angstrom fluctuations in the gap distance have an only minor influence on the measured
 electrophotonic modulation. Moreover, as described in our previous response letter, the timescale of
 the distance fluctuations during the measurements is on the order of microsecond scale, substantially
 shorter than minute-scale signal integration time in our measurements. Consequently, the influence of
 the fluctuations in the effective electrostatic field ($E_{\text{DC,eff}}$) would be time-averaged and would not
 manifest in the measured modulation curves. Based on these considerations, we can clearly rule out
 the possibility that $\pm 0.5\text{-\AA}$ variations in d affect the observed giant field-induced modulation
 behaviors.

 **Fig. II | Numerical analysis of gap distance sensitivity.** Calculated bias-dependent relative
 modulations of TE-SHG intensities (Eq. R4) for effective gap distances d_{eff} of 9.2 \AA (red), 9.7 \AA
 (orange), and 10.2 \AA (green), corresponding to $d = 6.5, 7.0, 7.5 \text{ \AA}$, respectively.

In addition to the above refined discussion on the effective tip–substrate distance d_{eff} and
 electrostatic field $E_{\text{DC,eff}}$ introduced for the analysis of the near-field EFISH effects, we have also
 reconsidered more appropriate theoretical treatments of the near-field enhancement factor ($K_{\text{gap}}(\mathbf{r})$)
 and the radiation efficiency ($L_{\text{gap}}(\mathbf{r})$) by explicitly incorporating their spatial distributions within the
 gap. In the earlier versions of the manuscript, the near-field intensity I_{gap} was treated as a
 representative constant value, taken from the field enhancement factor directly beneath the tip apex

$(I_{\text{gap}} = |K_{\text{gap}}|^2 I_0)$. Thus, the inherently non-uniform spatial distribution of $K_{\text{gap}}(\mathbf{r})$ (Supplementary
Fig. S20) was not explicitly taken into account in the analysis. Moreover, in our previous treatment,
L_{gap} was also treated as a representative constant value taken from the radiation efficiency of a single
dipole source placed directly beneath the tip apex. In realistic experimental conditions, however, the
position dependence of the radiation efficiency must be explicitly considered (Supplementary Fig.
S23). Therefore, both the field enhancement factor and the radiation efficiency should, in principle, be
evaluated by weighting contributions from the overall gap region, rather than adopting single
representative values sampled directly beneath the tip apex.

To verify the impact of position dependence of $K_{\text{gap}}(\mathbf{r})$ and $L_{\text{gap}}(\mathbf{r})$ on the tip-enhanced
nonlinear optical response, we have also explicitly incorporated the spatial distributions of both
$K_{\text{gap}}(\mathbf{r})$ and $L_{\text{gap}}(\mathbf{r})$ into our theoretical framework in Supplementary Section 15. In the presence
of the non-uniform near-field distributions, the near-field intensity is described by using the spatially
averaged mean-square value of $K_{\text{gap}}(\mathbf{r})$: $\langle I_{\text{gap}} \rangle = \langle |K_{\text{gap}}(\mathbf{r})|^2 \rangle I_0$ (Supplementary Eq. S10).
Furthermore, we have found that radiation efficiency is more appropriately described as its effective
value $L_{\text{gap,eff}}$, which corresponds to the weighted average of $L_{\text{gap}}(\mathbf{r})$ by the near-field distributions
within the gap (Supplementary Eq. S8). Although $\langle |K_{\text{gap}}(\mathbf{r})|^2 \rangle$ and $L_{\text{gap,eff}}$ are not explicitly
involved in Eq. R4 and thus not directly required in the fitting analysis of the bias-induced modulation
depth, these updated expressions are still essential for providing a proper physical interpretation of tip-
enhanced near-field nonlinear optical signals originating from multiple molecules within the gap. Thus,
we have revised Supplementary Section 15 to explicitly include these updated expressions.

Our revised theoretical framework has established physically valid formulations of tip-
enhanced nonlinear optical signals arising from a coherent sum of contributions from multiple
molecules, incorporating the spatial non-uniformity of both the optical near field and the electrostatic
field. To the best of our knowledge, such realistic theoretical examinations of near-field nonlinear
plasmonic responses have not been reported previously. Therefore, this refined framework not only
provides a clearer physical interpretation of our experimental observations but also establishes an
important foundation for future studies on electrophotonics modulations of nonlinear optical responses
in angstrom-scale plasmonic gaps. We sincerely thank the reviewer for this important and insightful
comment.

**Revision**

Pages 10–12, lines 289–328 in the main text: Notably, the optical near field E_{gap} and the electrostatic
field E_{DC} inherently exhibit non-uniform spatial distributions due to the curved geometry of the tip
apex (Supplementary Figs. S20 and S22), thereby giving rise to position-dependent nonlinear
polarizations within the nanoscale region beneath the tip apex. In such conditions, the TE-SHG electric
fields emitted from these individual nonlinear polarizations depend on their absolute spatial position,
and the experimentally observed TE-SHG intensity and its electrophotonic modulation arise from the

coherent sum of these position-dependent contributions. Specifically, the overall TE-SHG intensity
 (I_{TESHG}), including the contributions from both the bias-independent second-order signal and the E_{DC} -
 induced third-order signal, is given by the total intensity of SHG fields emitted from N molecules
 within the gap:

$$299 \quad I_{\text{TESHG}} \propto N^2 \left| \chi^{(2)} \langle L_{\text{gap}}(\mathbf{r}_n) K_{\text{gap},Z}(\mathbf{r}_n)^2 \rangle + \chi^{(3)} \left\langle \frac{L_{\text{gap}}(\mathbf{r}_n) K_{\text{gap},Z}(\mathbf{r}_n)^2}{d(\mathbf{r}_n)} \right\rangle V \right|^2 I_0^2, \quad (2)$$

where $\langle \dots \rangle$ denotes spatial averaging over molecules present within the gap region; \mathbf{r}_n is the
 position of n -th molecule; $d(\mathbf{r}_n)$ represents the position-dependent tip–substrate distance determined
 by the curvature radius of the tip apex; $K_{\text{gap},Z}(\mathbf{r}_n)$ is an enhancement factor of Z -directed field;
 $L_{\text{gap}}(\mathbf{r}_n)$ is efficiency of radiation from a nonlinear polarization located at the position \mathbf{r}_n ; and I_0 is
 incident field intensity (further details in Supplementary Section 15). To simplify this formulation, we
 introduce an effective tip-substrate distance d_{eff} , which satisfies the relation

$$306 \quad \left\langle \frac{L_{\text{gap}}(\mathbf{r}_n) K_{\text{gap},Z}(\mathbf{r}_n)^2}{d(\mathbf{r}_n)} \right\rangle = \frac{\langle L_{\text{gap}}(\mathbf{r}_n) K_{\text{gap},Z}(\mathbf{r}_n)^2 \rangle}{d_{\text{eff}}}. \quad (3)$$

Substituting this expression into Eq. 2 yields

$$308 \quad I_{\text{TESHG}} \propto N^2 \left| \chi^{(2)} + \chi^{(3)} \left(\frac{V}{d_{\text{eff}}} \right) \right|^2 |\langle L_{\text{gap}}(\mathbf{r}_n) K_{\text{gap},Z}(\mathbf{r}_n)^2 \rangle|^2 I_0^2. \quad (4)$$

By defining V/d_{eff} term as $E_{\text{DC,eff}}$ and omitting N^2 and $|\langle L_{\text{gap}}(\mathbf{r}_n) K_{\text{gap},Z}(\mathbf{r}_n)^2 \rangle|^2 I_0^2$ terms
 for simplicity, we arrive at the simplified expression of EFISH similar to that given in Eq. 1. Notably,
 d_{eff} defined in Eq. 3 represents an effective spatial average of the tip–substrate distance within the
 gap, weighted by the near-field intensity distribution and the curvature of the tip apex. Thus, the
 expression of $E_{\text{DC,eff}} = V/d_{\text{eff}}$ effectively incorporates the influences of spatial field inhomogeneity.
 Moreover, even in the presence of such spatial inhomogeneity, the effective electrostatic field $E_{\text{DC,eff}}$
 remains linearly proportional to the bias voltage V . Consequently, the overall TE-SHG intensity still
 exhibits a quadratic dependence on V , and thus the experimentally observed parabolic voltage
 dependence of TE-SHG intensity (Fig. 3) is a physically valid behavior.

Based on Eq. 4, the bias-dependent modulation of TE-SHG intensity is expressed as

$$319 \quad \frac{\Delta I_{\text{TESHG}}(V)}{I_{\text{TESHG}}(V=0)} = \frac{|\chi^{(3)}|^2}{|\chi^{(2)}|^2} \left(\frac{V}{d_{\text{eff}}} \right)^2 + \frac{2\text{Re}(\chi^{(3)}\chi^{(2)*})}{|\chi^{(2)}|^2} \left(\frac{V}{d_{\text{eff}}} \right). \quad (5)$$

The fitting analysis of the bias-dependent intensity change shown in Fig. 3 using this equation yielded
 the coefficients of V^2 and V in Eq. 5 as $|\chi^{(3)}|^2/d_{\text{eff}}^2|\chi^{(2)}|^2 = (21.1 \pm 0.7) \text{ V}^{-2}$ and $2\text{Re}(\chi^{(3)}\chi^{(2)*})/$
 $d_{\text{eff}}|\chi^{(2)}|^2 = (-3.9 \pm 0.6) \text{ V}^{-1}$, respectively. Moreover, for a tip apex curvature radius of $\sim 50 \text{ nm}$ (Fig.
 1b) and an actual tip–substrate distance d of $\sim 7 \text{ \AA}$, d_{eff} is determined to be 9.7 \AA (further details in
 Supplementary Sections 14 and 15). Based on these estimations, the relative values of $\chi^{(2)}$ and $\chi^{(3)}$
 were derived as $|\chi^{(3)}|^2/|\chi^{(2)}|^2 = (19.9 \pm 0.7) \text{ nm}^2/\text{V}^2$ and $2\text{Re}(\chi^{(3)}\chi^{(2)*})/|\chi^{(2)}|^2 = (-3.8 \pm 0.6)$

326 nm/V. From these relative values, we can also estimate the threshold field ($E_{\text{DC},0}$), at which $\chi^{(2)}$ and
327 $\chi^{(3)}E_{\text{DC}}$ effects equally contribute to the nonlinear optical generation ($|\chi^{(3)}E_{\text{DC},0}| = |\chi^{(2)}|$), to be
2.2×10^8 V/m. Using the effective gap distance value of $d_{\text{eff}} = 9.7 \text{ \AA}$, the corresponding applied
voltage is estimated to be ± 0.21 V.

Page 11, lines 332–335 in the main text: This is because our numerical analysis to examine the
distance-sensitivity of EFISH effects revealed that sub-angstrom variations in the gap distance have
an only minor influence on the measured electrophotonic modulation (Supplementary Fig. S24).

Page 21, lines 504–512 in Supplementary Section 13: Note that since the amplitudes of K_{gap} , L_{gap} ,
and E_{DC} inherently depend on spatial positions within the gap (see Section 14 for the spatial
distributions of these factors), the overall TE-SHG intensity should be evaluated by weighting
contributions from individual nonlinear polarizations generated over the whole gap region. However,
to provide a simple physical understanding of the fundamental mechanisms of the TE-SHG process,
we here exclusively focus on K_{gap} and L_{gap} values sampled at a single point directly beneath the tip
apex, without considering their spatial distributions over the gap region. More rigorous theoretical
treatments of TE-SHG intensities that explicitly incorporate the spatial distributions of K_{gap} , L_{gap} ,
and E_{DC} within the gap are provided in Section 15.

Page 29, lines 731–740 in Supplementary Section 14: In our TE-SHG/TE-SFG experiments, the
detected signal corresponds to the coherent summation of the radiation fields emitted from such
position-dependent nonlinear polarizations generated within the gap. As discussed in Section 13, the
efficiency of this radiation process is described by the factor L_{gap} . Importantly, owing to the curved
geometry of the tip apex, the plasmonic enhancement efficiency exhibits position dependence within
the gap. As a result, the radiation efficiencies of individual nonlinear polarizations also depend on their
positions \mathbf{r} : $L_{\text{gap}} \equiv L_{\text{gap}}(\mathbf{r})$. We have examined this source-position dependence of $L_{\text{gap}}(\mathbf{r})$ by
numerically calculating the radiation from dipole sources placed at different lateral positions across
the surface (Fig. S23a). As displayed in Fig. S23b, dipole signal radiation primarily occurs in a region
with a radius of approximately 10 nm from the tip apex position.

Pages 33–35, lines 820–894 in Supplementary Section 15: In Section 13, we overviewed the spectral
properties of both the field enhancement factor within the tip–substrate gap (K_{gap}) and the radiation
efficiency from a single emitter placed within the gap region (L_{gap}). Although the discussion in Section
13 provides physically important interpretation regarding the fundamental mechanisms of TE-SHG
and TE-SFG processes, it still disregards the presence of the inherent spatial non-uniformity in K_{gap} ,
L_{gap} , and E_{DC} demonstrated in Section 14. In the actual tip–substrate gap, spatially varying K_{gap} and
E_{DC} (Figs. S20 and S22, respectively) produce position-dependent nonlinear polarizations within the

nanoscale region, and radiation from each polarization occurs with different efficiencies depending on
 its absolute position (Fig. S23). The experimentally observed near-field nonlinear optical signals
 correspond to the coherent sum of these individual contributions. Here, we demonstrate that even in
 the presence of such spatial distributions, the overall TE-SHG intensity can be reduced to the simple
 form given in Eq. 1 in the main text by considering the spatially averaged signal intensity within the
 near-field enhancement region.

The overall TE-SHG intensity (I_{TESHG}), including the contributions from both the bias-
 independent second-order field and E_{DC} -induced third-order field, is given by the total intensity of
 SHG fields emitted from N molecules within the gap:

$$I_{\text{TESHG}} \propto \left| E_{\text{TESHG}}^{(2)} + E_{\text{TESHG}}^{(3)} \right|^2$$

$$= N^2 \left| \chi^{(2)} \langle L_{\text{gap}}(\mathbf{r}_n) K_{\text{gap},z}(\mathbf{r}_n)^2 \rangle + \chi^{(3)} \left\langle \frac{L_{\text{gap}}(\mathbf{r}_n) K_{\text{gap},z}(\mathbf{r}_n)^2}{d(\mathbf{r}_n)} \right\rangle V \right|^2 I_0^2, \quad (\text{S5})$$

where $\langle \dots \rangle$ denotes spatial averaging over the molecules present within the gap region; $\chi^{(2)}$ and $\chi^{(3)}$
 are second- and third-order nonlinear susceptibilities, respectively; \mathbf{r}_n is the position of n -th
 molecule; $d(\mathbf{r}_n)$ represents the position-dependent tip-substrate distance determined by the curvature
 radius of the tip apex; V denotes the applied STM bias voltage; and I_0 is incident field intensity. To
 simplify this formulation, we introduce an effective tip-substrate distance d_{eff} , which satisfies the
 relation

$$\left\langle \frac{L_{\text{gap}}(\mathbf{r}_n) K_{\text{gap},z}(\mathbf{r}_n)^2}{d(\mathbf{r}_n)} \right\rangle = \frac{\langle L_{\text{gap}}(\mathbf{r}_n) K_{\text{gap},z}(\mathbf{r}_n)^2 \rangle}{d_{\text{eff}}}. \quad (\text{S6})$$

Substituting this expression into Eq. S5 yields

$$I_{\text{TESHG}} \propto N^2 \left| \chi^{(2)} + \chi^{(3)} \left(\frac{V}{d_{\text{eff}}} \right) \right|^2 \left| \langle L_{\text{gap}}(\mathbf{r}_n) K_{\text{gap},z}(\mathbf{r}_n)^2 \rangle \right|^2 I_0^2. \quad (\text{S7})$$

Then, we define V/d_{eff} term as $E_{\text{DC,eff}}$, which corresponds to a spatially averaged effective intragap
 electrostatic field incorporating both the non-uniform field distribution in the gap and the curved
 geometry of the tip apex. Furthermore, we introduce an effective radiation efficiency $L_{\text{gap,eff}}$ defined
 as

$$\langle L_{\text{gap}}(\mathbf{r}_n) K_{\text{gap},z}(\mathbf{r}_n)^2 \rangle = L_{\text{gap,eff}} \langle K_{\text{gap},z}(\mathbf{r}_n)^2 \rangle, \quad (\text{S8})$$

where $L_{\text{gap,eff}}$ corresponds to the weighted average of $L_{\text{gap}}(\mathbf{r}_n)$ by $K_{\text{gap},z}(\mathbf{r}_n)^2$. Using these two
 effective values ($E_{\text{DC,eff}}$ and $L_{\text{gap,eff}}$) yields

$$I_{\text{TESHG}} \propto N^2 \left| L_{\text{gap,eff}} \right|^2 \left| \chi^{(2)} + \chi^{(3)} E_{\text{DC,eff}} \right|^2 \left| \langle K_{\text{gap},z}(\mathbf{r}_n)^2 \rangle \right|^2 I_0^2. \quad (\text{S9})$$

This equation shows that the spatially averaged near-field intensity within the gap should be given by
 mean-square intensity:

$$\langle I_{\text{gap}} \rangle = \left| \langle K_{\text{gap},z}(\mathbf{r}_n)^2 \rangle \right| I_0. \quad (\text{S10})$$

Using this average near-field intensity and omitting $\left| L_{\text{gap,eff}} \right|^2$ constant from Eq. S9 for simplicity,

we arrive at the simplified expression for EFISH similar to that given in Eq. 1 in the main text:

$$394 I_{\text{TESHG}} \propto |\chi^{(2)} + \chi^{(3)} E_{\text{DC,eff}}|^2 \langle I_{\text{gap}} \rangle^2. \quad (\text{S11})$$

Notably, while this equation incorporates the spatial distributions of both the optical near field and the
electrostatic field, it is still described by a formally equivalent expression for the EFISH effect under
homogeneous electrostatic field. Moreover, even within this refined formalism, $E_{\text{DC,eff}}$ is still
proportional to V ($E_{\text{DC,eff}} = V/d_{\text{eff}}$), and thus I_{TESHG} in Eq. S11 quadratically depends on V . This
validates the fitting analysis using a quadratic function presented in Figs. 3 and 4e in the main text.

Based on the scanning electron micrograph of the tip apex structure (Fig. 1b in the main text)
and the field distribution analyses shown in Section 14, we can estimate the explicit value of d_{eff} . The
bias-dependent TE-SHG/TE-SFG measurements (Figs. 3, 4d, and 4e in the main text) were performed
by using a tip with ~ 50 -nm apex curvature (Fig. 1b in the main text), and the tip–substrate distance d
employed in those experiments was ~ 7 Å. In such conditions, the effective tip–substrate distance d_{eff} ,
calculated by substituting the spatial distributions of $K_{\text{gap},z}(\mathbf{r}_n)$ (Figs. S20a and b) and the source
position dependence of $L_{\text{gap}}(\mathbf{r}_n)$ (Fig. S23b) into Eq. S6, is approximately 9.7 Å, indicating that d_{eff}
is larger than d by ~ 2.7 Å. This d_{eff} value was used to obtain the estimates of the relative values of
$\chi^{(2)}$ and $\chi^{(3)}$ presented in the main text.

As shown in Fig. S11d, the excitation light irradiation at the tip apex induces ~ 0.5 -Å
fluctuations in the tip–substrate distance d . These fluctuations lead to corresponding variations in both
the effective distance d_{eff} and the effective electrostatic field $E_{\text{DC,eff}}$, resulting in temporal
fluctuations in the overall TE-SHG intensity. To quantitatively examine the influences of such
fluctuations, we additionally calculated d_{eff} values for $d = 6.5$ Å and 7.5 Å conditions and obtained
$d_{\text{eff}} = 9.2$ Å and 10.2 Å, respectively. Therefore, when the tip–substrate distance d fluctuates by ± 0.5
Å around 7 Å (Fig. S11d), the effective tip–substrate distance d_{eff} also undergoes ± 0.5 -Å fluctuations
around its average value of 9.7 Å. The influences of such variations in d_{eff} were examined by
numerically calculating the relative modulation of the TE-SHG intensity ($\Delta I_{\text{TESHG}}(V)/I_{\text{TESHG}}(V=0)$,
Eq. 5 in the main text) for three effective gap distances of $d_{\text{eff}} = 9.2, 9.7,$ and 10.2 Å (Fig. S24).
Notably, within such distance range around 9.7 Å, the distance-dependent variations in modulation
depth are sufficiently small compared with the overall $\sim 2000\%$ modulation (Fig. S24). This is because
the amplitude of the distance variation (± 0.5 Å) is sufficiently smaller than the average value of d_{eff}
(9.7 Å), leading to only minor variations in the coefficient of V^2 term in Eq. 5 in the main text
($|\chi^{(3)}|^2/|\chi^{(2)}|^2 d_{\text{eff}}^2$). Therefore, sub-angstrom fluctuations in the gap distance have an only minor
influence on the measured electrophotonic modulation. Additionally, as described in the main text, the
timescale of the distance fluctuations during the measurements is on the order of microsecond scale,
substantially shorter than minute-scale signal integration time in our measurements. Consequently, the
influence of the fluctuations in the effective electrostatic field ($E_{\text{DC,eff}}$) would be time-averaged and
would not manifest in the measured modulation curves. Based on these considerations, we can clearly
rule out the possibility that ± 0.5 -Å variations in d (Fig. S11d) affect the observed giant field-induced

modulation behaviors.

**Reviewer #4's comment 4**

4) Follow-up to Comment (e) – Origin of Nonlinear Peaks: Thank you for outlining the general three-
step process of near-field nonlinear emission. However, the claim that all processes are predominantly
dipolar in nature would benefit from more direct or quantitative justification.

Given the Å-scale confinement and strong field gradients in the gap, higher-order modes (e.g.,
quadrupolar gap plasmons) could feasibly contribute to both the nonlinear polarization and radiated
signal. I recommend the authors consider:

Providing simulation results (FEM or FDTD) showing the spatial and spectral distribution of field
enhancements, or

A multipole decomposition or theoretical argument that supports dipolar dominance under their
conditions, or

Empirical tests (e.g., polarization, angle, or tip-shape dependence) that would suppress or enhance
higher-order modes.

This would significantly improve the mechanistic understanding of the TE-SHG/SFG signals and
bolster the claim of dipolar origin.

**Reply to reviewer #4's comment 4**

Thank you for your valuable comments. If strong field gradients are present within the gap,
quadrupolar and even higher-order molecular polarizations and gap plasmons should begin to play
roles. However, such steep field gradients are typically realized only when the tip is atomically
sharpened or when an atomic-scale protrusion is present at the apex [*J. Phys. Chem. C* **128**, 15985
(2024); *J. Chem. Phys.* **154**, 214706 (2021)]. In contrast, as shown in Figs. 1b and S15a–c, the tips
used in our experiments have larger apex radii on the order of several tens of nanometers, making the
characteristic length scale of the field variations within the gap significantly larger than the size of a
single MBT molecule (~6 Å) (Supplementary Fig. S20). Under such relatively gentle field gradients,
a dipolar approximation becomes valid for molecular optical transitions, and thus the induced
nonlinear polarizations in molecules are expected to be predominantly dipolar. Notably, this dipolar
dominance has been thoroughly verified in our recent publication by theoretically examining the dipole
and quadrupole contributions involved in molecular nonlinear polarizations within an angstrom-scale
tip–substrate gap [*J. Phys. Chem. C published*. DOI: 10.1021/acs.jpcc.5c05411]. In this recent study,
we also measured vibrationally resonant TE-SFG spectra and demonstrated that $\text{Im}(\chi^{(2)})$ signals of
surface-adsorbed molecules are governed by dipole-dominated features [*J. Phys. Chem. C published*.
DOI: 10.1021/acs.jpcc.5c05411]. To convey such dipolar dominance more clearly, we have revised
the manuscript to remark that the contribution of such higher-order terms is negligible and add citation
to our recent publication [*J. Phys. Chem. C published*. DOI: 10.1021/acs.jpcc.5c05411]. We believe

that consulting this reference will provide a deeper understanding of the dipolar dominance in the
 nonlinear polarizations relevant to the present study.

Furthermore, the influence of dipolar and quadrupolar plasmonic modes on the signal radiation
 process has already been comprehensively discussed in a previous work [*Nat. Commun.* **11**, 1021
 (2020)]. Figure IIIa illustrates theoretical far-field radiation spectra of Au nanosphere–Ag substrate
 nanocavities calculated for various Au nanosphere radii. As the reviewer mentioned, nanospheres with
 extremely small radii ($R < 5$ nm) exhibit a quadrupolar plasmonic mode (Fig. IIIc) at higher energies
 (~ 3.1 eV), as well as a dipolar mode (Fig. IIIb) at lower energies (~ 2.3 eV). However, for the larger
 nanosphere radius of > 10 nm, only the low-energy (dipolar) peak becomes apparent with substantially
 minor intensities of the high-energy (quadrupolar) mode. This indicates that the excitation of the
 quadrupolar plasmons is negligible for the larger tip apex on the order of several tens of nanometers.
 Since the tip apex employed in our experiments (Fig. 1b in the main text) falls into this dipole-
 dominated regime, the influence of quadrupolar plasmonic modes on the radiation process should be
 negligibly small under our experimental conditions.

Based on the above discussion, we can reasonably conclude that only the dipolar modes play
 dominant roles in the generation of nonlinear polarization and radiation of nonlinear optical signals.
 To clarify this dipolar dominance, we have revised our manuscript and Supplementary Information to
 include the above discussion. Please see the revisions listed below. Thank you for your comments.

 **Fig. III | Modelling the tip-surface nanocavity** [*Nat. Commun.* **11**, 1021 (2020)]. **a** Theoretical far-
 field (main panels) and photonic density of optical states (insets) spectra for nanocavities comprising
 a gold sphere (tip) on top of a silver flat surface (substrate) calculated for various sphere radii (R) at
 the fixed gap size of 0.5 nm. **b, c** Induced charge distribution (left) and electric field amplitude maps
 (right). The two radiative plasmonic modes with lower and higher energies indicated by grey arrows
 in **a** are illustrated in panels **b** and **c**, respectively.

**Revision**

Page 9, lines 256–260 in the main text: the generated nonlinear polarizations are predominantly dipolar
in nature, with negligible contributions from quadrupolar or higher-order multipolar components
(further details in Supplementary Section 14)⁶³. Moreover, since such relatively large tip apex can
support only dipolar plasmonic mode (Supplementary Fig. S21)⁷⁴, TE-SHG signal radiation observed
in Figs. 2 and 3 should be mediated by dipolar plasmons (further details in Supplementary Sections 13
and 14).

Page 23, lines 587–594 in Supplementary Section 13: Note that the influence of dipolar and
quadrupolar plasmonic modes on the signal radiation process has been comprehensively discussed in
a previous work²⁷. Although a sharpened tip with an apex radius of < 5 nm supports a quadrupolar
plasmonic mode, a larger tip with a radius of > 10 nm exhibits only dipolar plasmonic mode²⁷,
indicating that the excitation of the quadrupolar plasmons is negligible for the larger tip apex on the
order of several tens of nanometers. Since ~30-nm tip apex employed in our experiments (Fig. 1b in
the main text) falls into this dipole-dominated regime, the near-field radiation process in this work
should be mediated predominantly by dipolar gap-mode plasmons.

Page 28, lines 716–725 in Supplementary Section 14: The ~20-nm length scale of the field variations
within the gap is significantly larger than the size of a single MBT molecule (~6 Å). In such conditions,
contributions of quadrupolar and even higher-order molecular polarizations should be minor, and thus
the induced nonlinear polarizations in molecules are expected to be predominantly dipolar. Notably,
this dipolar dominance has been thoroughly verified in our recent publication by theoretically
examining the dipole and quadrupole contributions involved in molecular nonlinear polarizations
within an angstrom-scale tip–substrate gap¹⁰. In this recent study, we also measured vibrationally
resonant TE-SFG spectra and demonstrated that $\text{Im}(\chi^{(2)})$ signals of surface-adsorbed molecules are
governed by dipole-dominated features¹⁰. Therefore, in the present tip conditions, a dipolar
approximation is valid for molecular optical transitions, and the influences of quadrupolar or higher-
order polarizations are negligible.

**Reviewer #4's comment 5**

>To further improve the TE-SHG and TE-SFG efficiencies, it is essential to tailor the enhancement
spectrum such that the value of $|K_{\text{gap}}K_{\text{gap}}L_{\text{gap}}|^2$ terms in equations (S3) and (S4) are maximized.

It would be better state as follows $|K_{\text{gap}}(\omega_1)K_{\text{gap}}(\omega_2)L_{\text{gap}}|^2$.

**Reply to reviewer #4's comment 5**

In accordance with the reviewer's comment, we have revised Supplementary Information to explicitly

clarify the frequency parameters of K_{gap} and L_{gap} . Thank you for your suggestion.

**Revision**

Page 24, lines 617–619 in Supplementary Section 13: To further improve the TE-SHG and TE-SFG
efficiencies, it is essential to tailor the enhancement spectrum such that the value of
$|K_{\text{gap}}(\omega_1)K_{\text{gap}}(\omega_2)L_{\text{gap}}(\omega_1 + \omega_2)|^2$ terms in Eqs. S3 and S4 are maximized.

**Reviewer #4's comment 6**

>I also appreciate the authors' thoughtful response to my concerns regarding quantum plasmonic
suppression and nonlocality. Their revised interpretation and supporting references now reflect the
state-of-the-art understanding in this regime. However, I encourage the authors to make this
clarification more visible in the main text, not just the Supplementary. Additionally, it would be
valuable to indicate whether their modeling framework explicitly incorporates nonlocal corrections or
if the cited works are used purely as qualitative support.

**Reply to reviewer #4's comment 6**

We sincerely thank the reviewer for the constructive and encouraging feedback on our revised
interpretation regarding quantum plasmonic suppression and nonlocality. As detailed in our previous
response letter, we have already revised not only Supplementary Information but also the main text
(lines 160–173) to clarify the potential contributions of quantum plasmonic suppression and nonlocal
effects (see “Reply to Reviewer #4's comments 2 and 4” in our previous response letter).

Moreover, it should be noted that while the quantum plasmonic effects in an angstrom-scale
junction has been discussed primarily by referring to previous studies as qualitative support [Ref. 68,
71, 72, 75–78 in the main text], we would like to emphasize that the presence of such effects can also
be ensured from our experimental observations. Specifically, the bias-dependent TE-SHG intensities
measured under constant-distance conditions (Fig. 3) closely match those obtained under constant-
current conditions (Fig. S14b), where the tip–substrate distance varies in the range of 4–6 Å during the
measurement. This agreement clearly indicates that the overall field-enhancement strength remains
essentially insensitive to the tip–substrate distance within the experimentally adopted 4–6 Å distance
range. Therefore, the presence of quantum plasmonic effects in our experiments is supported not only
by previous studies [Refs. 68, 71, 72, 75–78 in the main text] but also by our experimental results. We
believe that providing such discussion in the main text will help us further strengthen the reliability of
our experimental findings. We greatly appreciate the reviewer's insightful comments.

**Revision**

Page 6, lines 165–168 in the main text: Particularly, substantially short tip–substrate distance of $< 4 \text{ \AA}$
leads to a regime where such quantum plasmonic quenching becomes dominant, resulting in a steep

decrease in the electric field enhancement factors^{68,71,72,75–78}. In contrast, the 4–7 Å regime represents
a crossover region just before this steep decline.

Pages 7–8, lines 199–213 in the main text: It should be noted that while the quantum plasmonic effects
in an angstrom-scale junction has been discussed primarily by referring to previous studies^{68,71,72,75–78}
as qualitative support, the presence of such quantum plasmonic effects can also be ensured from our
experimental observations. As shown in Supplementary Fig. S14, voltage-controlled quadratic
modulation of TE-SHG, similar to that shown in Fig. 3, was also observed under the constant tunneling
current conditions, where the tip–substrate distance varies in the range of 4–6 Å during the voltage
dependent measurement. This clearly demonstrates that the overall field-enhancement strength
remains essentially insensitive to the tip–substrate distance within the experimentally varied 4–6 Å
distance range. Therefore, the presence of quantum plasmonic effects that cause distance-invariant
field enhancement behavior is supported not only by previous studies^{68,71,72,75–78} but also by our
experimental results (Figs. 3 and S14). Moreover, the agreement of the bias-dependent profiles shown
in Figs. 3 and S14 also indicates negligible contributions of tunneling current to the observed variations
in the TE-SHG intensity (Fig. 3). Consequently, we can reasonably conclude that the observed
electrical modulation behavior is predominantly governed by voltage-induced electrophotonic effects,
distinct from variations in the near-field enhancement or contributions from the tunneling current.

Reviewer #5

**Reviewer #5's comment**

My concerns have been addressed, and the manuscript has been obviously improved. I would like to
recommend it for publication now.

**Reply to reviewer #5's comment**

We sincerely thank the reviewer for acknowledging the improvement of our manuscript and
recommending it for publication in *Nature Communications*.